# Benign Oscillation of Stochastic Gradient Descent with Large Learning Rates

**Miao Lu\*[1], Beining Wu\*[2], Xiaodong Yang[3], Difan Zou[4]**
[1]Stanford University, [2]University of Chicago, [3]Harverd University, [4]University of Hong Kong
`miaolu@stanford.edu.cn, beiningw@uchicago.edu`
`xyang@g.harvard.edu, dzou@cs.hku.hk`

## Abstract

In this work, we theoretically investigate the generalization properties of neural networks (NN) trained by stochastic gradient descent (SGD) with *large learning rates*. Under such a training regime, our finding is that, the *oscillation* of the NN weights caused by SGD with large learning rates turns out to be beneficial to the generalization of the NN, potentially improving over the same NN trained by SGD with small learning rates that converges more smoothly. In view of this finding, we call such a phenomenon "*benign oscillation*". Our theory towards demystifying such a phenomenon builds upon the *feature learning* perspective of deep learning. Specifically, we consider a feature-noise data generation model that consists of (i) *weak features* which have a small $\ell_2$-norm and appear in each data point; (ii) *strong features* which have a large $\ell_2$-norm but appear only in a certain fraction of all data points; and (iii) noise. We prove that NNs trained by oscillating SGD with a large learning rate can effectively learn the weak features in the presence of those strong features. In contrast, NNs trained by SGD with a small learning rate can only learn the strong features but make little progress in learning the weak features. Consequently, when it comes to the new testing data points that consist of only weak features, the NN trained by oscillating SGD with a large learning rate can still make correct predictions, while the NN trained by SGD with a small learning rate could not. Our theory sheds light on how large learning rate training benefits the generalization of NNs. Experimental results demonstrate our findings on the phenomenon of "*benign oscillation*".

## 1 Introduction

While deep neural networks (NNs) have achieved tremendous empirical success in various domains including images, language processing, decision-making, etc, the theoretical understanding of deep learning is still far behind satisfactory, especially the relationships between optimization of the NN and its generalization. From the viewpoint of optimization, using *large learning rates* in NN training has been empirically shown to be of vital importance for generalization (He et al., 2016; Xing et al., 2018; Damian et al., 2022; Kaur et al., 2023). Nevertheless, a principled theoretical understanding of the mechanism behind the benefits of large learning rate training still remains limited.

To better capture the key ingredients in the training dynamics of stochastic gradient descent (SGD) with large learning rates, we train a ResNet (He et al., 2016) using SGD with small and large learning rates and present the training and testing results in Figure 1. When using a large learning rate SGD, we can observe an "oscillating" training curve, i.e., the training loss fluctuates at different iterations (generally this happens only when the learning rate exceeds the inverse of the objective smoothness), while for small learning rate SGD, the training curve is smooth and converges rapidly. On the other hand, the smooth convergence in training loss cannot bring any benefit for the test accuracy – SGD with large learning rates achieves a significantly higher test accuracy than SGD with small learning rates. These empirical observations suggest that the *oscillation* during training can be closely tied to the better generalization performance achieved by SGD with large learning rates.

In this paper, we study the learning dynamics of SGD with large learning rates by investigating the *oscillation* happening during the optimization process, and explain its benefit to the generalization

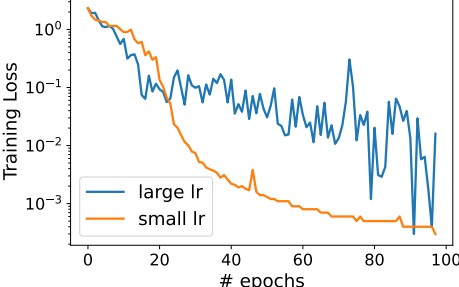 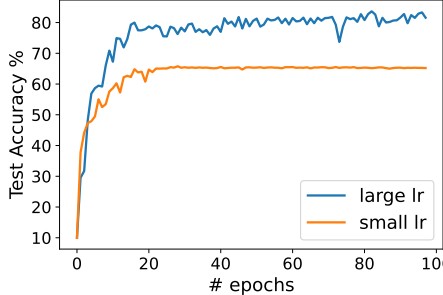

Figure 1: Training and test performance of ResNet-18 on CIFAR-10 dataset, when trained via SGD with small and large learning rates ($\eta = 0.01$ vs. $\eta = 0.75$). We adopt the same configuration as in Andriushchenko et al. (2023): using weight decay but no momentum and no data augmentation. A clear difference between the large learning rate training and small learning rate training can be observed: SGD with a large learning rate leads to an "oscillating" training curve with higher testing accuracy; SGD with a small learning rate has a rapid and smooth convergence but gives lower testing accuracy.

performance. The key message is that compared to the smooth convergence achieved by SGD with small learning rates, *the oscillation prevents the over-greedy convergence and serves as the engine that drives the learning of less-prominent data patterns*. These data patterns would be beneficial for the NN to generalize well on unseen testing data. Thus we explain from the theoretical side why large learning rate training can help NN to generalize better in practice.

Our investigation of SGD with large learning rates for NN training builds on the feature learning perspective of deep learning theory (Allen-Zhu and Li, 2022), which explicitly considers data models consisting of different types of features and noise. For the sake of our goal, we devise a feature-noise data model consisting of two types of features that have different strengths and different distributions. Based upon the new data model, by carefully tracking the process of feature learning of a NN trained by SGD with large or small learning rates, we prove that only when trained with large learning rates can the NN effectively learn the key features for generalizing to *each* new data point. The NN trained by small learning rate SGD fails to generalize to certain testing data because of the limited learning of the features which are crucial to the generalization to those new data points.

To explain such a phenomenon, our theory identifies the core incentives for the superior performance of learning the key features with large learning rate SGD as the *oscillation* during NN training, which is also related to the regime of "*edge of stability*" (Cohen et al., 2020). Intuitively, the oscillation can prevent over-greedy convergence which could only leverage the most prominent components of the data, thus allowing for all the useful components to be discovered and learned by gradient descent. In view of our finding that indicates oscillating NN training with large learning rates possibly resulting in better generalization, we refer to such a phenomenon as "*benign oscillation*".

## 1.1 Our Contributions

**Dynamic analysis framework for SGD with large learning rates.** We provide a theoretical framework to understand and explain the oscillation in NN training via SGD with a large learning rate. Specifically, we consider a feature-noise data generation model consisting of two types of features – the *strong features* and the *weak features* – that have different strengths and distributions to capture our core ideas towards explaining the relationships between large learning rate SGD training and generalization. Then, our theoretical framework establishes a sharp characterization of the training dynamics of these features and noises, based on which we can precisely analyze the generalization of NN trained by SGD with small or large learning rates. We remark that in general studying the NN optimization dynamics when the learning rate is greater than twice inversed smoothness is quite challenging, and our theoretical analysis framework based upon the feature-noise model potentially provides useful guidance which can be leveraged to study other nonconvex optimization problems.

**A new theoretical argument for feature learning driven by oscillation.** The key to explaining the large learning rate training regime is a new theory on learning the weak features driven by oscillation. As we illustrate in Section 3, the oscillation of the NN values (predictions) around the target (label) does *not* cancel with each other. Instead, the fluctuations accumulate linearly over time. This further serves as the engine driving the learning of the weak features, resulting in better generalization. This characterizes the distinctive training dynamics of SGD under the large learning rate training regime, revealing the benefits of the oscillation in learning useful data patterns.

**Division for generalization by different learning rates.** In contrast to effectively learning the weak features by large learning rate oscillating training, we also show that the smooth and rapid convergence achieved by SGD with small learning rates would *not* help NN learn the weak features, thus being unable to generalize to the new data without strong features. This gives a division of the generalization property of NNs trained by large and small learning rates.

## 2 PROBLEM SETTING

In this section, we introduce the theoretical setting for our investigation of generalization properties of SGD through the task of binary classification. We first introduce the multi-view data generation model and then define the two-layer convolutional neural network and the SGD algorithm.

**Data generation model.** We let $\mathbf{v} \perp \mathbf{u} \in \mathbb{R}^d$ be two fixed vectors, denoting the signal (or feature) part shared by each data point. Then each data point, denoted by $(\mathbf{x}, y)$ where $\mathbf{x} = (\mathbf{x}^{(1)}, \mathbf{x}^{(2)}, \mathbf{x}^{(3)})$ contains 3 patches, is generated as following: let $y \in \{1, -1\}$ be independently generated according to $\mathbb{P}(y = 1) = \mathbb{P}(y = -1) = 1/2$, and

- **Weak signal patch.** one patch of $\mathbf{x}$ is taken by the weak signal $y \cdot \mathbf{v}$;
- **Strong signal patch.** with probability $1 - \rho$, one patch of $\mathbf{x}$ that is different from $y \cdot \mathbf{v}$, is taken by the strong signal $y \cdot \mathbf{u}$, where $\rho \in (0, 1/2)$ is the probability.
- **Noise patch.** all remaining patches are taken by independent Gaussian noise $\boldsymbol{\xi} \sim N(0, \sigma_p^2(\mathbf{I}_d - \mathbf{v}\mathbf{v}^\top/\|\mathbf{v}\|_2^2 - \mathbf{u}\mathbf{u}^\top/\|\mathbf{u}\|_2^2))$ for some variance $\sigma_p > 0$.

For simplicity, we refer to the data with strong signal as the *strong data*, denoted by $((y\mathbf{u}, y\mathbf{v}, \boldsymbol{\xi}), y)$, and we refer to the data with only weak signal as the *weak data*, denoted by $((\widetilde{\boldsymbol{\xi}}, y\mathbf{v}, \boldsymbol{\xi}), y)$. Here by "strong", we mean a vector with a larger $\ell_2$-norm, as we would specify in the theory part. Intuitively, the weak signal $y \cdot \mathbf{v}$ can be interpreted as the invariant and common signals across data like the shape of key objects in an image. The strong signal $y \cdot \mathbf{u}$ can be understood as the background or the domain information which is stronger but only appears in a certain fraction of all data points. This indicates that in order for a classifier to generalize to new data, it must effectively learn the weak signal.

Our proposed data generation model is adapted from the feature-learning-based line of research on deep learning (Allen-Zhu and Li, 2022; Cao et al., 2022; Zou et al., 2023), and it can serve as a good theoretical platform to explain the relationships between oscillating NN training with large learning rates and NN generalization. Finally, we remark that this data model can be extended for generality, e.g., multiple features, more patches, multi-class data. In fact, as long as the signal and noise patches have properly different strength and fractions, our theoretical analysis can be directly applied.

**Two-layer CNN.** We consider a two-layer convolutional neural network (CNN) with filters applied to the three patches separately. We assign the parameters of the second layer of the CNN to a fixed $+1$ and $-1$, respectively. Formally, the CNN $f(\cdot; \mathbf{W}) : \mathbb{R}^{3d} \mapsto \mathbb{R}$ is defined as

$$f(\mathbf{x}; \mathbf{W}) = \sum_{j \in \{\pm 1\}} j F_j(\mathbf{x}; \mathbf{W}_j), \quad F_j(\mathbf{x}; \mathbf{W}_j) = \frac{1}{m} \sum_{r \in [m]} \sum_{p=1}^{3} \sigma(\langle \mathbf{w}_{j,r}, \mathbf{x}^{(p)} \rangle), \quad (1)$$

where $m \in \mathbb{N}_+$ is the number of filters (i.e., neurons), $\sigma(z) = (\max\{z, 0\})^2$ is the ReLU$^2$ activation function, and $\mathbf{w}_{j,r} \in \mathbb{R}^d$ denotes the weights of the $r$-th neuron of $F_j$. We use $\mathbf{W} = \{\mathbf{W}_j\}_{j \in \{\pm 1\}}$ and $\mathbf{W}_j = \{\mathbf{w}_{j,r}\}_{r \in [m]}$ to denote the collection of the weights.

**Loss function and stochastic gradient descent (SGD).** Having access to $n$ i.i.d. samples from the data generation model, $\mathcal{S} = \{(\mathbf{x}_i, y_i)\}_{i \in [n]}$, we solve a binary classification task by minimizing the following *mean squared loss*,

$$L(\mathbf{W}) = \frac{1}{n} \sum_{i \in [n]} \ell(f(\mathbf{x}_i; \mathbf{W}), y_i) = \frac{1}{2n} \sum_{i \in [n]} \left(f(\mathbf{x}_i; \mathbf{W}) - y_i\right)^2, \quad (2)$$

where $\ell(f(\mathbf{x}_i; \mathbf{W}), y_i) = (f(\mathbf{x}_i; \mathbf{W}) - y_i)^2/2$ is the loss on a single data point. Inspired by "edge of stability" (Cohen et al., 2020), adopting mean squared error is believed to make it easier to identify the effects of large learning rates. Besides, mean squared loss has also been demonstrated to be comparable or even better than cross-entropy loss in many classification tasks (Hui, 2020).

We optimize the loss function (2) via *multi-pass stochastic gradient descent* (SGD), initializing from some Gaussian weights, where each entry of $\mathbf{W}_{+1}^{(0)}$ and $\mathbf{W}_{-1}^{(0)}$ is sampled from $N(0, \sigma_0^2)$. The SGD goes for several epochs. In each epoch, we use each data $(\mathbf{x}_i, y_i)$ for exactly once, in the exact order of $(\mathbf{x}_1, y_1) \to (\mathbf{x}_2, y_2) \to \cdots \to (\mathbf{x}_n, y_n)$[1]. Thus, the weights of the CNN are updated obeying the following rule,

$$
\begin{aligned}
\mathbf{w}_{j,r}^{(t+1)} &= \mathbf{w}_{j,r}^{(t)} - \eta \cdot \nabla_{\mathbf{w}_{j,r}} \ell(f(\mathbf{W}^{(t)}, \mathbf{x}_{i_t}), y_{i_t}) \\
&= \mathbf{w}_{j,r}^{(t)} - \frac{j\eta}{m} \cdot \left( f(\mathbf{W}^{(t)}, \mathbf{x}_{i_t}) - y_{i_t} \right) \cdot \sum_{p=1}^{3} \sigma'(\langle \mathbf{w}_{j,r}^{(t)}, \mathbf{x}_{i_t}^{(p)} \rangle) \cdot \mathbf{x}_{i_t}^{(p)},
\end{aligned} \tag{3}
$$

for each $j \in \{\pm 1\}$ and $r \in [m]$, where $i_t = (t+1) \mod n$ and $\eta > 0$ is the learning rate.

**Generalization via signal (feature) learning.** Our goal is to study the generalization property of the CNN trained by SGD (3). Given a new testing data point $(\mathbf{x}^\diamond, y^\diamond)$ sampled from the data generation model, we measure the generalization of the CNN by the correctness of the classification,

$$
\mathbb{E}[\mathbf{1}\{y^\diamond \cdot f(\mathbf{x}^\diamond; \mathbf{W}_{\mathsf{sgd}}) > 0\}] = \mathbb{P}(y^\diamond \cdot f(\mathbf{x}^\diamond; \mathbf{W}_{\mathsf{sgd}}) > 0),
$$

where $\mathbf{W}_{\mathsf{sgd}}$ denotes the weights trained by SGD, Zhang et al. (2021).

We investigate the generalization property via looking through the process of signal (feature) learning. Specifically, by the SGD updates (3), the weights $\mathbf{w}_{j,r}^{(t)}$ of the CNN is a linear combination of the initialization $\mathbf{w}_{j,r}^{(0)}$, the strong signal $j \cdot \mathbf{u}$, the weak signal $y \cdot \mathbf{v}$, and the noise vectors $\boldsymbol{\xi}_i, \widetilde{\boldsymbol{\xi}}_i$. This motivates us to consider the following representation of the weights, for $j \in \{\pm 1\}$, $r \in [m]$,

$$
\mathbf{w}_{j,r}^{(t)} \approx \mathbf{w}_{j,r}^{(0)} + \frac{\langle \mathbf{w}_{j,r}^{(t)}, j\mathbf{u} \rangle}{\|\mathbf{u}\|_2^2} \cdot j\mathbf{u} + \frac{\langle \mathbf{w}_{j,r}^{(t)}, j\mathbf{v} \rangle}{\|\mathbf{v}\|_2^2} \cdot j\mathbf{v} + \text{noise parts.}
$$

The relative scales of these combination coefficients actually imply how the weights learn the strong signal $\mathbf{u}$, the weak signal $\mathbf{v}$, or memorizing the noise which determines how the CNN can generalize.

As is shown by Cao et al. (2022), the CNN tends to fit the training dataset using patches with higher strength when trained by small learning rate gradient descents. Therefore, in such a training regime, the CNN tends to fit the training data using the strong signal $y \cdot \mathbf{u}$, making less progress in learning the weak signal $y \cdot \mathbf{v}$, thus resulting in misclassification when the testing data lacks the strong signal component. On the contrary, our paper investigates the large learning rate regime, and suggests that the *oscillation* of SGD is beneficial for learning the weak signal, giving better generalization results.

Thus, our main focus in the sequel would be studying the dynamics of the inner products $\langle \mathbf{w}_{j,r}^{(t)}, j\mathbf{u} \rangle$, $\langle \mathbf{w}_{j,r}^{(t)}, j\mathbf{v} \rangle$, and $\langle \mathbf{w}_{j,r}^{(t)}, \boldsymbol{\xi} \rangle$. We will show that under large learning rate SGD training $\langle \mathbf{w}_{j,r}^{(t)}, j\mathbf{v} \rangle$ can be effectively learned to a relatively large scale compared to its initialization. This is further provably useful for generalizing to all new data points.

## 3 UNDERSTAND THE OSCILLATION: SINGLE TRAINING DATA CASE

Before giving our main theory on large learning rate SGD training, let's first study a simplified setup where we consider only a *single* training data point consisting only of a weak signal patch $y \cdot \mathbf{v}$ and a strong signal patch $y \cdot \mathbf{u}$, without any noise patch. Such a setting helps to illustrate the key insights behind our main theory regarding the understanding of oscillation. Without loss of generality, we denote the single training data as $(\mathbf{x}, y)$ with $\mathbf{x} = (y \cdot \mathbf{v}, y \cdot \mathbf{u})$, and we can also simplify the CNN expression (1) and the SGD updates (3) to

$$
f(\mathbf{x}; \mathbf{W}) = \sum_{j \in \{\pm 1\}} j F_j(\mathbf{x}; \mathbf{W}_j), \text{ with } F_j(\mathbf{x}; \mathbf{W}_j) = \frac{1}{m} \sum_{r \in [m]} \sigma(\langle \mathbf{w}_{j,r}, y\mathbf{u} \rangle) + \sigma(\langle \mathbf{w}_{j,r}, y\mathbf{v} \rangle), \tag{4}
$$

$$
\mathbf{w}_{j,r}^{(t+1)} = \mathbf{w}_{j,r}^{(t)} - \frac{\eta j}{m} \cdot \left( f(\mathbf{x}; \mathbf{W}^{(t)}) - y \right) \cdot \left( \sigma'(\langle \mathbf{w}_{j,r}^{(t)}, y\mathbf{u} \rangle) \cdot y\mathbf{u} + \sigma'(\langle \mathbf{w}_{j,r}^{(t)}, y\mathbf{v} \rangle) \cdot y\mathbf{v} \right). \tag{5}
$$

---

[1]We consider the same order for all epoch for the simplicity of analysis. Our analysis can be easily extended to multi-pass SGD with shuffling.

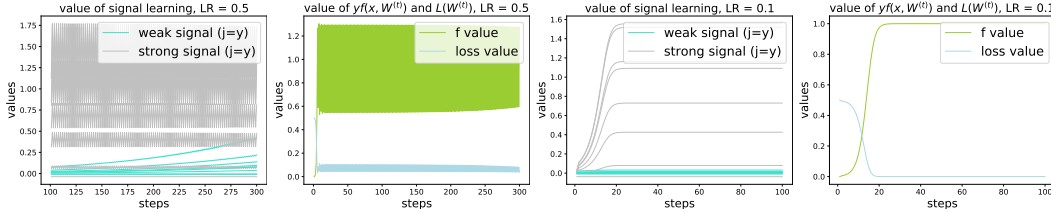

Figure 2: The progress of signal learning and the values of $yf(\mathbf{x}; \mathbf{W}^{(t)})$ and $L(\mathbf{W}^{(t)})$ under different learning rate $\eta$. The CNN in the first two figures is trained by SGD with $\eta = 0.5$ (large LR), while the CNN in the last two figures is trained by SGD with $\eta = 0.1$ (small LR). For signal learning (first and third figures), the gray lines depict the strong signal learning $\langle \mathbf{w}_{y,r}^{(t)}, y\mathbf{u} \rangle$ by all neurons $r \in [m]$, and the light blue lines depict the weak signal learning $\langle \mathbf{w}_{y,r}^{(t)}, y\mathbf{v} \rangle$ by all neurons $r \in [m]$. As we can see, with large LR, the value of CNN oscillates around $y$, and $\sum_t (1 - yf(\mathbf{x}; \mathbf{W}^{(t)}))$ is going to increase, which, as our theory indicates, incentivizes $\langle \mathbf{w}_{y,r}^{(t)}, y\mathbf{v} \rangle$ to increase. In contrast, with small LR, $\langle \mathbf{w}_{y,r}^{(t)}, y\mathbf{v} \rangle$ would stay at the same scale as its initialization.

In such a simplified setup, we aim to explain that, when SGD training belongs to certain *oscillation* regime, which typically occurs under large learning rate $\eta$, the CNN is guaranteed to make progress in learning the weak signal $y \cdot \mathbf{v}$. Here by oscillation, we mean that the values of the CNN $f(\mathbf{x}; \mathbf{W}^{(t)})$ keep oscillating around the label $y$ during training. This phenomenon greatly contrasts with known results for feature learning when gradient descent training converges smoothly under relatively small learning rates (Allen-Zhu and Li, 2022; Cao et al., 2022).

**Review: small learning rate training regime.** Firstly, we make a review of what may happen when using SGD with a small learning rate $\eta$. The following proposition proves that in this case the CNN can **not** make much progress in learning the weak signal $y \cdot \mathbf{v}$.

**Proposition 1** (Small learning rate training: single training data (informal)). *Under mild conditions on $(d, m, \sigma_0, \|\mathbf{u}\|_2, \|\mathbf{v}\|_2)$, if we choose learning rate $\eta \le m/(6\|\mathbf{u}\|_2^2)$ small enough, then with high probability, the training loss can smoothly converge with*

$$\max_{j \in \{\pm 1\}, r \in [m]} \left\{ |\langle \mathbf{w}_{j,r}^{(t)}, j\mathbf{v} \rangle| \right\} \le \widetilde{\mathcal{O}}(\sigma_0 \|\mathbf{v}\|_2).$$

Please refer to Appendix D for more details about the proposition. It shows that in the small learning rate training regime, the CNN only learns the weak signal $y \cdot \mathbf{v}$ to the same scale as its initialization. CNN trained in this manner may fail to generalize to testing data without strong features (substituted by a noise patch $\boldsymbol{\xi}$), because it would make predictions relying mainly on the random noise. On the contrary, in the following we intuitively explain that under certain large learning rate training regime, the CNN can learn the weak signal up $y \cdot \mathbf{v}$ to a constant level higher than its initialization. Such a phenomenon is depicted in Figure 2 on an 8-neuron CNN trained by SGD with $\eta = 0.1$.

**Theoretical motivations: large learning rate regime and oscillation.** When using a large enough learning rate $\eta$ that exceeds the twice inversed smoothness, the weights of the CNN would keep oscillating, which makes the value of $f(\mathbf{x}; \mathbf{W}^{(t)})$ fluctuate around $y$. The key finding towards our theory is that the fluctuations of $f(\mathbf{x}; \mathbf{W}^{(t)})$ around $y$ would *not* cancel with each other. Instead, the oscillation accumulates over time, which serves as the engine driving the learning of the weak signal $y \cdot \mathbf{v}$. In the sequel, we explain why the cancellation does not happen.

The core idea is that, with a reasonably large learning rate, the CNN weights will be quickly enlarged from the learning of strong feature $\mathbf{u}$ and then keep oscillating, but still stay well bounded. As a result of the SGD updates (5), the summation of the gradient terms is also well bounded. More specifically, let's look carefully into the dynamics of learning the strong signal $y \cdot \mathbf{u}$. For some time steps $t_0, t_1$ and certain neuron $r \in [m]$, it holds from (5) that

$$\mathcal{O}(1) = \left| \langle \mathbf{w}_{y,r}^{(t_1+1)}, y\mathbf{u} \rangle - \langle \mathbf{w}_{y,r}^{(t_0)}, y\mathbf{u} \rangle \right| \approx \Theta\left( \sum_{s=t_0}^{t_1} \left(1 - yf(\mathbf{x}; \mathbf{W}^{(s)})\right) \cdot \langle \mathbf{w}_{y,r}^{(s)}, y\mathbf{u} \rangle \right). \quad (6)$$

Now we split the summation on the right hand side of (6) into two parts: one part is $\mathcal{S}^+$ containing $s$ such that $yf(\mathbf{x}; \mathbf{W}^{(s)}) > 1$ and the other part is $\mathcal{S}^-$ containing $s$ such that $yf(\mathbf{x}; \mathbf{W}^{(s)}) < 1$. It turns out that if the weak signal component of the CNN is relatively small compared with the strong signal component, the whole behavior will be dominated by the dynamics of the strong signal component. In other words, when $yf(\mathbf{x}; \mathbf{W}^{(s)}) > 1$, the inner products $\langle \mathbf{w}_{y,r}^{(s)}, y\mathbf{u} \rangle$ would also take a relatively

large value. Conversely, when $yf(\mathbf{x}; \mathbf{W}^{(s)}) < 1$, the inner products $\langle \mathbf{w}_{y,r}^{(s)}, y\mathbf{u} \rangle$ would also take a relatively small value. Consequently, in view of (6), we can see that the total increase of $\langle \mathbf{w}_{y,r}^{(s)}, y\mathbf{u} \rangle$ and decrease of $\langle \mathbf{w}_{y,r}^{(s)}, y\mathbf{u} \rangle$ during the oscillation period are approximately balanced, i.e.,

$$\sum_{s \in \mathcal{S}^+} \underbrace{\left( yf(\mathbf{x}; \mathbf{W}^{(s)}) - 1 \right)}_{yf(\mathbf{x};\mathbf{W}^{(s)})>1} \cdot \underbrace{\langle \mathbf{w}_{y,r}^{(s)}, y\mathbf{u} \rangle}_{\text{relatively } \textcolor{red}{\text{large}}} \approx \sum_{s \in \mathcal{S}^-} \underbrace{\left( 1 - yf(\mathbf{x}; \mathbf{W}^{(s)}) \right)}_{yf(\mathbf{x};\mathbf{W}^{(s)})<1} \cdot \underbrace{\langle \mathbf{w}_{y,r}^{(s)}, y\mathbf{u} \rangle}_{\text{relatively } \textcolor{blue}{\text{small}}}.$$

Consequently, the summation of $1 - yf(\mathbf{x}; \mathbf{W}^{(s)})$ over $s \in \mathcal{S}^-$ would take a larger value than the summation of $yf(\mathbf{x}; \mathbf{W}^{(s)}) - 1$ over $s \in \mathcal{S}^+$. This means that the whole summation

$$\sum_{s \in \mathcal{S}^+ \cup \mathcal{S}^-} \left( yf(\mathbf{x}; \mathbf{W}^{(s)}) - 1 \right) = \sum_{s \in \mathcal{S}^-} \left( 1 - yf(\mathbf{x}; \mathbf{W}^{(s)}) \right) - \sum_{s \in \mathcal{S}^+} \left( yf(\mathbf{x}; \mathbf{W}^{(s)} - 1) \right) \gtrsim 0.$$

That is, the oscillation of $f(\mathbf{x}; \mathbf{W}^{(s)})$ around the label $y$ over time does *not* tend to cancel with each other. Instead, the summation of the fluctuations would have a determined sign. Furthermore, if the CNN values are bounded away from the label by a uniform constant $\delta$ (i.e., the magnitude of the oscillation), we can further prove that

$$\sum_{s \in \mathcal{S}^+ \cup \mathcal{S}^-} \left( yf(\mathbf{x}; \mathbf{W}^{(s)}) - 1 \right) \gtrsim \Omega\left( \delta \cdot \max\{|\mathcal{S}^+|, |\mathcal{S}^-|\} \right) \gtrsim \Omega\left( \delta \cdot (t_1 - t_0) \right). \tag{7}$$

This is the key motivation behind our theory for studying the oscillating SGD. With (7) in hand, we can further show that once the weak signal component of the CNN is still small, which means that the weak signal hasn't been well learned, the linear accumulation of the oscillation would incentivize the learning of the weak signal $\langle \mathbf{w}_{y,r}^{(t)}, y\mathbf{v} \rangle$ by a careful analysis of its updates (5).

**Outcome of oscillation.** Based on previous discussions, we can arrive at our result for the simplified setup of this explanatory section: the oscillating SGD can indeed make progress in learning the weak signal $y \cdot \mathbf{v}$, which helps the CNN to generalize to new data points which possibly lacks strong signal $y \cdot \mathbf{u}$. This is summarized in the following (informal) theorem and corollary.

**Theorem 2** (Large LR training, single data case (informal)). *Under mild conditions on dimension $d$, width $m$, initialization $\sigma_0$, and learning rate $\eta > m/(4\|\mathbf{u}\|_2^2)$, if the SGD training (5) oscillates in the sense that $|yf(\mathbf{x}; \mathbf{W}^{(t)}) - 1| \geq \delta$ for some constant $\delta > 0$ and each $t \geq 0$, then with high probability there exists a $t^\star \leq T_{\max}$ with $T_{\max} \in \text{poly}(d, m, \eta^{-1}, \delta^{-1}, \|\mathbf{u}\|_2^{-1}, \|\mathbf{v}\|_2^{-1})$ such that*

$$\frac{1}{m} \sum_{r \in [m]} \sigma(\langle \mathbf{w}_{y,r}^{(t)}, y\mathbf{v} \rangle) \geq \delta, \quad \forall t \geq t^\star.$$

Please refer to Appendix C for formal and detailed statement of Theorem 2 and its proofs. Theorem 2 shows that via oscillating SGD training, the CNN learns the weak signal $y \cdot \mathbf{v}$ up to a constant scale of $\delta$, which is typically much larger than the scale of its initialization, since

$$\frac{1}{m} \sum_{r \in [m]} \sigma(\langle \mathbf{w}_{y,r}^{(0)}, y\mathbf{v} \rangle) \lesssim \widetilde{\mathcal{O}}(\sigma_0^2 \cdot \|\mathbf{v}\|_2^2) \ll \delta \leq \frac{1}{m} \sum_{r \in [m]} \sigma(\langle \mathbf{w}_{y,r}^{(t^\star)}, y\mathbf{v} \rangle),$$

whenever the initializations of the CNN is small. We remark that here we mainly consider neurons with $j = y$ and testing data with label $y$ since the CNN is trained only on a single data with label $y$.

**Implications to the simple signal-noise model.** Our findings could also be adapted to explain the setting with data model considered by Cao et al. (2022). In that case, one data point $\mathbf{x} = (y \cdot \boldsymbol{\mu}, \boldsymbol{\xi})$ consists of a single signal patch $y \cdot \boldsymbol{\mu}$ and a single Gaussian noise patch $\boldsymbol{\xi}$. Therefore a trained neural network can generalize to new data points only when it learns the common signal vector $\boldsymbol{\mu}$.

As is shown by Cao et al. (2022), under small learning rate training regime, if the data model has a low *signal-to-noise ratio* (SNR), that is, the strength of the noise patch is relatively stronger than the the strength of the signal patch, then overfitting the training data would result in poor generalization (harmful overfitting). That is because the neural network would memorize the noise patch quickly so as to fit the data, and consequently the signal patch is not well learned. In contrast, we can show that under the oscillating SGD training regime, the signal can also be well learned even with a low SNR. The mechanism behind this is still that the oscillation during training would accumulate and incentivize the neural network towards signal learning.

## 4 MAIN THEORY

In this section, we present our main theory on benign oscillation of SGD training with large learning rates for the setup introduced in Section 2. We will first introduce the key conditions and assumptions required by our theory in Section 4.1. Then we present our theoretical results in Section 4.2. Finally in Section 4.2, we also compare large learning rate oscillating training to small learning rate training.

### 4.1 KEY CONDITIONS AND ASSUMPTIONS

Before presenting our theoretical results, we first outline the key conditions and assumptions needed on the model and the training dynamics. Firstly, our results are based upon the following conditions on the initialization scale $\sigma_0$, dimension $d$, number of data $n$, and neural network width $m$.

**Assumption 3** (Conditions on hyperparameters). *Suppose that the following conditions hold: (i) the CNN weight initialization scale $\sigma_0 = \widetilde{\Theta}(\max\{\|\mathbf{u}\|_2, \|\mathbf{v}\|_2, \sigma_p\sqrt{d}\}^{-1} \cdot d^{-1/2})$; (ii) the dimension $d = \Omega(n^2, \mathrm{polylog}(m))$; (iii) the signal strength: $\|\mathbf{v}\|_2 \leq 0.1\|\mathbf{u}\|_2$, $\|\mathbf{u}\|_2^{-2} + \|\mathbf{v}\|_2^{-2} \leq n(\sigma_p^2 d)^{-1}$. (iv) the learning rate $m/(4\|\mathbf{u}\|_2^2) \leq \eta \leq 2m/(5\|\mathbf{u}\|_2^2)$. (v) the weak data fraction $\rho \leq c$ for some small constant c.*

We explain the conditions in Assumption 3 one by one. The conditions on the initialization scale $\sigma_0$ and the learning rate $\eta$ are to ensure that the whole training process is well bounded while oscillates (rather than converging smoothly). The condition on the dimension $d$ puts us in the regime of high dimensions for which independent Gaussian noise has small correlations. Finally, the conditions on the signal strength separate the strong signal from the weak signal by $\ell_2$-norm. Also, we ensure that the data are not too noisy by restricting the variance of the Gaussian noise.

The next assumption is on the training process, which requires that the SGD oscillates. For simplicity, we denote the index of weak training data points lacking the strong feature patch as $\mathcal{W}$.

**Assumption 4** (Oscillating SGD). *We assume that there exists a constant $\delta \in (0.2, 0.8)$, such that $|y_{i_t} f(\mathbf{x}_{i_t}; \mathbf{W}^{(t)}) - 1| \geq \delta$ holds for any $t \geq 0$ such that $i_t \notin \mathcal{W}$.*

Through Assumption 4, we require that the value of $yf(\mathbf{x}; \mathbf{W}^{(t)})$ on data points with strong features oscillates around the desired value, 1, by a scale of $\delta \in (0.2, 0.8)$, i.e., the magnitude of the oscillation is at least $\delta$. Here the range for $\delta$ is only for technical considerations to simplify the theoretical analyses.

It is notable that Assumption 4 implicitly requires that the learning rate $\eta$ should be scaled properly. A large $\eta$ forces the training trajectories to escape from the regular region, while a small $\eta$ shall result in smooth convergence. In both cases the phenomenon described in Assumption 4 does not happen. We also remind readers that the $\eta$ condition in Assumption 3 is only sufficient for the regularities such as boundedness and sign stability. Readers can refer to Appendix C.7 for a discussion of the necessary conditions of Assumption 4.

We remark that in general the dynamics of the training process could be quite subtle when oscillation happens, and there exist other more complicated patterns of oscillations if one deliberately chooses a specific learning rate $\eta$. Our work focuses on a relatively simple but common pattern of oscillation. It turns out that under the oscillation pattern in Assumption 4, we can show the benefits of oscillation on the generalization properties of the CNN. Actually, we can also extend our theoretical analysis to a weakened version of Assumption 4 that the time average of $|y_{i_t} f(\mathbf{x}_{i_t}; \mathbf{W}^{(t)}) - 1|$ is larger than $\delta$.

Finally, we remark that we only assume the oscillation on strong data, since intuitively on weak data the CNN fits the label via the weak features and the noise (both have smaller strength than strong features) and may converge slower and more smoothly. *See Section 4.3 for experimental evidence.*

### 4.2 MAIN THEORETICAL RESULTS

Our main results are that, under previous conditions and assumptions on the hyperparameters and the training dynamics, the CNN can make enough progress in learning the weak signal $\mathbf{v}$ thanks to the oscillation happening during training. We refer to Appendix E for a detailed proof of the results.

**Theorem 5** (Weak signal learning: oscillating training with large learning rate). *Under Assumptions 3 and 4, w.p. at least $1 - 1/\mathrm{poly}(d)$, there exists $t^\star \leq \mathrm{poly}(d, m, n, \delta^{-1}, \eta^{-1}\sigma_p^{-1}, \|\mathbf{u}\|_2^{-1}, \|\mathbf{v}\|_2^{-1})$*

Figure 3: The dynamics of signal learning under a large learning rate $\eta_{\text{large}} = 1.2$ and a small learning rate $\eta_{\text{small}} = 0.1$. The values of signal learning are obtained by characterizing the inner products $\langle \mathbf{w}_{j,r}^{(t)}, \mathbf{u} \rangle$ and $\langle \mathbf{w}_{j,r}^{(t)}, \mathbf{v} \rangle$. It can be seen that when using the large LR, strong signal learning as well as the NN outputs will oscillate, during which weak signal will be gradually learned. When using the small LR, strong signal learning will converge quickly, and the weak signal learning will stay at the same scale as its initialization.

*such that*

$$\max_{j \in \{\pm 1\}} \left\{ \frac{1}{m} \sum_{r \in [m]} \sigma(\langle \mathbf{w}_{j,r}^{(t)}, j\mathbf{v} \rangle) - \frac{1}{m} \sum_{r \in [m]} \sigma(\langle \mathbf{w}_{-j,r}^{(t)}, j\mathbf{v} \rangle) \right\} \geq \frac{\delta}{4}.$$

In contrast, under the small learning rate regime, the CNN would not learn the weak features, which is the following proposition with proofs in Appendix F.

**Proposition 6** (Small LR training). *Under Assumption 35 on $(d, m, n, \sigma_0, \|\mathbf{u}\|_2, \|\mathbf{v}\|_2, \sigma_p)$, if we choose learning rate $\eta \leq m/(6\|\mathbf{u}\|_2^2)$ small enough, then with high probability, the training loss can smoothly converge with*

$$\max_{j \in \{\pm 1\}, r \in [m]} \left\{ |\langle \mathbf{w}_{j,r}^{(t)}, j\mathbf{v} \rangle| \right\} \leq \widetilde{\mathcal{O}}(\sigma_0 \|\mathbf{v}\|_2).$$

**Division of generalization.** Suppose we are given a new testing data point $(\mathbf{x}^\diamond, y^\diamond)$ with an input $\mathbf{x}^\diamond = (y^\diamond \mathbf{v}, \boldsymbol{\xi}^\diamond, \widetilde{\boldsymbol{\xi}}^\diamond)$ only consisting of the weak signal $\mathbf{v}$. Then a reliable prediction can only count on utilizing the weak signal $\mathbf{v}$. In the regime specified by Theorem 5, there holds $y^\diamond \cdot f(\mathbf{x}^\diamond; \mathbf{W}^{(t)}) \geq \delta/4 - o(1) > 0$ corresponding to correct prediction almost certainly. In contrast, when applying the small learning rate, as specified by Proposition 6, the trained NN fails to take advantage of the weak signal $\mathbf{v}$ from the data $\mathbf{x}^\diamond$. Therefore, it will be likely to make the prediction based on a random guess (the randomness stems from the random initialization and the noise patches $\boldsymbol{\xi}^\diamond, \widetilde{\boldsymbol{\xi}}^\diamond$). Consequently, note that the weak data takes up $\rho$ fraction of the dataset (see our data model in Section 2), SGD with large LR will achieve a $\Theta(\rho)$ higher test accuracy than SGD with small LR, demonstrating the benefit of oscillation and large learning training in terms of the generalization ability.

### 4.3 NUMERICAL EXPERIMENTS

In this part, we conduct numerical experiments to demonstrate our findings on "*benign oscillation*". We follow the same data generation model and optimization algorithm as we described as Section 2. Specifically, we consider a dataset with $n = 16$ and $|\mathcal{W}| = 2$, that is, $\rho \approx 0.125$. The dimension is $d = 64$, and the number of neurons for each direction $j$ is $m = 8$. We generate the data with strong signal $\|\mathbf{u}\|_2 = 2$, weak signal $\|\mathbf{v}\|_2 = 0.4$, and noise $\|\boldsymbol{\xi}\|_2 \approx \sigma_p d^{1/2} = 0.8$.

**Weak signal learning.** We run the SGD to train the CNN with two different scale of learning rates: a large learning rate $\eta_{\text{large}} = 1.2$, a small learning rate $\eta_{\text{small}} = 0.1$. We plot the dynamics of signal learning for each neuron $r \in [m]$ from these two training regimes in Figure 3. The first two figures plot the large learning rate training, and the last two figures plot the small learning rate training.

As we can see from Figure 3, with large learning rate SGD training, the CNN can effectively learn the weak signal to a scale much larger than the initialization. On the contrary, by small learning rate SGD training, the CNN does not learn the weak signal since it just remains at the same level as the initialization. This demonstrates our main theory in Section 4.

Furthermore, in Figure 3, we plot the values of $y_i f(\mathbf{x}_i; \mathbf{W}^{(t)})$ on certain data points $i \in [n]$ and the value of $L(\mathbf{W}^{(t)})$ for the two training regimes. In specific, we plot the values of $y_i f(\mathbf{x}_i; \mathbf{W}^{(t)})$ on a strong data $i \notin \mathcal{W}$ (randomly sampled) and the value of $y_i f(\mathbf{x}_i; \mathbf{W}^{(t)})$ on the weak data $i \in \mathcal{W}$. As we can see from the large learning rate training case, the value of $f$ on the strong data oscillates while the values of $f$ on the weak data do not. This matches our theoretical assumption in Assumption 4 that the oscillation in $f$ value only happens for strong data. Also, for the small learning rate training

case, the values of $f$ on the weak data converge slower than those on the strong data. This is because on the weak data the CNN mainly uses the noise to fit the target which is of lower strength than the strong signals on strong data. But still, the noise out weights the weak signals and consequently the CNN makes no progress in learning the weak signal if trained smoothly.

**Generalization properties.** Finally, we test the CNN trained by two different learning rates on new testing data generated in the same way as the training data, The testing data size 32 with 4 weak data points. We repeat the testing evaluation over 5 random seeds and take the average. The result is that for the CNN trained by $\eta_{\text{large}}$ the test accuracy is $99.38\%$, and for the CNN trained by $\eta_{\text{small}}$ the test accuracy is $93.75\%$, matching our theoretical insights that large learning rate training benefits NN generalization. For the CNN trained by $\eta_{\text{small}}$, it misclassifies certain weak data points. As we previously discussed, on data without strong signal, the CNN approximately uses a random guess.

## 5 RELATED WORKS

**Large learning rate NN training.** Gradient descent training coped with large learning rates for deep learning is receiving an ever increasing attention for recent years (Cohen et al., 2020; Jastrzebski et al., 2021; Andriushchenko et al., 2023). For GD training, the phenomenon of "*edge of stability*" (Cohen et al., 2020; 2022) showed that the sharpness of the loss Hessian would finally hover just above $2/\eta$ and thus a larger learning rate would prefer a flatter minimum and possibly better generalization, and have received great attention in recent years (Arora et al., 2022; Chen and Bruna, 2022; Damian et al., 2022; Wang et al., 2022; Zhu et al., 2022b). Besides, Li et al. (2019) studied the regularization effect of large learning rates of SGD at initialization which results in better generalization than using a small initial learning rate training. Wu et al. (2021) studied the implicit bias of SGD with a moderate large learning rate for overparametrized linear regression. Wu et al. (2023) then studied the implicit bias of large learning rate GD training in logistic regression. In addition, Andriushchenko et al. (2023) showed that SGD with a large learning rate can help NNs to learn sparse features from data, but did not provide rigorous theoretical justifications. We highlight that our theoretical work on large learning rate SGD builds upon a multi-pass fashion of SGD and a feature-noise data generation model (see Section 2), which is different from previous works (Li et al., 2019; Wu et al., 2021; Andriushchenko et al., 2023) where noise-approximated-SGD is adopted for analysis. Also, we study the behavior of large LR SGD by focusing on the role of *oscillation*, which is also largely different from the prior works.

**Feature learning in deep learning theory.** There has been a long line of research in deep learning theory from the perspective of *feature learning* during training of neural network (Allen-Zhu and Li, 2022; Wen and Li, 2021; Zou et al., 2022; Cao et al., 2022; Chen et al., 2022; Zou et al., 2023; Huang et al., 2023; Yang et al., 2023). The idea is that, by explicitly characterizing the dynamics of feature learning during training, one can figure out how different algorithms and data structures can influence the learning of features by the neural network, further uncovering the properties of interest in deep learning, e.g., ensemble (Allen-Zhu and Li, 2022), adaptive gradients (Zou et al., 2022), the phenomenon of benign overfitting (Cao et al., 2022), data augmentation via mixup (Zou et al., 2023), etc. Specifically, the work of Cao et al. (2022) showed that under small learning rate regimes, training on data with low *signal-to-noise ratio* (SNR) would result in *harmful overfitting*, leading to poor generalization abilities of the neural network. Our work extends this line of research to the less theoretically understood regime of large learning rates by characterizing the feature learning process when oscillation happens during gradient descent and explaining its benefits to generalization.

**Further related works.** Please see Appendix A for more related works.

## 6 CONCLUSIONS

This work theoretically investigated NN training with large learning rates and established a theoretical framework to understand the oscillation phenomenon. We revealed the benefit of oscillation to the NN generalization, which we summarize as the phenomenon of "*benign oscillation*". Our theory demystified the phenomenon based on a feature learning perspective and showed that the oscillation can drive the learning of weak but important patterns from data that are crucial to generalization. Our theory shed light on the understanding of large learning rate NN training and provided useful guidance towards the optimization analysis when smooth convergence is guaranteed.

## ACKNOWLEDGEMENTS

The authors would like to thank the anonymous reviewers and area chair for their helpful comments. This work was done when ML and BW were visiting the Department of Computer Science at the University of Hong Kong. DZ acknowledges the support from NSFC 62306252.

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

CONTENTS

## A FURTHER RELATED WORKS

**Additional related works on large LR NN Training.** Ziyin et al. (2021); Zhu et al. (2022a); Ziyin et al. (2023); Even et al. (2023) also study neural network training with large learning rates. More specifically, the work of Ziyin et al. (2021; 2023) is from the perspective of loss landscape of neural network optimization under large learning rates and different signal-to-noise ratios. In contrast, we explain the better generalization properties given by large learning rate training through the lens of feature learning, which has a different target and provides a different viewpoint. Zhu et al. (2022a) explain the "catapult phase" exhibited by large learning rate NN training on neural quadratic models. Even et al. (2023) investigate the impact of large learning rate on the implicit regularisation of (S)GD over 2-layer diagonal linear networks.

**Diagonal linear neural networks.** There is a line of theoretical works on the training of diagonal linear neural networks (Pesme et al., 2020; 2021; Andriushchenko et al., 2023; Even et al., 2023), where the impact of large learning rate is also investigated (Andriushchenko et al., 2023; Even et al., 2023). As is shown (Andriushchenko et al., 2023; Even et al., 2023), for the diagonal linear neural networks, large learning rate training can lead to sparse features, which is somehow related to our dynamic analysis. More specifically, it turns out that our theoretical analysis not only indicates that the weak signal is properly learned (i.e., for $\langle \mathbf{w}_{y,r}, y\mathbf{v}\rangle > 0$, it increases), but can also be leveraged to prove that all the negative parts (i.e., $\langle \mathbf{w}_{-y,r}, y\mathbf{v}\rangle > 0$ and $\langle \mathbf{w}_{-y,r}, y\mathbf{u}\rangle > 0$) are "unlearned", that is, they are pushed towards zero (we did not explicitly prove it since an upper bound of these negative parts are sufficient for our analysis, see Lemma 16 or Lemma 31). Therefore, once we prove this, we can actually conclude that the model not only learns the weak signal, but also implicitly learns a "sparse" solution to some extent. Also, learning more useful features, which is the outcome of large learning rate SGD, can sortly suppress the noise memorization, since the more noisy model could lead to higher sharpness than the model that largely learns features to fit the training data. This may also connect to the sparsity of the learned model.

## B PRELIMINARY LEMMAS ON CONCENTRATION

In this section, we give finite-sample concentration results to characterize the high-probability concentration properties of the random elements involved in our problem. Throughout this paper, we fix a small constant failure probability $p = 1/\mathrm{poly}(d)$.

**Lemma 7.** *Suppose that $n \geq 8\log(4/p)$, then with probability at least $1 - p$, we have that*

$$|\{i \in [n] : y_i = 1\}| \wedge |\{i \in [n] : y_i = -1\}| \geq \frac{n}{4}.$$

*Proof of Lemma 7.* See Lemma B.1 in Cao et al. (2022) for a proof. □

**Lemma 8.** *Suppose that $n \geq 8 \log(4/p)$, then with probability at least $1 - p$, we have that*

$$|\mathcal{W}| \leq \frac{2\rho}{n}.$$

*Proof of Lemma 8.* This follows from the same proof as Lemma 7. □

**Lemma 9.** *Suppose that $d = \Omega(\log(4n/p))$, then with probability $1 - p$, we have that*

$$\sigma_p^2 d/2 \leq \|\boldsymbol{\xi}\|_2^2 \leq 3\sigma_p^2 d/2, \quad |\langle \boldsymbol{\xi}_i, \boldsymbol{\xi}_{i'} \rangle| \leq 2\sigma_p^2 \cdot \sqrt{d \log(2n/p)}$$

*hold for all $i, i' \in [n]$.*

*Proof of Lemma 9.* See Lemma B.2 in Cao et al. (2022) for a proof. □

**Lemma 10.** *Suppose that $d \geq \Omega(\log(mn/p))$. Then with probability at least $1 - p$, we have that*

$$\sigma_0 \|\mathbf{u}\|_2/2 \leq \max_{j \in \{\pm 1\}, r \in [m]} \langle \mathbf{w}_{j,r}^{(0)}, j\mathbf{u} \rangle \leq \sqrt{2 \log(16m/p)} \cdot \sigma_0 \|\mathbf{u}\|_2,$$

$$\min_{j \in \{\pm 1\}, r \in [m]} \langle \mathbf{w}_{j,r}^{(0)}, j\mathbf{u} \rangle \geq -\sqrt{2 \log(16m/p)} \cdot \sigma_0 \|\mathbf{u}\|_2,$$

$$\sigma_0 \|\mathbf{v}\|_2/2 \leq \max_{j \in \{\pm 1\}, r \in [m]} \langle \mathbf{w}_{j,r}^{(0)}, j\mathbf{v} \rangle \leq \sqrt{2 \log(16m/p)} \cdot \sigma_0 \|\mathbf{v}\|_2,$$

$$\min_{j \in \{\pm 1\}, r \in [m]} \langle \mathbf{w}_{j,r}^{(0)}, j\mathbf{v} \rangle \geq -\sqrt{2 \log(16m/p)} \cdot \sigma_0 \|\mathbf{v}\|_2.$$

$$\sigma_0 \sigma_p \sqrt{d}/4 \leq \max_{r \in [m]} j \cdot \langle \mathbf{w}_{j,r}^{(0)}, \boldsymbol{\xi}_i \rangle \leq 2\sqrt{\log(16mn/p)} \cdot \sigma_0 \sigma_p \sqrt{d}, \quad \forall i \in [n],$$

*Proof of Lemma 10.* See Lemma B.3 in Cao et al. (2022) for a proof. □

## C  PROOFS FOR SINGLE TRAINING DATA CASE (SECTION 3)

In this section, we give a formal statement and a detailed proof for our main results on the single noiseless training data setup, Theorem 2.

We begin with the formal statement on the conditions and assumptions required for Theorem 2. Firstly, we put requirements on the data model, initialization, and learning rates.

**Assumption 11** (Conditions on hyperparameters). *Suppose that the following holds:*

1. *The learning rate $m/(4\|\mathbf{u}\|_2^2) \leq \eta \leq 2m/(5\|\mathbf{u}\|_2^2)$;*

2. *The weight initialization scale $\sigma_0 = \widetilde{\Theta}(\max\{\|\mathbf{u}\|_2, \|\mathbf{v}\|_2\}^{-1} \cdot d^{-1/2})$;*

3. *The signal strength $\|\mathbf{v}\|_2 < 0.01 \cdot \|\mathbf{u}\|_2$;*

4. *The dimension $d$ satisfies $d = \Omega(\text{polylog}(m))$.*

These conditions are the simplified version of conditions of Assumption 3 for the multiple data setup. The first condition on the learning rate guarantees the regularity of the training trajectories, including the boundedness and the sign stability. The second condition on the weight initialization scale makes sure that the CNN is not initialized too large, which is common in practice and also helps regularize the training trajectory. The third condition on the signal strength separates the strong signal from the weak signal by their $\ell_2$-norm. Finally, the last condition on the dimension $d$ puts us in the regime of high dimensions for which independent Gaussian noise has small correlations.

We make some additional remarks on the choice of the weight initialization scale $\sigma_0$. It is to ensure that the scale of initialization is smaller than the oscillation magnitude (at least in the high dimensional region). A direct benefit of this is that by Theorem 2, one can immediately conclude that the weak signal is guaranteed to be learned to the scale much higher than its initialization, improving

over the small learning rate training regime (Proposition 1). Also, small weight initialization is also assumed in the original edge-of stability paper (Cohen et al., 2020), as they need to guarantee "progressive sharpening", where the sharpness at the initialization should be smaller than the sharpness at the oscillation phase. So the whole thing we want is a weight initialization that is not too large so that it is smaller than the scale of oscillation. Any other conditions on initialization would also work if this is guaranteed, e.g., it can be relaxed to $\sigma_0 = \widetilde{\Theta}(\max\{\|\mathbf{u}\|_2, \|\mathbf{v}\|_2\}^{-1} \cdot d^{-\alpha})$ for some $\alpha > 0$.

The next assumption is on the training process, which requires that the SGD oscillates.

**Assumption 12** (Oscillations SGD: single training data case). *We assume that there exists some constant $\delta \in (0.2, 0.8)$, such that $|yf(\mathbf{x}; \mathbf{W}^{(t)}) - 1| \geq \delta$ for any $t \geq 0$, where $(\mathbf{x}, y)$ denotes the single data point, $\mathbf{W}^{(t)}$ denotes the weights found by SGD (5).*

Again, we refer the readers to Section 4.1 for an explanation of Assumption 12. With Assumptions 11 and 12, our formal statement of Theorem 2 is the following theorem.

**Theorem 13** (Restatement of Theorem 2). *Under Assumptions 11 and 12, with probability at least $1 - 1/\text{poly}(d)$, there exists a step $T_{(\mathbf{v})}$ such that for $j = y$ and any $t \geq T_{(\mathbf{v})}$, it holds that*

$$\frac{1}{m} \sum_{r \in [m]} \sigma(\langle \mathbf{w}_{j,r}^{(t)}, j\mathbf{v} \rangle) - \frac{1}{m} \sum_{r \in [m]} \sigma(\langle \mathbf{w}_{-j,r}^{(t)}, j\mathbf{v} \rangle) \geq \frac{\delta}{4},$$

*where $\delta > 0$ is specified in Assumption 12 and $T_{(\mathbf{v})} \leq \widetilde{\Theta}(m \cdot \eta^{-1} \cdot \|\mathbf{v}\|_2^{-2} \cdot \delta^{-1} \cdot \log(m\delta\sigma_0^{-1}\|\mathbf{v}\|_2^{-1}))$.*

*Proof of Theorem 13.* Please refer to Appendix C.3 for a detailed proof. □

The following of this section is organized as following. Appendix C.1 presents important properties of the whole training dynamics and the CNN, which serve as the basis for all the following proofs. Appendix C.2 presents the fundamental step that allows for proving weak signal learning. Based on that, we prove the main theorem in Appendix C.3. Finally, the remaining subsections collect the proof for all the lemmas and technical results involved in Appendix C.

## C.1 BASIC PROPERTIES OF TRAINING DYNAMICS AND THE TWO-LAYER CNN

**Properties of training dynamics.** We first define some neuron subsets. For $j \in \{\pm 1\}$, we define that

$$\mathcal{U}_{j,+}^{(t)} = \left\{ r \in [m] : \langle \mathbf{w}_{y,r}^{(t)}, y\mathbf{u} \rangle > 0 \right\}, \quad \mathcal{U}_{j,-}^{(t)} = \left\{ r \in [m] : \langle \mathbf{w}_{y,r}^{(t)}, y\mathbf{u} \rangle \leq 0 \right\}.$$

and

$$\mathcal{V}_{j,+}^{(t)} = \left\{ r \in [m] : \langle \mathbf{w}_{y,r}^{(t)}, y\mathbf{v} \rangle > 0 \right\}, \quad \mathcal{V}_{j,-}^{(t)} = \left\{ r \in [m] : \langle \mathbf{w}_{y,r}^{(t)}, y\mathbf{v} \rangle \leq 0 \right\}.$$

By the update formula (5), we know that the gradient descent iterates the inner products $\{\langle \mathbf{w}_{j,r}^{(t)}, j\mathbf{u} \rangle\}_{t \geq 0}$ as follows:

$$\langle \mathbf{w}_{y,r}^{(t+1)}, y\mathbf{u} \rangle = \langle \mathbf{w}_{y,r}^{(t)}, y\mathbf{u} \rangle + \frac{\eta\|\mathbf{u}\|_2^2}{m} \cdot \left(1 - yf(\mathbf{x}; \mathbf{W}^{(t)})\right) \cdot \sigma'\left(\langle \mathbf{w}_{y,r}^{(t)}, y\mathbf{u} \rangle\right), \tag{8}$$

$$\langle \mathbf{w}_{-y,r}^{(t+1)}, -y\mathbf{u} \rangle = \langle \mathbf{w}_{-y,r}^{(t)}, -y\mathbf{u} \rangle + \frac{\eta\|\mathbf{u}\|_2^2}{m} \cdot \left(1 - yf(\mathbf{x}; \mathbf{W}^{(t)})\right) \cdot \sigma'\left(-\langle \mathbf{w}_{-y,r}^{(t)}, -y\mathbf{u} \rangle\right). \tag{9}$$

Analogously, we have that

$$\langle \mathbf{w}_{y,r}^{(t+1)}, y\mathbf{v} \rangle = \langle \mathbf{w}_{y,r}^{(t)}, y\mathbf{v} \rangle + \frac{\eta\|\mathbf{v}\|_2^2}{m} \cdot \left(1 - yf(\mathbf{x}; \mathbf{W}^{(t)})\right) \cdot \sigma'\left(\langle \mathbf{w}_{y,r}^{(t)}, y\mathbf{v} \rangle\right), \tag{10}$$

$$\langle \mathbf{w}_{-y,r}^{(t+1)}, -y\mathbf{v} \rangle = \langle \mathbf{w}_{-y,r}^{(t)}, -y\mathbf{v} \rangle + \frac{\eta\|\mathbf{v}\|_2^2}{m} \cdot \left(1 - yf(\mathbf{x}; \mathbf{W}^{(t)})\right) \cdot \sigma'\left(-\langle \mathbf{w}_{-y,r}^{(t)}, -y\mathbf{v} \rangle\right).$$

Then we invite readers to some facts that helps to understand the behavior of the inner products during the training processes. First, we note that by (8), for any $r \in \mathcal{U}_{y,-}^{(0)}$,

$$\left\{ \langle \mathbf{w}_{y,r}^{(t)}, y\mathbf{u} \rangle \right\}_{t \geq 0}$$

stays fixed at its initialization, thus automatically keeps a fixed sign. The same phenomenon that the inner products are fixed can be verified on following neuron sets,

$$\left\{\langle \mathbf{w}_{-y,r}^{(t)}, -y\mathbf{u}\rangle\right\}_{t\geq 0}, \ \forall r \in \mathcal{U}_{-y,+}^{(0)}; \qquad \left\{\langle \mathbf{w}_{y,r}^{(t)}, y\mathbf{v}\rangle\right\}_{t\geq 0}, \ \forall r \in \mathcal{V}_{y,-}^{(0)}; \qquad \left\{\langle \mathbf{w}_{-y,r}^{(t)}, -y\mathbf{v}\rangle\right\}_{t\geq 0}, \ \forall r \in \mathcal{V}_{-y,+}^{(0)}.$$

The benefit of this phenomenon is when looking at the prediction $f(\cdot; \mathbf{W}^{(t)})$ on the single data with label $y$,

$$f(\mathbf{x}; \mathbf{W}^{(t)}) = \frac{y}{m} \sum_{r\in[m]} \sigma\big(\langle \mathbf{w}_{y,r}^{(t)}, y\mathbf{u}\rangle\big) + \sigma\big(\langle \mathbf{w}_{y,r}^{(t)}, y\mathbf{v}\rangle\big) - \sigma\big(\langle \mathbf{w}_{-y,r}^{(t)}, y\mathbf{u}\rangle\big) - \sigma\big(\langle \mathbf{w}_{-y,r}^{(t)}, -y\mathbf{v}\rangle\big).$$

(11)

By splitting the summation over $r \in [m]$ according to our defined neuron sets, we can simplify (11) as

$$f(\mathbf{x}; \mathbf{W}^{(t)}) = \frac{y}{m} \sum_{r\in\mathcal{U}_{y,+}^{(t)}} \sigma\big(\langle \mathbf{w}_{y,r}^{(t)}, y\mathbf{u}\rangle\big) + \frac{y}{m} \sum_{r\in\mathcal{V}_{y,+}^{(t)}} \sigma\big(\langle \mathbf{w}_{y,r}^{(t)}, y\mathbf{v}\rangle\big)$$
$$- \frac{y}{m} \sum_{r\in\mathcal{U}_{-y,-}^{(t)}} \sigma\big(-\langle \mathbf{w}_{-y,r}^{(t)}, -y\mathbf{u}\rangle\big) - \frac{y}{m} \sum_{r\in\mathcal{V}_{-y,-}^{(t)}} \sigma\big(-\langle \mathbf{w}_{-y,r}^{(t)}, -y\mathbf{v}\rangle\big), \quad (12)$$

which only involves $\mathbf{w}_{y,r}^{(t)}$ with $r \in \mathcal{U}_{y,+}^{(t)} \cup \mathcal{V}_{y,+}^{(t)}$ and $\mathbf{w}_{-y,r}^{(t)}$ with $r \in \mathcal{U}_{-y,-}^{(t)} \cup \mathcal{V}_{-y,-}^{(t)}$. Combined with previous observations, we only need to track the training dynamics of the neurons appearing in (12).

Thanks to these facts and the signal strength regime in Assumption 12, we can then retreat to tracking the movements of

$$\left\{\langle \mathbf{w}_{y,r}^{(t)}, y\mathbf{u}\rangle\right\}_{t\geq 0}, \ \forall r \in \mathcal{U}_{y,+}^{(0)} \quad \text{and} \quad \left\{\langle \mathbf{w}_{-y,r}^{(t)}, -y\mathbf{u}\rangle\right\}_{t\geq 0}, \ \forall r \in \mathcal{U}_{-y,-}^{(0)}$$

whenever the weak signal part is not yet effectively learned and the strong signal part still dominates. Ideally, only the inner products $\langle \mathbf{w}_{y,r}^{(t)}, y\mathbf{u}\rangle, r \in \mathcal{U}_{y,+}^{(t)}$ take the lead.

Then a natural and crucial question is the boundedness of these inner products, which turns out to be the cornerstone for the subsequent analysis. A straightforward but helpful lemma indicates that the inner products that are initialized to be the maximal (resp. minimal) among all inner products continue to be the maximal (resp. minimal) throughout the training process. To put formally, we have the following.

**Lemma 14** (Maximum and minimum neurons). *Suppose that the signs of all the related inner products do not change throughout $[t_1, t_2]$, i.e., $\mathcal{U}_{y,+}^{(t)} = \mathcal{U}_{y,+}^{(t_1)}$ and $\mathcal{U}_{-y,-}^{(t)} = \mathcal{U}_{-y,-}^{(t_1)}$ for all $t \in [t_1, t_2]$. Then we have that*

$$\operatorname*{argmax}_{r\in[m]}\langle \mathbf{w}_{y,r}^{(t_1)}, y\mathbf{u}\rangle = \operatorname*{argmax}_{r\in[m]}\langle \mathbf{w}_{y,r}^{(t)}, y\mathbf{u}\rangle,$$
$$\operatorname*{argmin}_{r\in[m]}\langle \mathbf{w}_{-y,r}^{(t_1)}, -y\mathbf{u}\rangle = \operatorname*{argmin}_{r\in[m]}\langle \mathbf{w}_{-y,r}^{(t)}, -y\mathbf{u}\rangle$$

*hold for all $t \in [t_1, t_2]$.*

*Proof of Lemma 14.* See Appendix C.4.1 for a detailed proof. $\square$

A direct profit of this lemma is that when the signs keep invariant, it suffices to track two specific indices in $\mathcal{U}_{y,+}^{(t)}$ and $\mathcal{U}_{-y,-}^{(t)}$, the maximum and the minimum, to analyze upper and lower bounds for all neurons.

**Single neuron behaves similarly to CNN.** The proof of boundedness utilizes another property of two-layer CNN defined in Equation (1) that exhibits the connections between the behavior of inner products $\langle \mathbf{w}_{y,r}, y\mathbf{u} \rangle$ and the outcome of the model $f(\cdot; \mathbf{W})$. We state it as follows.

**Lemma 15** (Single neuron imitates entire CNN). *Define the major part of $y \cdot f(\mathbf{x}; \mathbf{W})$ as*

$$g(\mathbf{x}, y; \mathbf{W}) = \frac{1}{m} \sum_{r \in [m]} \sigma\big(\langle \mathbf{w}_{y,r}, y\mathbf{u} \rangle\big).$$

*Suppose that there is $t_1 < t_2$ such that $\mathcal{U}_{y,+}^{(t)} = \mathcal{U}_{y,+}^{(t_1)}$ for all $t \in [t_1, t_2]$. Then, for any $c > 0$, $g(\mathbf{x}, y; \mathbf{W}^{(t)}) \geq c$ implies that for all $t \in [t_1, t_2]$,*

$$\max_{r \in [m]} \langle \mathbf{w}_{y,r}^{(t)}, y\mathbf{u} \rangle \geq (\beta_{\mathbf{u}}^{*(t_1)} mc)^{1/2}.$$

*On the other hand, $g(\mathbf{x}; \mathbf{W}^{(t)}) \leq c$ implies that for all $t \in [t_1, t_2]$,*

$$\max_{r \in [m]} \langle \mathbf{w}_{y,r}^{(t)}, y\mathbf{u} \rangle \leq (\beta_{\mathbf{u}}^{*(t_1)} mc)^{1/2}.$$

*Here $\beta_{\mathbf{u}}^{*(t)}$ is defined as*

$$\beta_{\mathbf{u}}^{*(t)} = \frac{\max_{r \in [m]} \sigma\big(\langle \mathbf{w}_{y,r}^{(t)}, y\mathbf{u} \rangle\big)}{\sum_{r \in [m]} \sigma\big(\langle \mathbf{w}_{y,r}^{(t)}, y\mathbf{u} \rangle\big)}.$$

*Proof of Lemma 15.* See Appendix C.4.2 for a detailed proof. For multiple training data setting, the proof is in Appendix E.6.2. □

Being a subtle condition required in the previous two lemmas, whether the signs of these inner products are invariant throughout the process remains unknown, making the behaviors of these inner products complicated. The answer to this question is affirmative, as we are able to prove that, under proper conditions, the signs of these inner products are fixed throughout the process. Using Lemmas 14 and 15, we prove the boundedness and the sign stability simultaneously through a sophisticated inductive argument. The formal statement of this result is as follows.

We first define a stopping time. Let

$$T_{(\mathbf{v})} = \min_{t \geq 0} \left\{ t : \frac{1}{m} \sum_{r \in [m]} \sigma\big(\langle \mathbf{w}_{y,r}^{(t)}, y\mathbf{v} \rangle\big) > \delta \right\}. \tag{13}$$

Such a stopping time helps to control the non-dominating terms in the CNN. Moreover, once the process reaches $T_{(\mathbf{v})}$, the conclusion in Theorem 13 is nearly achieved. Our results are summarized as follows.

**Lemma 16** (Boundedness and sign stability: single training data case). *Suppose that Assumptions 11 and 12 hold. With probability at least $1 - p = 1 - 1/\mathrm{poly}(d)$, we have the following bounds:*

$$\max_r \langle \mathbf{w}_{y,r}^{(t)}, y\mathbf{u} \rangle \leq 1.5 \cdot \big(1.05 \beta_{\mathbf{u}}^* m\big)^{1/2}, \quad \forall t \leq T_{(\mathbf{v})}, \tag{14}$$

$$\min_r \langle -\mathbf{w}_{-y,r}^{(t)}, -y\mathbf{u} \rangle \geq -\sqrt{2 \log(16m/p)} \cdot \sigma_0 \|\mathbf{u}\|_2, \quad \forall t \leq T_{(\mathbf{v})}, \tag{15}$$

$$\min_r \langle -\mathbf{w}_{-y,r}^{(t)}, -y\mathbf{v} \rangle \geq -\sqrt{2 \log(16m/p)} \cdot \sigma_0 \|\mathbf{v}\|_2, \quad \forall t \leq T_{(\mathbf{v})}, \tag{16}$$

$$0 \leq y f(\mathbf{x}; \mathbf{W}^{(t)}) \leq 3, \quad \forall t \leq T_{(\mathbf{v})}.$$

*Here $\beta_{\mathbf{u}}^* := \beta_{\mathbf{u}}^{*(0)}$ is defined in Lemma 15. Besides, the sign stability holds before $T_{(\mathbf{v})}$, i.e. $\mathcal{U}_{\pm y,\pm}^{(t)}$ and $\mathcal{V}_{\pm y,\pm}^{(t)}$ remain invariant in $t$, and the superscript $(t)$ can be dropped.*

*Proof of Lemma 16.* See Appendix C.4.3 for a detailed proof. □

Thanks to the stopping time defined in (13) which puts controls on the scale of $\langle \mathbf{w}_{j,r}^{(t)}, j\mathbf{v}\rangle$, the major part function $g$ defined in the previous lemma dominates the entire CNN, as the negative parts $\langle \mathbf{w}_{-y,r}^{(t)}, -y\mathbf{u}\rangle$, $\langle \mathbf{w}_{-y,r}^{(t)}, -y\mathbf{v}\rangle$ can be lower bounded in Lemma 16 through a delicate analysis of the training dynamics.

Lemmas 14, 15, and 16 reveal the key properties of the training dynamics and the two-layer CNN. Several remarks are put here again. Lemmas 14 and 15 are temporarily *local*, with the condition on the local sign stability. They are not informative until we are able to extend the sign stability to a wider sense, which is achieved by Lemma 16. Nevertheless, these two local lemmas are used frequently throughout the subsequent analysis, so we single them out here to make the proof more readable.

## C.2 Fundamental Reasoning towards the Weak Signal Learning

The previous section presents several basic properties of the training dynamics and the CNN. However, they are insufficient in interpreting the driving force of the weak signal learning during oscillation. In the lemma below, we discover a quantitative interpretation towards the increasing on $\langle \mathbf{w}_{y,r}^{(t)}, y\mathbf{v}\rangle$, which formalizes the illustration of the function of oscillation in Section 3.

**Lemma 17** (Weak signal learning: single training data case). *Under Assumptions 11 and 12, suppose that there exists $t_0 \le t_1$, such that:*

1. *The sign stability holds on $[t_0, t_1]$, i.e., i.e. $\mathcal{U}_{\pm y, \pm}^{(t)}$ and $\mathcal{V}_{\pm y, \pm}^{(t)}$ remain invariant in $t$;*

2. *$\max_{r \in [m]} \langle \mathbf{w}_{y,r}^{(t)}, y\mathbf{u}\rangle < B \cdot (\beta_{\mathbf{u}}^* m)^{1/2}, \ \forall t \in [t_0, t_1]$ for some $B > 0$;*

3. *$\min_{r \in [m]} \langle \mathbf{w}_{-y,r}^{(t)}, -y\mathbf{u}\rangle > -0.1$ and $\min_{r \in [m]} \langle \mathbf{w}_{-y,r}^{(t)}, -y\mathbf{v}\rangle > -0.1, \ \forall t \in [t_0, t_1]$;*

4. *$\frac{1}{m}\sum_{r \in [m]} \sigma(\langle \mathbf{w}_{y,r}^{(t)}, y\mathbf{v}\rangle) < \delta, \ \forall t \in [t_0, t_1]$;*

5. *$-2 \le 1 - yf(\mathbf{x}; \mathbf{W}^{(t)}) \le 1, \ \forall t \in [t_0, t_1]$.*

*Then we have that*

$$\sum_{s=t_0}^{t_1-1} \left(1 - yf(\mathbf{x}; \mathbf{W}^{(s)})\right) \ge 2\epsilon \cdot (t_1 - t_0) - \frac{mB}{\eta \|\mathbf{u}\|_2^2 \sqrt{1.05 - \delta}}.$$

*Consequently, we further have that for any $r \in \mathcal{V}_{y,+} = \{r \in [m] : \langle \mathbf{w}_{y,r}^{(t_0)}, y\mathbf{v}\rangle > 0\}$, it holds that*

$$\langle \mathbf{w}_{y,r}^{(t_1)}, y\mathbf{v}\rangle \ge \langle \mathbf{w}_{y,r}^{(t_0)}, y\mathbf{v}\rangle \cdot \exp\left\{\frac{\eta\|\mathbf{v}\|_2^2\epsilon}{m} \cdot (t_1 - t_0) - \frac{\|\mathbf{v}\|_2^2}{\|\mathbf{u}\|_2^2} \cdot \frac{B}{(1.05 - \delta)^{1/2}}\right\}.$$

*Here $\epsilon = (\delta - \delta(1.05 - \delta)^{1/2})/4$ with $\delta$ specified in Assumption 12.*

*Proof of Lemma 17.* See Appendix C.5 for a detailed proof. □

This lemma asserts that, stable oscillation in a bounded and favorable training area leads to a linear increasing lower bound for $\sum_t (1 - yf_t)$. This is the first part of Lemma 17. The second part of this lemma relates the increasing speed of $\langle \mathbf{w}_{y,r}^{(t)}, y\mathbf{v}\rangle$ to the summation of $1 - yf_t$. The derivation of this part relies on the fact that $\alpha = \|\mathbf{v}\|_2^2 / \|\mathbf{u}\|_2^2 \in (0, 1)$ and is close to 0. This observation motivates us to approximate the ratio with a first-order Taylor expansion,

$$\frac{\langle \mathbf{w}_{y,r}^{(t_1)}, y\mathbf{v}\rangle}{\langle \mathbf{w}_{y,r}^{(t_0)}, y\mathbf{v}\rangle} = \prod_{t'=t_0}^{t_1-1} \left\{1 + \alpha\tilde{\eta}\left(1 - yf(\mathbf{x}; \mathbf{W}^{(t')})\right)\right\}$$

$$\approx 1 + \alpha\tilde{\eta}\sum_{t'=t_0}^{t_1-1} \left(1 - yf(\mathbf{x}; \mathbf{W}^{(t')})\right).$$

The proof of the second part of this lemma justifies this intuition formally with more delicate analysis. With all these collected results, we are ready to present the proof of the main theorem.

## C.3 PROOF OF THEOREM 13

*Proof of Theorem 13.* We prove Theorem 13 by contradiction. Recall that in the previous section we have defined that

$$T_{(\mathbf{v})} = \min_{t \geq 0} \left\{ t : \frac{1}{m} \sum_{r \in [m]} \sigma\big(\langle \mathbf{w}_{y,r}^{(t)}, y\mathbf{v} \rangle\big) > \delta \right\}.$$

With this definition, our first goal is to prove that $T_{(\mathbf{v})}$ is bounded by a finite time with explicit expression. To put it precisely, we are going to prove by contradiction that

$$T_{(\mathbf{v})} < T_0 := \frac{m}{\eta \|\mathbf{v}\|_2^2 \epsilon} \cdot \left\{ \log \left( \frac{2\sqrt{m}\delta}{\sigma_0 \|\mathbf{v}\|_2} \right) + 1.5 \cdot \frac{\|\mathbf{v}\|_2^2}{\|\mathbf{u}\|_2^2} \cdot \sqrt{\frac{1.05}{1 - \delta}} \right\},$$

where $\epsilon$ is specified in Lemma 17 and $\delta$ is specified in Assumption 12. Suppose otherwise that $T_{(\mathbf{v})} \geq T_0$. By Lemma 16 we can see that for $t \in [0, T_0]$ all the conditions of Lemma 17 are satisfied. Then Lemma 17 implies that,

$$\langle \mathbf{w}_{y,r^*}^{(T_0)}, y\mathbf{v} \rangle \geq \langle \mathbf{w}_{y,r^*}^{(0)}, y\mathbf{v} \rangle \cdot \exp \left\{ \frac{\eta \|\mathbf{v}\|_2^2 \epsilon}{m} \cdot T_0 - 1.5 \cdot \frac{\|\mathbf{v}\|_2^2}{\|\mathbf{u}\|_2^2} \cdot \sqrt{\frac{1.05}{1.05 - \delta}} \right\}$$

$$\geq \frac{1}{2} \sigma_0 \|\mathbf{v}\|_2 \cdot \frac{2\sqrt{m}\delta}{\sigma_0 \|\mathbf{v}\|_2} \geq \sqrt{m}\delta.$$

where $r^* = \operatorname{argmax}_{r \in [m]} \langle \mathbf{w}_{y,r}^{(t)}, y\mathbf{v} \rangle$ are fixed throughout $t \in [0, T_0]$ (see Lemma 14) and in the second inequality we apply Lemma 10 to lower bound the initialization. This leads to the following,

$$\frac{1}{m} \sum_{r \in [m]} \sigma\big(\langle \mathbf{w}_{y,r}^{(T_0)}, y\mathbf{v} \rangle\big) \geq \frac{1}{m} \sigma\big(\langle \mathbf{w}_{y,r^*}^{(T_0)}, y\mathbf{v} \rangle\big) \geq \delta,$$

which contradicts the definition of $T_{(\mathbf{v})}$, and therefore $T_{(\mathbf{v})} < T_0$.

The rest part of the proof is to show that the sequence

$$\left\{ \frac{1}{m} \sum_{r \in [m]} \sigma\big(\langle \mathbf{w}_{y,r}^{(t)}, y\mathbf{v} \rangle\big) \right\}_{t \geq T_{(\mathbf{v})}} \tag{17}$$

does not fall below $\delta/2$. Intuitively, as long as the sequence above falls below $\delta$, the sequence would have the incentive to increase, as the results in Lemma 17 are revived again based on another boundedness argument. The analysis resembles the proof of Lemma 16 with slight differences. We provide the following proposition with the proofs delayed to Appendix C.6.

**Proposition 18** (Weak signal memorization). *It holds that*

$$\frac{1}{m} \sum_{r \in [m]} \sigma\big(\langle \mathbf{w}_{y,r}^{(t)}, y\mathbf{v} \rangle\big) > \frac{\delta}{2}, \quad \forall t \geq T_{(\mathbf{v})}.$$

*Proof of Proposition 18.* See Appendix C.6.1 for a detailed proof. □

This proposition finalizes the proof of Theorem 13, and we are done. □

## C.4 PROOF OF LEMMAS IN APPENDIX C.1

### C.4.1 PROOF OF LEMMA 14

*Proof of Lemma 14.* By our assumption that the signs of the inner products does not change throughout $[t_1, t_2]$, it is straightforward that $\mathcal{U}_{y,+}^{(t)} = \mathcal{U}_{y,+}^{(t_1)}$ for every $t \in [t_1, t_2]$. The same is true for $\mathcal{U}_{-y,-}^{(t)}$. Therefore, we are able to drop the superscript $(t)$ temporarily, as the attention is restricted to a local interval $[t_1, t_2]$.

Regarding the maximal index, introducing $r' \neq r \in \mathcal{U}_{y,+}$, we have following relation

$$
\begin{aligned}
\operatorname*{argmax}_{r \in [m]} \langle \mathbf{w}_{y,r}^{(t)}, y\mathbf{u} \rangle &= \operatorname*{argmax}_{r \in \mathcal{U}_{y,+}} \langle \mathbf{w}_{y,r}^{(t)}, y\mathbf{u} \rangle, \\
&= \operatorname*{argmax}_{r \in \mathcal{U}_{y,+}} \langle \mathbf{w}_{y,r}^{(t)}, y\mathbf{u} \rangle / \langle \mathbf{w}_{y,r'}^{(t)}, y\mathbf{u} \rangle \\
&= \operatorname*{argmax}_{r \in \mathcal{U}_{y,+}} \frac{\langle \mathbf{w}_{y,r}^{(t_1)}, y\mathbf{u} \rangle \cdot \prod_{t'=t_1}^{t-1} \left( 1 + \frac{2\eta \|\mathbf{u}\|_2^2}{m} \cdot \left( 1 - yf(\mathbf{x}; \mathbf{W}^{(t)}) \right) \right)}{\langle \mathbf{w}_{y,r'}^{(t_1)}, y\mathbf{u} \rangle \cdot \prod_{t'=t_1}^{t-1} \left( 1 + \frac{2\eta \|\mathbf{u}\|_2^2}{m} \cdot \left( 1 - yf(\mathbf{x}; \mathbf{W}^{(t)}) \right) \right)} \\
&= \operatorname*{argmax}_{r \in \mathcal{U}_{y,+}} \langle \mathbf{w}_{y,r}^{(t_1)}, y\mathbf{u} \rangle / \langle \mathbf{w}_{y,r'}^{(t_1)}, y\mathbf{u} \rangle \\
&= \operatorname*{argmax}_{r \in \mathcal{U}_{y,+}} \langle \mathbf{w}_{y,r}^{(t_1)}, y\mathbf{u} \rangle.
\end{aligned}
$$

The same relation can be verified for $\langle \mathbf{w}_{-y,r}^{(t)}, y\mathbf{u} \rangle$ with $r \in \mathcal{U}_{-y,-}$, finishing the proof. $\qquad \square$

### C.4.2 PROOF OF LEMMA 15

*Proof of Lemma 15.* Again, the local sign stability assumption ensures that each inner product grows proportionally and the superscript $(t)$ in the neuron index sets can be dropped, with

$$
\begin{aligned}
g(\mathbf{x}, y; \mathbf{W}^{(t)}) &= \frac{1}{m} \sum_{r \in [m]} \sigma\left( \langle \mathbf{w}_{y,r}^{(t)}, y\mathbf{u} \rangle \right) \\
&= \frac{1}{m} \sum_{r \in \mathcal{U}_{y,+}} \sigma\left( \langle \mathbf{w}_{y,r}^{(t_1)}, y\mathbf{u} \rangle \right) \cdot \prod_{t'=t_1}^{t-1} \left( 1 + \frac{2\eta \|\mathbf{u}\|_2^2}{m} \cdot \left( 1 - yf(\mathbf{x}; \mathbf{W}^{(t')}) \right) \right)^2 \\
&= \frac{\max_{r \in [m]} \sigma\left( \langle \mathbf{w}_{y,r}^{(t_1)}, y\mathbf{u} \rangle \right)}{m \beta_{\mathbf{u}}^{*,(t_1)}} \prod_{t'=t_1}^{t-1} \left( 1 + \frac{2\eta \|\mathbf{u}\|_2^2}{m} \cdot \left( 1 - yf(\mathbf{x}; \mathbf{W}^{(t')}) \right) \right)^2 \\
&= \frac{\sigma\left( \max_{r \in [m]} \langle \mathbf{w}_{y,r}^{(t)}, y\mathbf{u} \rangle \right)}{m \beta_{\mathbf{u}}^{*,(t_1)}}.
\end{aligned}
$$

Here the second line and the last equality is true because (8) implies that all the positive $\langle \mathbf{w}_{y,r}^{(t)}, y\mathbf{u} \rangle$ iterates by sequentially multiplying the same factor

$$
1 + \frac{2\eta \|\mathbf{u}\|_2^2}{m} \cdot \left( 1 - yf(\mathbf{x}; \mathbf{W}^{(t')}) \right).
$$

The third equality comes from the definition of $\beta_{\mathbf{u}}^{*,(t_1)}$ in Lemma 15. Thus, $g(\mathbf{x}, y; \mathbf{W}^{(t)}) \geq c$ implies that

$$
\sigma\left( \max_{r \in [m]} \langle \mathbf{w}_{y,r}^{(t)}, y\mathbf{u} \rangle \right) > \beta_{\mathbf{u}}^{*,(t_1)} \cdot mc
$$

and the desired lower bound follows. The upper bound can be proved analogously and is omitted here. $\qquad \square$

### C.4.3 PROOF OF LEMMA 16

*Proof of Lemma 16.* A roadmap is provided to help understand how every single step is achieved so that the readers can skip the details without leaving the key ideas behind.

**Recap on notations.** Recall that, $\mathcal{U}_{j,+}^{(t)}$ is the set of indices $r \in [m]$ such that $\langle \mathbf{w}_{j,r}^{(t)}, j\mathbf{u} \rangle > 0$ and $\mathcal{U}_{j,-}^{(t)}$ is the set of indices $r \in [m]$ such that $\langle \mathbf{w}_{j,r}^{(t)}, j\mathbf{u} \rangle \leq 0$. Specially, let $\mathcal{U}_{y,+} = \mathcal{U}_{y,+}^{(0)}$ and $\mathcal{U}_{-y,-} = \mathcal{U}_{-y,-}^{(0)}$. With probability one, it holds that $\mathcal{U}_{j,-} \cup \mathcal{U}_{j,+} = [m]$. Let $r^* := \operatorname*{argmax}_{r \in [m]} \langle \mathbf{w}_{y,r}^{(0)}, y\mathbf{u} \rangle$

and $r_* := \text{argmin}_{r \in [m]} \langle \mathbf{w}_{-y,r}^{(0)}, -y\mathbf{u} \rangle$, i.e., $r^*$ (resp. $r_*$) denote the index of the maximum (resp. minimum) throughout the process as we are able to extend the results in Lemma 14 globally.

We also introduce several supplemental notations here to facilitate the proof. Recursively, we define

$$\bar{T}_k := \min_{t \geq 0} \left\{ t : t > \bar{T}_{k-1}, \ yf(\mathbf{x}; \mathbf{W}^{(t)}) \geq 1 \text{ and } yf(\mathbf{x}; \mathbf{W}^{(t-1)}) < 1 \right\}, \tag{18}$$

with $\bar{T}_0 = 0$. Similarly, we define that

$$\underline{T}_k := \min_{t \geq 0} \left\{ t : t > \underline{T}_{k-1}, \ yf(\mathbf{x}; \mathbf{W}^{(t)}) < 1 \text{ and } yf(\mathbf{x}; \mathbf{W}^{(t-1)}) \geq 1 \right\},$$

and $\underline{T}_0 = 0$. Intuitively, $\bar{T}_k$ captures the times that $yf(\mathbf{x}; \mathbf{W}^{(t)})$ just exceeds 1, and similarly for $\underline{T}_k$.

**Roadmap.** From a high level, three steps are required to establish the full proof:

1. We verify that the lower bound in Inequality (15) in Lemma 16 holds for all $t \in [0, \bar{T}_1]$ with a direct monotonicity argument. Additionally, we can prove that the upper bound in Inequality (14) holds for $t \in [0, \bar{T}_1)$ by using Lemma 15 and the definition of $\bar{T}_1$. The signs do not change in this stage, as shown in the details below.

2. We extend the results in Lemma 16 to $t \in [\bar{T}_1, \bar{T}_2)$ with repeated use of Lemma 15. The sign stability is guaranteed from an intermediate upper bound on $|1 - yf(\mathbf{x}; \mathbf{W}^{(t)})|$.

3. Note that the condition on $[0, \bar{T}_1)$ (which is proved in the first step) required for the proof of the second step is again true for $t \in [0, \bar{T}_2)$, which is a consequence of the second step. Therefore we can repeat the second step to extend the results in Lemma 16 to $t \in [\bar{T}_2, \bar{T}_3)$, and so on. So the results are true for all $t \leq T_{(\mathbf{v})}$.

The first and the last step above are relatively straightforward. However, the second step requires a delicate break-down analysis. Here we provide a more detailed roadmap for the second step. The goal is to prove the results in Lemma 16, restricted to $t \in [\bar{T}_1, \bar{T}_2)$. This would be achieved in four split steps:

2.1 Firstly, we prove the upper bound in Inequality (14) for $t = \bar{T}_1$ by tracking one-step gradient descent and the upper bound for $\langle \mathbf{w}_{y,r^*}^{(\bar{T}_1 - 1)}, y\mathbf{u} \rangle$. Then with a monotonicity argument,

$$\langle \mathbf{w}_{y,r^*}^{(t)}, y\mathbf{u} \rangle \leq \langle \mathbf{w}_{y,r^*}^{(\bar{T}_1)}, y\mathbf{u} \rangle, \quad \forall t \in [\bar{T}_1, \underline{T}_1].$$

Besides, for $t \in [\bar{T}_1, \underline{T}_1)$ we can give a lower bound on $\langle \mathbf{w}_{y,r^*}^{(t)}, y\mathbf{u} \rangle$ with the help of Lemma 15.

2.2 Based on the previous lower and upper bounds on $\langle \mathbf{w}_{y,r^*}^{(\bar{T}_1)}, y\mathbf{u} \rangle$, we can derive a lower bound on $\langle \mathbf{w}_{y,r^*}^{(\underline{T}_1)}, y\mathbf{u} \rangle$ by tracking one-step gradient descent. We apply this worst-case tight lower bound to conclude the sign stability. We note that this step is free of the lower bound on $\langle \mathbf{w}_{-y,r_*}^{(t)}, -y\mathbf{u} \rangle$ for $t \in (\bar{T}_1, \underline{T}_2]$, which we have not yet proved to be true.

2.3 Now we give lower bounds on $\langle \mathbf{w}_{-y,r_*}^{(\underline{T}_1)}, -y\mathbf{u} \rangle$ and $\langle \mathbf{w}_{-y,r_*}^{(\underline{T}_1)}, -y\mathbf{v} \rangle$. We achieve this by a delicate usage of the lower bound on $\langle \mathbf{w}_{y,r^*}^{(\underline{T}_1)}, y\mathbf{u} \rangle$ (which we have proved in Step 2.2) plus an inequality that connects the relative increment of $\langle \mathbf{w}_{y,r^*}^{(t)}, y\mathbf{u} \rangle$ and $\langle \mathbf{w}_{-y,r_*}^{(t)}, -y\mathbf{u} \rangle$ (or $\langle \mathbf{w}_{-y,r_*}^{(t)}, -y\mathbf{v} \rangle$). Thus we prove Inequalities (15) and (16) for $t = \underline{T}_1$, and this can be further extended to the entire $[\bar{T}_1, \bar{T}_2]$ by another monotonicity argument since $\underline{T}_1$ is the local minima.

2.4 The remaining to is upper bound $\langle \mathbf{w}_{y,r^*}^{(t)}, y\mathbf{u} \rangle$ for $t \in (\underline{T}_1, \bar{T}_2)$. This is again a consequence of Lemma 15, as exactly what has been done to upper bound $\langle \mathbf{w}_{y,r^*}^{(t)}, y\mathbf{u} \rangle, t \leq \bar{T}_1$ in Step 1.

Now with the roadmap in mind, we are ready to dive into the details of every step.

**Step 1: Pre-$\bar{T}_1$ Analysis.** Lemma 10 indicates that the lower bound in Inequality (15) holds at initialization $t = 0$ under Assumption 11. Moreover, the upper bounds on the maximal initial inner products in Lemma 10 with Assumption 11 indicate that

$$|yf(\mathbf{x}; \mathbf{W}^{(0)})| \leq 2 \cdot \max_{r \in [m]} \langle \mathbf{w}_{y,r}^{(0)}, y\mathbf{u} \rangle^2 \vee \max_{r \in [m]} \langle \mathbf{w}_{-y,r}^{(0)}, -y\mathbf{u} \rangle^2$$

$$= \widetilde{\mathcal{O}}\big(\sigma_0^2 \|\mathbf{u}\|_2^2\big) \ll 1.$$

From this we know that $\bar{T}_1 \geq 1$ and the upper bound in Inequality (14) is true at $t = 0$.

Then the first step to do is to extend the lower on $\langle \mathbf{w}_{-y,r}^{(t)}, -y\mathbf{u} \rangle$ to $[1, \bar{T}_1]$. Definition of $\bar{T}_1$ implies that $yf(\mathbf{x}; \mathbf{W}^{(t)}) \leq 1$ for $t \in [0, \bar{T}_1)$. Therefore for $r \in \mathcal{U}_{-y,-}^{(0)}$, Equation (9) gives that

$$\langle \mathbf{w}_{-y,r}^{(t+1)}, -y\mathbf{u} \rangle = \langle \mathbf{w}_{-y,r}^{(t)}, -y\mathbf{u} \rangle + \frac{\eta \|\mathbf{u}\|_2^2}{m} \cdot \big(1 - yf(\mathbf{x}; \mathbf{W}^{(t)})\big) \cdot \sigma'\big(-\langle \mathbf{w}_{-y,r}^{(t)}, -y\mathbf{u} \rangle\big)$$

$$= \langle \mathbf{w}_{-y,r}^{(t)}, -y\mathbf{u} \rangle - \frac{2\eta \|\mathbf{u}\|_2^2}{m} \cdot \big(1 - yf(\mathbf{x}; \mathbf{W}^{(t)})\big) \cdot \langle \mathbf{w}_{-y,r}^{(t)}, -y\mathbf{u} \rangle \quad (19)$$

$$\geq \langle \mathbf{w}_{-y,r}^{(t)}, -y\mathbf{u} \rangle.$$

And furthermore

$$\langle \mathbf{w}_{-y,r}^{(\bar{T}_1)}, -y\mathbf{u} \rangle \geq \langle \mathbf{w}_{-y,r}^{(0)}, -y\mathbf{u} \rangle \geq -\sqrt{2\log(16m/p)} \cdot \sigma_0 \|\mathbf{u}\|_2.$$

Taking minimum with respect to $r \in [m]$ gives the result. Same argument can be applied to $\langle \mathbf{w}_{-y,r}^{(\bar{T}_1)}, -y\mathbf{v} \rangle$. So the lower bounds in Inequalities (15) and (16) hold for $t \in [1, \bar{T}_1]$.

Also, (8) and (10) imply that $\langle \mathbf{w}_{y,r}^{(t)}, y\mathbf{u} \rangle, r \in \mathcal{U}_{y,+}^{(0)}$ and $\langle \mathbf{w}_{y,r}^{(t)}, y\mathbf{v} \rangle, r \in \mathcal{V}_{y,+}^{(0)}$ increase for all $t < \bar{T}_1$. A natural consequence is that $yf(\mathbf{x}; \mathbf{W}^{(t)})$ is non-decreasing in $t$ in this stage, since every summand (possibly with the negative sign before) in the summation is non-decreasing.

Now we can prove the sign stability. For $r \in \mathcal{U}_{y,+}^{(0)}$, we know from (8) that $\langle \mathbf{w}_{y,r}^{(t)}, y\mathbf{u} \rangle > \langle \mathbf{w}_{y,r}^{(0)}, y\mathbf{u} \rangle > 0$, hence $r \in \mathcal{U}_{y,+}^{(t)}$ is non-vanishing in $t$ by induction. Additionally, for $r \in \mathcal{U}_{y,-}^{(0)}$, $\langle \mathbf{w}_{y,r}^{(t)}, y\mathbf{u} \rangle$ stays fixed in $t$, as mentioned in Appendix C.1. Two points together ensure that $\mathcal{U}_{y,+}^{(t)} = \mathcal{U}_{y,+}^{(0)}$ for $t \in [1, \bar{T}_1]$.

For $r \in \mathcal{U}_{-y,-}^{(0)}$, let's take a closer look at Equation (19) with $t = 0$:

$$\langle \mathbf{w}_{-y,r}^{(1)}, -y\mathbf{u} \rangle = \langle \mathbf{w}_{-y,r}^{(0)}, -y\mathbf{u} \rangle - \frac{2\eta \|\mathbf{u}\|_2^2}{m} \cdot \big(1 - yf(\mathbf{x}; \mathbf{W}^{(0)})\big) \cdot \langle \mathbf{w}_{-y,r}^{(0)}, -y\mathbf{u} \rangle$$

$$= \langle \mathbf{w}_{-y,r}^{(0)}, -y\mathbf{u} \rangle \cdot \underbrace{\left(1 - \frac{2\eta \|\mathbf{u}\|_2^2}{m} \cdot \big(1 - yf(\mathbf{x}; \mathbf{W}^{(0)})\big)\right)}_{>0 \text{ if } \eta < (1-o(1))/2 \cdot m \|\mathbf{u}\|_2^{-2}}.$$

Assumption 11 ensures that $\eta < 0.4m\|\mathbf{u}\|_2^{-2} < (1 - o(1))/2 \cdot m\|\mathbf{u}\|_2^{-2}$ so the sign change does not happen at $t = 0$. As mentioned before, $yf(\mathbf{x}; \mathbf{W}^{(t)}) \leq 1$ is non-decreasing in $t$ at this stage, and putting them together we know that

$$1 - \frac{2\eta \|\mathbf{u}\|_2^2}{m}\big(1 - yf(\mathbf{x}; \mathbf{W}^{(t)})\big) \geq 0, \quad \forall t \in [1, \bar{T}_1).$$

The same sign stability can be verified for $\langle \mathbf{w}_{-y,r}^{(t)}, -y\mathbf{v} \rangle$, and therefore, the sign stability for $\mathcal{U}_{-y,-}^{(t)}$ and $\mathcal{V}_{-y,-}^{(t)}$ is ensured for $t \in [0, \bar{T}_1]$ and the lower bound in Inequality (15) holds for $t \in [1, \bar{T}_1]$.

Now we turn to prove the upper bound (14). Note that $yf(\mathbf{x}; \mathbf{W}^{(t)}) \leq 1$ for $t < \bar{T}_1$, and the definition of $T_{(\mathbf{v})}$ then implies that

$$g(\mathbf{x}, y; \mathbf{W}^{(t)}) \leq 1 + 2\big(\sqrt{2\log(16m/p)} \cdot \sigma_0 \|\mathbf{u}\|_2\big)^2 \leq 1.05.$$

Now that the sign stability holds for $t \in [0, \bar{T}_1)$, Lemma 15 gives that

$$\max_{r \in [m]} \langle \mathbf{w}_{y,r}^{(t)}, y\mathbf{u} \rangle \leq \big(1.05\beta_{\mathbf{u}}^* m\big)^{1/2}. \quad (20)$$

Here $\beta_{\mathbf{u}}^*$ is defined in 15 with the superscript $(0)$ dropped.

**Step 2.1: Bounding** $\max_r \langle \mathbf{w}_{y,r}^{(t)}, y\mathbf{u} \rangle$ **for** $t \in [\bar{T}_1, \underline{T}_1)$**.** In order for the upper bound to be tight, we need a lower bound on $yf(\mathbf{x}; \mathbf{W}^{(\bar{T}_1-1)})$. Let $\tilde{\eta} = 2\eta \|\mathbf{u}\|_2^2 / m > 1/2$, we have the following result.

**Proposition 19.** *For every $k \geq 1$, suppose that*

$$E_{\bar{T}_k - 1} := \frac{1}{m} \sum_{r \in [m]} \sigma\big( -\langle \mathbf{w}_{-y,r}^{(\bar{T}_k-1)}, -y\mathbf{u} \rangle \big) + \sigma\big( -\langle \mathbf{w}_{-y,r}^{(\bar{T}_k-1)}, -y\mathbf{v} \rangle \big) < \frac{\delta}{2}.$$

*Then we have that*

$$yf(\mathbf{x}; \mathbf{W}^{(\bar{T}_k-1)}) \geq \frac{2 + \tilde{\eta} - \sqrt{\tilde{\eta}^2 + 4\tilde{\eta}}}{2\tilde{\eta}}. \tag{21}$$

*Moreover, it holds that*

$$yf(\mathbf{x}; \mathbf{W}^{(\bar{T}_k)}) \leq yf(\mathbf{x}; \mathbf{W}^{(\bar{T}_k-1)}) \cdot \Big(1 + \tilde{\eta} \cdot \big(1 - yf(\mathbf{x}; \mathbf{W}^{(\bar{T}_k-1)})\big)\Big)^2 + 2E_{\bar{T}_k-1}. \tag{22}$$

*Also, it is notable that the result here still holds for $\bar{T}_k \geq T_{(\mathbf{v})}$.*

*Proof of Proposition 19.* See Appendix C.6.2 for a detailed proof. $\qquad\square$

Clearly, the conditions required for Proposition 19 is true for before $\bar{T}_1$. So consider a one-step gradient descent at $t = \bar{T}_1 - 1$, by Proposition 19 and Equation (8),

$$\begin{aligned}
\langle \mathbf{w}_{y,r^*}^{(\bar{T}_1)}, y\mathbf{u} \rangle &= \langle \mathbf{w}_{y,r^*}^{(\bar{T}_1-1)}, y\mathbf{u} \rangle + \frac{\eta \|\mathbf{u}\|_2^2}{m} \cdot \big(1 - yf(\mathbf{x}; \mathbf{W}^{(\bar{T}_1-1)})\big) \cdot \sigma'\big(\langle \mathbf{w}_{y,r^*}^{(\bar{T}_1-1)}, y\mathbf{u} \rangle\big) \\
&\leq \langle \mathbf{w}_{y,r^*}^{(\bar{T}_1-1)}, y\mathbf{u} \rangle \cdot \left(1 + \tilde{\eta}\left(1 - \frac{\tilde{\eta} + 2 - \sqrt{\tilde{\eta}^2 + 4\tilde{\eta}}}{2\tilde{\eta}}\right)\right) \\
&\leq 1.5 \cdot \big(1.05 \beta_{\mathbf{u}}^* m\big)^{1/2}. 
\end{aligned} \tag{23}$$

Here the first inequality above comes from Inequality (21), and the second inequality is derived from taking the suprema $\tilde{\eta} = 1/2$ and Inequality (20). Additionally, we can further deliver an upper bound on $yf(\mathbf{x}; \mathbf{W}^{(\bar{T}_1)})$. Inequality (73) and Inequality (74) that have been proved to be true on $[0, \bar{T}_1]$ indicate that $E_{\bar{T}_1} = \widetilde{\mathcal{O}}(\sigma_0^2 \|\mathbf{u}\|_2^2) \ll 1$. Combining with Inequality (22), it holds that

$$\begin{aligned}
yf(\mathbf{x}; \mathbf{W}^{(\bar{T}_1)}) &\leq yf(\mathbf{x}; \mathbf{W}^{(\bar{T}_k-1)}) \cdot \Big(1 + \tilde{\eta} \cdot \big(1 - yf(\mathbf{x}; \mathbf{W}^{(\bar{T}_k-1)})\big)\Big)^2 + o(1) \\
&\leq \Big(1 + \tilde{\eta} \cdot \big(1 - yf(\mathbf{x}; \mathbf{W}^{(\bar{T}_k-1)})\big)\Big)^2 + o(1) \\
&\leq \Big(1 + \big(\tilde{\eta} - 1 - \tilde{\eta}/2 + \sqrt{\tilde{\eta}^2/4 + \tilde{\eta}}\big)\Big)^2 + o(1) \\
&= \big(\tilde{\eta}/2 + \sqrt{\tilde{\eta}^2/4 + \tilde{\eta}}\big)^2 + o(1).
\end{aligned} \tag{24}$$

Since $\tilde{\eta} < 1$, we know that $yf(\mathbf{x}; \mathbf{W}^{(\bar{T}_1)}) \leq 3$ and $|yf(\mathbf{x}; \mathbf{W}^{(\bar{T}_1)}) - 1| \leq 2$. (We keep $\tilde{\eta}$ in the upper bound above for deriving a sufficient condition on $\tilde{\eta}$ for the sign stability later.)

Now we look to the lower bound. The definition of $\bar{T}_1$, $T_{(\mathbf{v})}$ and Assumption 12 implies that $yf(\mathbf{x}; \mathbf{W}^{(t)}) > 1 + \delta$, hence $g(\mathbf{x}, y; \mathbf{W}^{(t)}) > 1 + \delta - \delta = 1$ for $t \in [\bar{T}_1, \underline{T}_1)$. Thus Lemma 15 implies that

$$\langle \mathbf{w}_{y,r^*}^{(t)}, y\mathbf{u} \rangle > (\beta_{\mathbf{u}}^* m)^{1/2}, \quad t \in [\bar{T}_1, \underline{T}_1).$$

**Step 2.2: Lower bounding** $\langle \mathbf{w}_{y,r}^{(\underline{T}_1)}, y\mathbf{u} \rangle$**.** Note that $yf(\mathbf{x}; \mathbf{W}^{(t)}) \leq yf(\mathbf{x}; \mathbf{W}^{(\bar{T}_1)})$ from the local monotonicity. Consider doing one-step gradient descent with Equation (8):

$$\begin{aligned}
\langle \mathbf{w}_{y,r^*}^{(\underline{T}_1)}, y\mathbf{u} \rangle &= \langle \mathbf{w}_{y,r^*}^{(\underline{T}_1-1)}, y\mathbf{u} \rangle \cdot \Big(1 - \tilde{\eta} \cdot \big(yf(\mathbf{x}; \mathbf{W}^{(\underline{T}_1-1)}) - 1\big)\Big) \\
&\geq \langle \mathbf{w}_{y,r^*}^{(\underline{T}_1-1)}, y\mathbf{u} \rangle \cdot \Big(1 - \tilde{\eta} \cdot \big(yf(\mathbf{x}; \mathbf{W}^{(\bar{T}_1-1)}) - 1\big)\Big) \\
&\geq (\beta_{\mathbf{u}}^* m)^{1/2} \cdot \underbrace{\Big(1 - \tilde{\eta}\big(\big(\tilde{\eta}/2 + \sqrt{\tilde{\eta}^2/4 + \tilde{\eta}}\big)^2 - 1\big) - o(1)\Big)}_{>0 \text{ with } \tilde{\eta} < 4/5}.
\end{aligned} \tag{25}$$

Here the last inequality is a consequence of Inequality (24) and the deterministic estimation that

$$\min_{\tilde{\eta}\in[1/2,4/5]}\left\{1 - \tilde{\eta}\Big(\big(\tilde{\eta}/2 + \sqrt{\tilde{\eta}^2/4 + \tilde{\eta}}\big)^2 - 1\Big)\right\} > 1/4. \tag{26}$$

We note that every step above is free of the lower bound in Inequality (15). Therefore, the sign stability is true on $[\underline{T}_1, \overline{T}_1]$, because all the inner products are non-decreasing in $t \in [\underline{T}_1, \overline{T}_2]$

**Step 2.3: Lower bounding $\langle \mathbf{w}_{-y,r}^{(T_1)}, -y\mathbf{u}\rangle$ and $\langle \mathbf{w}_{-y,r}^{(T_1)}, -y\mathbf{v}\rangle$.** The key is to notice that the sign stability on $[0, \underline{T}_1]$ guarantees that for every $t \leq \underline{T}_1 - 1$,

$$1 \pm \tilde{\eta} \cdot \big(1 - yf(\mathbf{x}; \mathbf{W}^{(t)})\big) > 0, \quad 1 \pm \tilde{\eta} \cdot \frac{\|\mathbf{v}\|_2^2}{\|\mathbf{u}\|_2^2} \cdot \big(1 - yf(\mathbf{x}; \mathbf{W}^{(t)})\big) > 0.$$

From Equation (9), we have that

$$\langle \mathbf{w}_{-y,r}^{(T_1)}, -y\mathbf{u}\rangle = \langle \mathbf{w}_{-y,}^{(0)}, -y\mathbf{u}\rangle \cdot \prod_{t=0}^{T_1-1} \Big(1 - \tilde{\eta}\big(1 - yf(\mathbf{x}; \mathbf{W}^{(t)})\big)\Big)$$

$$\geq \langle \mathbf{w}_{-y,}^{(0)}, -y\mathbf{u}\rangle \cdot \exp\left\{-\tilde{\eta}\sum_{t=0}^{T_1-1}\big(1 - yf(\mathbf{x}; \mathbf{W}^{(t)})\big)\right\}. \tag{27}$$

On the other hand,

$$\frac{\langle \mathbf{w}_{y,r^*}^{(T_1)}, y\mathbf{u}\rangle}{\langle \mathbf{w}_{y,r^*}^{(0)}, y\mathbf{u}\rangle} = \prod_{t=0}^{T_1-1}\Big(1 + \tilde{\eta}\cdot\big(1 - yf(\mathbf{x}; \mathbf{W}^{(t)})\big)\Big) \leq \exp\left\{\sum_{t=0}^{T_1-1}\tilde{\eta}\cdot\big(1 - yf(\mathbf{x}; \mathbf{W}^{(t)})\big)\right\}. \tag{28}$$

However, $\langle \mathbf{w}_{y,r^*}^{(0)}, y\mathbf{u}\rangle \leq (\beta_{\mathbf{u}}^*m)^{1/2}\cdot\sqrt{2\log(16m/p)}\cdot\sigma_0\|\mathbf{u}\|_2$ and Inequality (25) imply that

$$\frac{\langle \mathbf{w}_{y,r^*}^{(T_1)}, y\mathbf{u}\rangle}{\langle \mathbf{w}_{y,r^*}^{(0)}, y\mathbf{u}\rangle} \geq \frac{(\beta_{\mathbf{u}}^*m)^{1/2}\cdot(1/4 - o(1))}{(\beta_{\mathbf{u}}^*m)^{1/2}\cdot\sqrt{2\log(16m/p)}\cdot\sigma_0\|\mathbf{u}\|_2} \geq 1. \tag{29}$$

Combining Inequalities (28) and (29), we can obtain that $\sum_{t=0}^{T_1-1}\big(1 - yf(\mathbf{x}; \mathbf{W}^{(t)})\big) \geq 0$. Together with Inequality (27), this in turns leads to

$$0 > \min_{r\in[m]}\langle \mathbf{w}_{-y,r}^{(T_1)}, -y\mathbf{u}\rangle = \min_{r\in[m]}\langle \mathbf{w}_{-y,r}^{(0)}, -y\mathbf{u}\rangle \cdot \exp\left\{-\tilde{\eta}\sum_{t=0}^{T_1-1}\big(1 - yf(\mathbf{x}; \mathbf{W}^{(t)})\big)\right\}$$

$$\geq \min_{r\in[m]}\langle \mathbf{w}_{-y,r}^{(0)}, -y\mathbf{u}\rangle \geq -\sqrt{2\log(16m/p)}\cdot\sigma_0\|\mathbf{u}\|_2.$$

Analogously we have that

$$0 > \min_{r\in[m]}\langle \mathbf{w}_{-y,r}^{(T_1)}, -y\mathbf{v}\rangle = \langle \mathbf{w}_{-y,r}^{(0)}, -y\mathbf{v}\rangle \cdot \exp\left\{-\tilde{\eta}\cdot\frac{\|\mathbf{v}\|_2^2}{\|\mathbf{u}\|_2^2}\cdot\sum_{t=0}^{T_1-1}\big(1 - yf(\mathbf{x}; \mathbf{W}^{(t)})\big)\right\}.$$

$$\geq \min_{r\in[m]}\langle \mathbf{w}_{-y,r}^{(0)}, -y\mathbf{v}\rangle \geq -\sqrt{2\log(16m/p)}\cdot\sigma_0\|\mathbf{v}\|_2.$$

In conclusion, we have that

$$\max_{r\in[m]}\langle \mathbf{w}_{y,r}^{(t)}, y\mathbf{u}\rangle \geq 0.2(\beta_{\mathbf{u}}^*m)^{1/2}, \quad t \in [\overline{T}_1, \overline{T}_2],$$

$$\min_{r\in[m]}\langle \mathbf{w}_{-y,r}^{(t)}, -y\mathbf{u}\rangle \geq -\sqrt{2\log(16m/p)}\cdot\sigma_0\|\mathbf{u}\|_2, \quad t \in [\overline{T}_1, \overline{T}_2],$$

$$\min_{r\in[m]}\langle \mathbf{w}_{-y,r}^{(t)}, -y\mathbf{v}\rangle \geq -\sqrt{2\log(16m/p)}\cdot\sigma_0\|\mathbf{v}\|_2, \quad t \in [\overline{T}_1, \overline{T}_2].$$

Additionally, the lower bound on $\max_{r\in[m]}\langle \mathbf{w}_{y,r}^{(t)}, y\mathbf{u}\rangle$ indicates that the sign stability holds on $[\overline{T}_1, \overline{T}_2]$, since the last drop step does not change the sign of $\langle \mathbf{w}_{y,r}^{(t)}, y\mathbf{u}\rangle$. Furthermore, once the $\mathbf{u}$-sign stability holds, the $\mathbf{v}$-sign stability can be easily derived. To see this, we note that the $\mathbf{u}$-sign stability implies that, for any $t \in [0, T_{(\mathbf{v})}]$, it holds that $1 \pm 2\eta\|\mathbf{u}\|_2^2\cdot(1 - yf(\mathbf{x}; \mathbf{W}^{(t)}))/m > 0$. Now that $\|\mathbf{u}\|_2 > \|\mathbf{v}\|_2$, one clearly sees that $1 \pm 2\eta\|\mathbf{v}\|_2^2\cdot(1 - yf(\mathbf{x}; \mathbf{W}^{(t)}))/m > 0$ and the $\mathbf{v}$-sign stability holds.

**Step 2.4: Upper bounding** $\langle \mathbf{w}_{y,r}^{(t)}, y\mathbf{u} \rangle$ **for** $t \in (\underline{T}_1, \bar{T}_2)$**.** This is exactly the same as the proof of Inequality (20) in **Step 1**, and is thus omitted.

**Step 3. Finalizing proofs.** At this point, all the results in Lemma 16 have been proved to be true on $t \in [\bar{T}_1, \bar{T}_2]$. It is important to note that the only inductive hypothesis used for the local extension on $[\bar{T}_1, \bar{T}_2)$ is the lower bound on $\langle \mathbf{w}_{-y,r}^{(t)}, -y\mathbf{u} \rangle$ for $t \leq \bar{T}_1$. The rest part merely comes from the definition of $\bar{T}_1$ and $\underline{T}_2$ and Assumption 12. So repeating the same argument extends previous steps to all $t \leq T_{(\mathbf{v})}$. $\qquad \square$

**Remark 20.** *In the proof, we implicitly utilize the fact that* $\bar{T}_k, \underline{T}_k < +\infty, \ \forall k \geq 0$*. One may conjecture that there could be cases that for some* $k$*,* $yf(\mathbf{x}; \mathbf{W}^{(t)}) > 1$ *(resp.* $< 1$*) for all* $t \geq \bar{T}_k$ *(resp.* $\underline{T}_k$*). However, Assumption 12 (guaranteed by tuning a proper* $\eta$*) indicates that this cannot happen. One can argue that once* $yf(\mathbf{x}; \mathbf{W}^{(t)}) > 1$*, the Assumption 12 enables the dynamic to bounce back towards* $1$ *with at least exponential rate. Therefore,* $yf(\mathbf{x}; \mathbf{W}^{(t)})$ *falls below* $1 + \delta$ *within a few steps and Assumption 12 forces* $yf(\mathbf{x}; \mathbf{W}^{(t)}) < 1 - \delta$*, and* $\underline{T}_k < +\infty$*. Same argument can be used to prove that* $\bar{T}_k < +\infty$*.*

## C.5 PROOF OF FUNDAMENTAL REASONING (LEMMA 17)

*Proof of Lemma 17.* Let $r^* := \arg\max_{r \in [m]} \langle \mathbf{w}_{y,r}^{(t_0)}, y\mathbf{u} \rangle$. For the step $t_0 < t_1$, (8) implies that

$$
\langle \mathbf{w}_{y,r^*}^{(t_1)}, y\mathbf{u} \rangle = \langle \mathbf{w}_{y,r^*}^{(t_0)}, y\mathbf{u} \rangle + \frac{2\eta \|\mathbf{u}\|_2^2}{m} \cdot \sum_{\substack{s \in [t_0, t_1-1]: \\ yf(\mathbf{x}; \mathbf{W}^{(s)}) \geq 1}} \left(1 - yf(\mathbf{x}; \mathbf{W}^{(s)})\right) \cdot \langle \mathbf{w}_{y,r^*}^{(s)}, y\mathbf{u} \rangle
$$

$$
+ \frac{2\eta \|\mathbf{u}\|_2^2}{m} \cdot \sum_{\substack{s \in [t_0, t_1-1]: \\ yf(\mathbf{x}; \mathbf{W}^{(s)}) < 1}} \left(1 - yf(\mathbf{x}; \mathbf{W}^{(s)})\right) \cdot \langle \mathbf{w}_{y,r^*}^{(s)}, y\mathbf{u} \rangle. \tag{30}
$$

By Condition 4 in Lemma 17, for all $s \in [t_0, t_1]$, $\frac{1}{m} \sum_{r \in [m]} \sigma(\langle \mathbf{w}_{y,r}^{(s)}, y\mathbf{v} \rangle) < \delta$. Therefore by Assumption 12, for $s$ such that $yf(\mathbf{x}; \mathbf{W}^{(s)}) \geq 1$ (and thus $> 1 + \delta$), it holds from (12) that

$$
g(\mathbf{x}, y; \mathbf{W}^{(t)}) = \frac{1}{m} \sum_{r \in [m]} \sigma\left(\langle \mathbf{w}_{y,r}^{(s)}, y\mathbf{u} \rangle\right) \geq 1 + \delta - \delta = 1.
$$

Hence by Lemma 15, we have that

$$
\langle \mathbf{w}_{y,r^*}^{(s)}, y\mathbf{u} \rangle \geq (\beta_{\mathbf{u}}^* m)^{1/2}. \tag{31}
$$

On the other hand, by Conditions 3 in Lemma 17, for all $s \in [t_0, t_1]$, it holds that $\min_{r \in [m]} \langle \mathbf{w}_{-y,r}^{(s)}, -y\mathbf{u} \rangle > -0.1$ and $\min_{r \in [m]} \langle \mathbf{w}_{-y,r}^{(s)}, -y\mathbf{v} \rangle > -0.1$, which lead to

$$
\frac{1}{m} \sum_{r \in [m]} \sigma\left(-\langle \mathbf{w}_{-y,r}^{(s)}, -y\mathbf{u} \rangle\right) + \sigma\left(-\langle \mathbf{w}_{-y,r}^{(s)}, -y\mathbf{v} \rangle\right) \leq 2 \times 0.1^2 < 0.05.
$$

Therefore by Assumption 12, for $s$ such that $yf(\mathbf{x}; \mathbf{W}^{(s)}) \leq 1$ (and thus $< 1 - \delta$), it holds from (12) that

$$
g(\mathbf{x}, y; \mathbf{W}^{(t)}) = \frac{1}{m} \sum_{r \in [m]} \sigma\left(\langle \mathbf{w}_{y,r}^{(s)}, y\mathbf{u} \rangle\right) \leq 1 - \delta + 0.05.
$$

Hence by Lemma 15, we know that

$$
\langle \mathbf{w}_{y,r^*}^{(s)}, y\mathbf{u} \rangle \leq (1.05 - \delta)^{1/2} \cdot (\beta_{\mathbf{u}}^* m)^{1/2}. \tag{32}
$$

Meanwhile, Condition 1 (**u**-sign stability) and Condition 2 (boundedness) in Lemma 17 imply that

$$
\left| \langle \mathbf{w}_{y,r^*}^{(t_1)}, y\mathbf{u} \rangle - \langle \mathbf{w}_{y,r^*}^{(t_0)}, y\mathbf{u} \rangle \right| \leq \left| \langle \mathbf{w}_{y,r^*}^{(t_1)}, y\mathbf{u} \rangle \right| \vee \left| \langle \mathbf{w}_{y,r^*}^{(t_0)}, y\mathbf{u} \rangle \right| \leq B \cdot (\beta_{\mathbf{u}}^* m)^{1/2}. \tag{33}
$$

Putting Inequality (31), (32), (33) and Equation (30) together, we have that

$$
B \cdot (\beta_{\mathbf{u}}^* m)^{1/2} \geq \left| \frac{2\eta \|\mathbf{u}\|_2^2}{m} \cdot \sum_{\substack{s \in [t_0, t_1-1]: \\ y f(\mathbf{x}; \mathbf{W}^{(s)}) \geq 1}} \left(1 - y f(\mathbf{x}; \mathbf{W}^{(s)})\right) \cdot \langle \mathbf{w}_{y,r^*}^{(s)}, y\mathbf{u} \rangle \right.
$$

$$
\left. + \frac{2\eta \|\mathbf{u}\|_2^2}{m} \cdot \sum_{\substack{s \in [t_0, t_1-1]: \\ y f(\mathbf{x}; \mathbf{W}^{(s)}) < 1}} \left(1 - y f(\mathbf{x}; \mathbf{W}^{(s)})\right) \cdot \langle \mathbf{w}_{y,r^*}^{(s)}, y\mathbf{u} \rangle \right|
$$

$$
\geq \frac{2\eta \|\mathbf{u}\|_2^2}{m} \cdot \sum_{\substack{s \in [t_0, t_1-1]: \\ y f(\mathbf{x}; \mathbf{W}^{(s)}) \geq 1}} \left(y f(\mathbf{x}; \mathbf{W}^{(s)}) - 1\right) \cdot \langle \mathbf{w}_{y,r^*}^{(s)}, y\mathbf{u} \rangle
$$

$$
- \frac{2\eta \|\mathbf{u}\|_2^2}{m} \cdot \sum_{\substack{s \in [t_0, t_1-1]: \\ y f(\mathbf{x}; \mathbf{W}^{(s)}) < 1}} \left(1 - y f(\mathbf{x}; \mathbf{W}^{(s)})\right) \cdot \langle \mathbf{w}_{y,r^*}^{(s)}, y\mathbf{u} \rangle
$$

$$
\geq \frac{2\eta \|\mathbf{u}\|_2^2}{m} \cdot (\beta_{\mathbf{u}}^* m)^{1/2} \cdot \sum_{\substack{s \in [t_0, t_1-1]: \\ y f(\mathbf{x}; \mathbf{W}^{(s)}) \geq 1}} \left(y f(\mathbf{x}; \mathbf{W}^{(s)}) - 1\right)
$$

$$
- \frac{2\eta \|\mathbf{u}\|_2^2}{m} \cdot (\beta_{\mathbf{u}}^* m)^{1/2} \cdot (1.05 - \delta)^{1/2} \cdot \sum_{\substack{s \in [t_0, t_1-1]: \\ y f(\mathbf{x}; \mathbf{W}^{(s)}) < 1}} \left(1 - y f(\mathbf{x}; \mathbf{W}^{(s)})\right).
$$

This is also equivalent to

$$
\sum_{\substack{s \in [t_0, t_1-1]: \\ y f(\mathbf{x}; \mathbf{W}^{(s)}) < 1}} \left(1 - y f(\mathbf{x}; \mathbf{W}^{(s)})\right) \cdot \langle \mathbf{w}_{y,r^*}^{(s)}, y\mathbf{u} \rangle \geq \sum_{\substack{s \in [t_0, t_1-1]: \\ y f(\mathbf{x}; \mathbf{W}^{(s)}) \geq 1}} (1.05 - \delta)^{-1/2} \cdot \left(y f(\mathbf{x}; \mathbf{W}^{(s)}) - 1\right)
$$

$$
- \underbrace{\frac{mB}{2\eta \|\mathbf{u}\|_2^2 \sqrt{1.05 - \delta}}}_{:= \Delta(B)}. \tag{34}
$$

Now we are ready to lower bound the summation that we are interested in. Since

$$
|\{s \in [t_0, t_1 - 1] : y f(\mathbf{x}; \mathbf{W}^{(s)}) < 1\}| + |\{s \in [t_0, t_1 - 1] : y f(\mathbf{x}; \mathbf{W}^{(s)}) > 1\}| = t_1 - t_0,
$$

we have either $|\{s \in [t_0, t_1 - 1] : y f(\mathbf{x}; \mathbf{W}^{(s)}) > 1\}| > (t_1 - t_0)/2$, which by (34) implies that

$$
\sum_{s=t_0}^{t_1-1} \left(1 - y f(\mathbf{x}; \mathbf{W}^{(s)})\right) = \sum_{\substack{s \in [t_0, t_1-1]: \\ y f(\mathbf{x}; \mathbf{W}^{(s)}) < 1}} \left(1 - y f(\mathbf{x}; \mathbf{W}^{(s)})\right) - \sum_{\substack{s \in [t_0, t_1-1]: \\ y f(\mathbf{x}; \mathbf{W}^{(s)}) > 1}} \left(y f(\mathbf{x}; \mathbf{W}^{(s)}) - 1\right)
$$

$$
\geq \left((1.05 - \delta)^{-1/2} - 1\right) \cdot \sum_{\substack{s \in [t_0, t_1-1]: \\ y f(\mathbf{x}; \mathbf{W}^{(s)}) > 1}} \left(y f(\mathbf{x}; \mathbf{W}^{(s)}) - 1\right) - \Delta(B)
$$

$$
\geq \frac{1}{2} \left(\delta(1.05 - \delta)^{-1/2} - \delta\right) \cdot (t_1 - t_0) - \Delta(B), \tag{35}
$$

or $|\{s \in [t_0, t_1 - 1] : y f(\mathbf{x}; \mathbf{W}^{(s)}) < 1\}| > (t - t_0)/2$, which by (34) implies that

$$
\sum_{s=t_0}^{t_1-1} \left(1 - y f(\mathbf{x}; \mathbf{W}^{(s)})\right) = \sum_{\substack{s \in [t_0, t_1-1]: \\ y f(\mathbf{x}; \mathbf{W}^{(s)}) < 1}} \left(y f(\mathbf{x}; \mathbf{W}^{(s)}) - 1\right) - \sum_{\substack{s \in [t_0, t_1-1]: \\ y f(\mathbf{x}; \mathbf{W}^{(s)}) > 1}} \left(1 - y f(\mathbf{x}; \mathbf{W}^{(s)})\right)
$$

$$\geq \big(1 - (1.05 - \delta)^{1/2}\big) \cdot \sum_{\substack{s \in [t_0, t_1-1]: \\ yf(\mathbf{x}; \mathbf{W}^{(s)}) < 1}} \big(1 - yf(\mathbf{x}; \mathbf{W}^{(s)})\big) - (1.05 - \delta)^{1/2} \cdot \Delta(B)$$

$$\geq \frac{1}{2}\big(\delta - \delta(1.05 - \delta)^{1/2}\big) \cdot (t_1 - t_0) - (1.05 - \delta)^{1/2} \cdot \Delta(B). \tag{36}$$

In both two cases we have used Assumption 12 to bound $yf(\mathbf{x}; \mathbf{W}^{(s)})$ from 1. Combining (35) and (36), we have that

$$\sum_{s=t_0}^{t_1-1} \big(1 - yf(\mathbf{x}; \mathbf{W}^{(s)})\big) \geq \frac{1}{2}\big(\delta - \delta(1.05 - \delta)^{1/2}\big) \cdot (t_1 - t_0) - \Delta(B). \tag{37}$$

Now plugging in the definition of $\Delta(B)$ in (34), we have proved the linear increasing lower bound for the summation, which is the first conclusion of Lemma 17.

Then we turn to prove the second conclusion of Lemma 17. For simplicity, we denote $\alpha := \|\mathbf{v}\|_2^2/\|\mathbf{u}\|_2^2$ and $\epsilon = (\delta - \delta(1.05 - \delta)^{1/2})/4$. Note that from the $\mathbf{v}$-sign stability (Condition 1) and (10), we have that

$$1 + \frac{2\eta\|\mathbf{v}\|_2^2}{m} \cdot \big(1 - yf(\mathbf{x}; \mathbf{W}^{(t)})\big) > 0, \quad \forall t \in [t_0, t_1 - 1].$$

Therefore, we can lower bound the logarithmic ratio $\langle \mathbf{w}_{y,r}^{(t)}, y\mathbf{v}\rangle / \langle \mathbf{w}_{y,r}^{(0)}, y\mathbf{v}\rangle$ for $r \in \mathcal{V}_{y,+}$ as (recall that $\tilde{\eta} = 2\eta\|\mathbf{u}\|_2^2/m$)

$$\sum_{t=t_0}^{t_1-1} \log\Big(1 + \alpha\tilde{\eta}\big(1 - yf(\mathbf{x}; \mathbf{W}^{(t)})\big)\Big)$$

$$= \sum_{t=t_0}^{t_1-1} \int_0^\alpha \frac{\tilde{\eta}\big(1 - yf(\mathbf{x}; \mathbf{W}^{(t)})\big)}{1 + \tilde{\eta}z\big(1 - yf(\mathbf{x}; \mathbf{W}^{(t)})\big)} dz$$

$$= \sum_{t=t_0}^{t_1-1} \int_0^\alpha \frac{\tilde{\eta} \cdot \big(\big(1 - yf(\mathbf{x}; \mathbf{W}^{(t)})\big) + 2\big)}{1 + \tilde{\eta}z\big(1 - yf(\mathbf{x}; \mathbf{W}^{(t)})\big)} dz - \sum_{t=t_0}^{t_1-1} \int_0^\alpha \frac{2\tilde{\eta}}{1 + \tilde{\eta}z\big(1 - yf(\mathbf{x}; \mathbf{W}^{(t)})\big)} dz \tag{38}$$

Note that by Condition 5 in Lemma 37, $-2 \leq 1 - yf(\mathbf{x}; \mathbf{W}^{(t)}) \leq 1$, which further lower bounds (38) as

$$(38) \geq \sum_{t=t_0}^{t_1-1} \int_0^\alpha \frac{\tilde{\eta} \cdot \big(\big(1 - yf(\mathbf{x}; \mathbf{W}^{(t)})\big) + 2\big)}{1 + \tilde{\eta}z} dz - \sum_{t=t_0}^{t_1-1} \int_0^\alpha \frac{2\tilde{\eta}}{1 - 2\tilde{\eta}z} dz$$

$$= \int_0^\alpha \frac{\tilde{\eta} \cdot \big(\sum_{t=t_0}^{t_1-1} \big(1 - yf(\mathbf{x}; \mathbf{W}^{(t)})\big) + 2(t_1 - t_0)\big)}{1 + \tilde{\eta}z} dz - \int_0^\alpha \frac{2\tilde{\eta}(t_1 - t_0)}{1 - 2\tilde{\eta}z} dz. \tag{39}$$

Now applying (37) to the summation in (39), we can arrive at

$$(39) \geq \int_0^\alpha \frac{\tilde{\eta} \cdot \big(2\epsilon(t_1 - t_0) - \Delta(B) + 2(t_1 - t_0)\big)}{1 + \tilde{\eta}z} dz - \int_0^\alpha \frac{2\tilde{\eta}(t_1 - t_0)}{1 - 2\tilde{\eta}z} dz$$

$$\geq \big(2\epsilon(t_1 - t_0) - \Delta(B) + 2(t_1 - t_0)\big) \cdot \log(1 + \alpha\tilde{\eta}) + (t_1 - t_0) \cdot \log(1 - 2\alpha\tilde{\eta})$$

$$\geq \big(2\epsilon(t_1 - t_0) - \Delta(B)\big) \cdot \log(1 + \alpha\tilde{\eta}) + (t_1 - t_0) \cdot \log\Big((1 + \alpha\tilde{\eta})^2 \cdot (1 - 2\alpha\tilde{\eta})\Big). \tag{40}$$

To further lower bound (40), we note that $\alpha\tilde{\eta} > \log(1 + \alpha\tilde{\eta}) \geq \frac{1}{2}\alpha\tilde{\eta}$ since by Assumption 11, $0 < \alpha\tilde{\eta} < 1$. Furthermore, Assumption 11 guarantees that $\alpha < \epsilon/2 = (\delta - \delta \cdot (1.05 - \delta)^{1/2})/8$ and $2\alpha^3\tilde{\eta}^3 + 3\alpha^2\tilde{\eta}^2 \leq 4\alpha^2 < 1/5 \wedge 2\epsilon/5$, so we have that

$$\log\Big((1 + \alpha\tilde{\eta})^2 \cdot (1 - 2\alpha\tilde{\eta})\Big) = \log\big(1 - 3\alpha^2\tilde{\eta}^2 - 2\alpha^3\tilde{\eta}^3\big)$$

$$= -\log\left(1 + \frac{3\alpha^2\tilde{\eta}^2 + 2\alpha^3\tilde{\eta}^3}{1 - 3\alpha^2\tilde{\eta}^2 - 2\alpha^3\tilde{\eta}^3}\right)$$

$$\geq \frac{-3\alpha^2\tilde{\eta}^2 - 2\alpha^3\tilde{\eta}^3}{1 - 3\alpha^2\tilde{\eta}^2 - 2\alpha^3\tilde{\eta}^3}$$

$$\geq -5\alpha^2. \tag{41}$$

Putting (40) and (41) together, we have that

$$\sum_{t=t_0}^{t_1-1} \log\left(1 + \alpha\tilde{\eta}\left(1 - yf(\mathbf{x}; \mathbf{W}^{(t)})\right)\right) \geq \left(\alpha\tilde{\eta}\epsilon - 5\alpha^2\right)\cdot(t_1 - t_0) - \Delta(B)\cdot\log(1 + \alpha\tilde{\eta})$$

$$\geq \frac{1}{2}\alpha\tilde{\eta}\epsilon\cdot(t_1 - t_0) - \Delta(B)\alpha\tilde{\eta} \tag{42}$$

And consequently we have the following result, for all $r \in \mathcal{V}_{y,+}$,

$$\langle\mathbf{w}_{y,r}^{(t_1)}, y\mathbf{v}\rangle = \langle\mathbf{w}_{y,r}^{(t_0)}, y\mathbf{v}\rangle \cdot \prod_{t=t_0}^{t_1-1}\left(1 + \alpha\tilde{\eta}\left(1 - yf(\mathbf{x}; \mathbf{W}^{(t)})\right)\right)$$

$$= \langle\mathbf{w}_{y,r}^{(t_0)}, y\mathbf{v}\rangle \cdot \exp\left\{\sum_{t=t_0}^{t_1-1} \log\left(1 + \alpha\tilde{\eta}\left(1 - yf(\mathbf{x}; \mathbf{W}^{(t)})\right)\right)\right\}$$

$$\geq \langle\mathbf{w}_{y,r}^{(t_0)}, y\mathbf{v}\rangle \cdot \exp\left\{\frac{1}{2}\alpha\tilde{\eta}\epsilon\cdot(t_1 - t_0) - \Delta(B)\alpha\tilde{\eta}\right\},$$

where in the last inequality we use (42). Plugging in the expression of $\epsilon$, $\alpha$, $\tilde{\eta}$, and $\Delta(B)$, we have proved the second conclusion of Lemma 17. This concludes the proof of Lemma 17. $\quad\square$

## C.6 Proof of Technical Results

### C.6.1 Proof of Proposition 18

*Proof of Proposition 18.* We want to track the sequence (17) after it falls below $\delta$. To this end, we define two stopping times

$$T_{(\mathbf{v}),\delta,L} = \min\left\{t \geq T_{(\mathbf{v})} : \frac{1}{m}\sum_{r\in[m]}\sigma\left(\langle\mathbf{w}_{y,r}^{(t)}, y\mathbf{v}\rangle\right) < \delta\right\},$$

$$T_{(\mathbf{v})}^{+,2} = \min\left\{t \geq T_{(\mathbf{v}),\delta,L} : \frac{1}{m}\sum_{r\in[m]}\sigma\left(\langle\mathbf{w}_{y,r}^{(t)}, y\mathbf{v}\rangle\right) \geq \delta\right\}.$$

If $T_{(\mathbf{v}),\delta,L} = +\infty$, then the proof is over. Otherwise we prove that, before $T_{(\mathbf{v})}^{+,2} \leq +\infty$ (possibly equal), the sequence never falls below $\delta/2$.

Let's take a closer look at the controls over the negative parts while the weak signal remain learned. Note that for $t \in [T_{(\mathbf{v})}, T_{(\mathbf{v}),\delta,L}]$ and $r \in \mathcal{V}_{y,+}$, we have that

$$\langle\mathbf{w}_{y,r}^{(t)}, y\mathbf{v}\rangle = \langle\mathbf{w}_{y,r}^{(0)}, y\mathbf{v}\rangle \cdot \prod_{s=0}^{t-1}\left(1 + \frac{2\eta\|\mathbf{v}\|}{m}\cdot\left(1 - yf(\mathbf{x}; \mathbf{W}^{(s)})\right)\right)$$

$$\leq \langle\mathbf{w}_{y,r}^{(0)}, y\mathbf{v}\rangle \cdot \exp\left\{\frac{2\eta\|\mathbf{v}\|_2^2}{m}\cdot\sum_{s=0}^{t-1}\left(1 - yf(\mathbf{x}; \mathbf{W}^{(s)})\right)\right\}.$$

Meanwhile, for $t \in [T_{(\mathbf{v})}, T_{(\mathbf{v}),\delta,L}]$, we have that $\langle\mathbf{w}_{y,r}^{(t)}, y\mathbf{v}\rangle > (\beta_{\mathbf{v}}^* m\delta)^{1/2} \gg \langle\mathbf{w}_{y,r}^{(0)}, y\mathbf{v}\rangle$. Hence

$$\exp\left\{\frac{2\eta\|\mathbf{v}\|_2^2}{m}\cdot\sum_{s=0}^{t-1}\left(1 - yf(\mathbf{x}; \mathbf{W}^{(s)})\right)\right\} > 1, \quad \forall t \in [T_{(\mathbf{v})}, T_{(\mathbf{v}),\delta,L} - 1],$$

and in consequence we have that

$$\sum_{s=0}^{t-1} \left(1 - yf(\mathbf{x}; \mathbf{W}^{(s)})\right) > 0, \quad \forall t \in [T_{(\mathbf{v})}, T_{(\mathbf{v}),\delta,L} - 1]. \tag{43}$$

On the other hand, for $r \in \mathcal{V}_{-y,-}$, it holds that $\langle \mathbf{w}_{-y,r}^{(0)}, -y\mathbf{v}\rangle < 0$ and thus by (43) we have that

$$\begin{aligned}
\langle \mathbf{w}_{-y,r}^{(t)}, -y\mathbf{v}\rangle &= \langle \mathbf{w}_{-y,r}^{(0)}, -y\mathbf{v}\rangle \cdot \prod_{s=0}^{t-1} \left(1 - \frac{2\eta\|\mathbf{v}\|_2^2}{m} \cdot \left(1 - yf(\mathbf{x}; \mathbf{W}^{(s)})\right)\right) \\
&\geq \langle \mathbf{w}_{-y,r}^{(0)}, -y\mathbf{v}\rangle \cdot \exp\left\{-\frac{2\eta\|\mathbf{v}\|_2^2}{m} \cdot \sum_{s=0}^{t-1} \left(1 - yf(\mathbf{x}; \mathbf{W}^{(s)})\right)\right\} \\
&\geq \langle \mathbf{w}_{-y,r}^{(0)}, -y\mathbf{v}\rangle \geq -\sqrt{2\log(16m/p)} \cdot \sigma_0\|\mathbf{v}\|_2,
\end{aligned} \tag{44}$$

for all $t \in [T_{(\mathbf{v})}, T_{(\mathbf{v}),\delta,L}]$. Analogously, we also have that

$$\langle \mathbf{w}_{-y,r}^{(t)}, -y\mathbf{u}\rangle \geq \langle \mathbf{w}_{-y,r}^{(0)}, -y\mathbf{u}\rangle \geq -\sqrt{2\log(16m/p)} \cdot \sigma_0\|\mathbf{u}\|_2 \tag{45}$$

for all $t \in [T_{(\mathbf{v})}, T_{(\mathbf{v}),\delta,L}]$. Therefore, we have that for all $t \in [T_{(\mathbf{v})}, T_{(\mathbf{v}),\delta,L}]$,

$$\begin{aligned}
E_t &:= \frac{1}{m} \sum_{r \in [m]} \sigma\left(-\langle \mathbf{w}_{-y,r}^{(t)}, -y\mathbf{u}\rangle\right) + \sigma\left(-\langle \mathbf{w}_{-y,r}^{(t)}, -y\mathbf{v}\rangle\right) \\
&\leq 2 \cdot \max_{r \in [m]} \left\{\sigma\left(-\langle \mathbf{w}_{-y,r}^{(t)}, -y\mathbf{u}\rangle\right) \vee \sigma\left(-\langle \mathbf{w}_{-y,r}^{(t)}, -y\mathbf{v}\rangle\right)\right\} \\
&\leq 2\left(\sqrt{2\log(16m/p)} \cdot \sigma_0\|\mathbf{u}\|_2\right)^2 \\
&\ll \frac{\delta}{2}.
\end{aligned} \tag{46}$$

This allows us to leverage Proposition 19 to upper bound $yf(\mathbf{x}; \mathbf{W}^{(t)})$ for $t \in [T_{(\mathbf{v})}, T_{(\mathbf{v}),\delta,L}]$. Specifically, we locate the last step before $T_{(\mathbf{v}),\delta,L}$ when $yf$ just bounces up over 1, which is,

$$\bar{T}_{k^*} := \max\left\{\bar{T}_k : \bar{T}_k \leq T_{(\mathbf{v}),\delta,L}\right\},$$

where $\bar{T}_k$ is defined in (18). Then Proposition 19 with Inequality (46) implies that

$$\begin{aligned}
yf(\mathbf{x}; \mathbf{W}^{(\bar{T}_{k^*})}) &\leq yf(\mathbf{x}; \mathbf{W}^{(\bar{T}_{k^*}-1)}) \cdot \left(1 + \tilde{\eta}\left(1 - yf(\mathbf{x}; \mathbf{W}^{(\bar{T}_k-1)})\right)\right)^2 + 2E_{\bar{T}_{k^*}-1} \\
&\leq \left(1 + \tilde{\eta}\left(1 - yf(\mathbf{x}; \mathbf{W}^{(\bar{T}_{k^*}-1)})\right)\right)^2 + o(1) \\
&\leq \left(1 + \tilde{\eta}\left(1 - \frac{1}{2\tilde{\eta}}\left(\tilde{\eta} + 2 - \sqrt{\tilde{\eta}^2 + 4\tilde{\eta}}\right)\right)\right)^2 + o(1) \\
&\leq 3,
\end{aligned} \tag{47}$$

where we have applied the fact that $\tilde{\eta} < 1$. On the other hand, we have that

$$\frac{1}{m} \sum_{r \in [m]} \sigma\left(\langle \mathbf{w}_{y,r}^{(\bar{T}_{k^*}-1)}, y\mathbf{u}\rangle\right) \leq 1.05,$$

for which Lemma 15 indicates that $\max_{r \in [m]} \langle \mathbf{w}_{y,r}^{(\bar{T}_{k^*}-1)}, y\mathbf{u}\rangle \leq (1.05\beta_{\mathbf{u}}^* m)^{1/2}$. As in Inequality (23), one step gradient descent then gives that

$$\langle \mathbf{w}_{y,r}^{(\bar{T}_{k^*})}, y\mathbf{u}\rangle \leq \langle \mathbf{w}_{y,r}^{(\bar{T}_{k^*}-1)}, y\mathbf{u}\rangle \cdot \left(1 + \frac{2\eta\|\mathbf{u}\|_2^2}{m} \cdot \left(1 - yf(\mathbf{x}; \mathbf{W}^{(\bar{T}_{k^*}-1)})\right)\right) \leq 1.5 \cdot \left(1.05\beta_{\mathbf{u}}^* m\right)^{1/2}.$$

Now we consider the scale of these inner products right at $T_{(\mathbf{v}),\delta,L}$. From the definitions of $T_{(\mathbf{v}),\delta,L}$ and $\bar{T}_{k^*}$ we know that $yf_t > 1$ between these steps (otherwise the sequence wouldn't fall below $\delta$). Thus by (47),

$$1 < yf(\mathbf{x}; \mathbf{W}^{(t)}) < yf(\mathbf{x}; \mathbf{W}^{(\bar{T}_{k^*})}) \leq 3, \quad \forall t \in [\bar{T}_{k^*}, T_{(\mathbf{v}),\delta,L}].$$

We first state that $\frac{1}{m} \sum_{r \in [m]} \sigma(\langle \mathbf{w}_{y,r}^{(T_{(\mathbf{v}),\delta,L})}, y\mathbf{v} \rangle)$ is not far away from $\delta$. Note that by (43),

$$\frac{1}{m} \sum_{r \in [m]} \sigma\big(\langle \mathbf{w}_{y,r}^{(T_{(\mathbf{v}),\delta,L})}, y\mathbf{v} \rangle\big) = \frac{1}{m} \sum_{r \in [m]} \sigma\big(\langle \mathbf{w}_{y,r}^{(T_{(\mathbf{v}),\delta,L}-1)}, y\mathbf{v} \rangle\big) \cdot \left(1 + \frac{2\eta\|\mathbf{v}\|_2^2}{m} \cdot \big(1 - yf(\mathbf{x}; \mathbf{W}^{(T_{(\mathbf{v}),\delta,L}-1)})\big)\right)$$

$$\geq \delta \cdot \left(1 + \frac{2\eta\|\mathbf{v}\|_2^2}{m} \cdot (1 - 3)\right)$$

$$\geq \delta \cdot \left(1 - 2 \cdot \frac{\|\mathbf{v}\|_2^2}{\|\mathbf{u}\|_2^2}\right)$$

$$\geq \frac{3}{4}\delta. \tag{48}$$

Last inequality uses the fact that $\|\mathbf{v}\|_2^2/\|\mathbf{u}\|_2^2 \leq 1/4$. This shows that the sequence does not fall below $3\delta/4$ at the step $T_{(\mathbf{v}),\delta,L}$. In the sequel, we study the behavior of the sequence after step $T_{(\mathbf{v}),\delta,L}$.

Firstly, by Inequalities (44) and (45) with $t = T_{(\mathbf{v}),\delta,L}$, we obtain that the negative parts satisfy

$$\min_{r \in [m]} \langle \mathbf{w}_{-y,r}^{(T_{(\mathbf{v}),\delta,L})}, -y\mathbf{u} \rangle \geq -\sqrt{2\log(16m/p)} \cdot \sigma_0 \|\mathbf{u}\|_2,$$

$$\min_{r \in [m]} \langle \mathbf{w}_{-y,r}^{(T_{(\mathbf{v}),\delta,L})}, -y\mathbf{v} \rangle \geq -\sqrt{2\log(16m/p)} \cdot \sigma_0 \|\mathbf{v}\|_2.$$

which by definition of $E_t$ in (46) implies that

$$E_{T_{(\mathbf{v}),\delta,L}} \leq 2 \left(\sqrt{2\log(16m/p)} \cdot \sigma_0\|\mathbf{u}\|_2\right)^2 \ll 1. \tag{49}$$

For the positive part, we have that

$$\max_{r \in [m]} \langle \mathbf{w}_{y,r}^{(T_{(\mathbf{v}),\delta,L})}, y\mathbf{u} \rangle < \max_{r \in [m]} \langle \mathbf{w}_{y,r}^{(\bar{T}_{k^*})}, y\mathbf{u} \rangle < 1.5 \cdot (1.05\beta_{\mathbf{u}}^* m)^{1/2}, \tag{50}$$

and by the same argument as Inequality (25) we can obtain the lower bound of $\max_{r \in [m]} \langle \mathbf{w}_{y,r}^{(T_{(\mathbf{v}),\delta,L})}, y\mathbf{u} \rangle$,

$$\max_{r \in [m]} \langle \mathbf{w}_{y,r}^{(T_{(\mathbf{v}),\delta,L})}, y\mathbf{u} \rangle \geq \max_{r \in [m]} \langle \mathbf{w}_{y,r}^{(T_{(\mathbf{v}),\delta,L}-1)}, y\mathbf{u} \rangle \left(1 + \frac{2\eta\|\mathbf{u}\|_2^2}{m} \big(1 - yf(\mathbf{x}; \mathbf{W}^{(T_{(\mathbf{v}),\delta,L}-1)})\big)\right)$$

$$\geq 0.2 \cdot \big(\beta_{\mathbf{u}}^* m\big)^{1/2}, \tag{51}$$

and finally for $t \in [T_{(\mathbf{v}),\delta,L}, T_{(\mathbf{v})}^{+,2}]$, we have that

$$\frac{1}{m} \sum_r \sigma(\langle \mathbf{w}_{y,r}^{(t)}, y\mathbf{v} \rangle) < \delta. \tag{52}$$

With all these initial conditions (49), (50), (51), and (52), one can then consider all the steps $\bar{T}_{k'}$, $\bar{T}_{k'} > T_{(\mathbf{v}),\delta,L}$ and apply an inductive argument exactly the same as in the proof of Lemma 16 to conclude that for all $t \in [T_{(\mathbf{v}),\delta,L}, T_{(\mathbf{v})}^{+,2}]$ it holds that

$$0.2 \cdot (\beta_{\mathbf{u}}^* m)^{1/2} \leq \max_r \langle \mathbf{w}_{y,r}^{(t)}, y\mathbf{u} \rangle \leq 1.5 \cdot (1.05\beta_{\mathbf{u}}^* m)^{1/2},$$

$$\min_r \langle -\mathbf{w}_{-y,r}^{(t)}, -y\mathbf{u} \rangle > -\sqrt{2\log(16m/p)} \cdot \sigma_0 \|\mathbf{u}\|_2,$$

$$\min_r \langle -\mathbf{w}_{-y,r}^{(t)}, -y\mathbf{v} \rangle > -\sqrt{2\log(16m/p)} \cdot \sigma_0 \|\mathbf{v}\|_2,$$

$$0 \leq yf(\mathbf{x}; \mathbf{W}^{(t)}) \leq 3,$$

and the sign stability is also true throughout $t \in [T_{(\mathbf{v}),\delta,L}, T_{(\mathbf{v})}^{+,2}]$. Therefore, Lemma 17 and (48) implies that for any $t \in [T_{(\mathbf{v}),\delta,L}, T_{(\mathbf{v})}^{+,2}]$, it holds that

$$\frac{1}{m} \sum_r \sigma\big(\langle \mathbf{w}_{y,v}^{(t)}, y\mathbf{v} \rangle\big) \geq \frac{1}{m} \sum_r \sigma\big(\langle \mathbf{w}_{y,v}^{(T_{(\mathbf{v}),\delta,L})}, y\mathbf{v} \rangle\big) \cdot \exp\left\{-2 \cdot \frac{\|\mathbf{v}\|_2^2}{\|\mathbf{u}\|_2^2} \cdot \frac{1.5\sqrt{1.05}}{\sqrt{1.05 - \delta}}\right\} \geq \frac{1}{2}\delta.$$

In conclusion, we have obtained that

$$\frac{1}{m} \sum_{r \in [m]} \sigma\big(\langle \mathbf{w}_{y,r}, y\mathbf{v} \rangle\big) \geq \delta/2, \quad \forall t \in [T_{(\mathbf{v}),\delta,L}, T_{(\mathbf{v})}^{+,2}].$$

If $T_{(\mathbf{v})}^{+,2} \neq +\infty$, repeat the above argument and we can finish the proof of Proposition 18. $\qquad \square$

### C.6.2 PROOF OF PROPOSITION 19

*Proof of Proposition 19.* We continue with the notation $\tilde{\eta} = 2\eta \|\mathbf{u}\|_2^2/m$. Define the function

$$h_{\tilde{\eta}}(z) := \big(1 + \tilde{\eta}(1 - z)\big)^2 \cdot z.$$

Note that $\|\mathbf{v}\|_2^2 \leq \|\mathbf{u}\|_2^2$ and that

$$\frac{2\eta\|\mathbf{u}\|_2^2}{m} \cdot \big(1 - yf(\mathbf{x}; \mathbf{W}^{(\bar{T}_k - 1)})\big) < \frac{2\eta\|\mathbf{u}\|_2^2}{m} < 1,$$

from the definition of $\bar{T}_k - 1$. Thus we have the following,

$$yf(\mathbf{x}, y; \mathbf{W}^{(\bar{T}_k)})$$

$$= \frac{1}{m} \sum_{r \in [m]} \sigma\big(\langle \mathbf{w}_{y,r}^{(\bar{T}_k - 1)}, y\mathbf{u} \rangle\big) \cdot \left(1 + \frac{2\eta\|\mathbf{u}\|_2^2}{m} \cdot \big(1 - yf(\mathbf{x}; \mathbf{W}^{(\bar{T}_k - 1)})\big)\right)^2$$

$$+ \frac{1}{m} \sum_{r \in [m]} \sigma\big(\langle \mathbf{w}_{y,r}^{(\bar{T}_k - 1)}, y\mathbf{v} \rangle\big) \cdot \left(1 + \frac{2\eta\|\mathbf{v}\|_2^2}{m} \cdot \big(1 - yf(\mathbf{x}; \mathbf{W}^{(\bar{T}_k - 1)})\big)\right)^2$$

$$- \frac{1}{m} \sum_{r \in [m]} \sigma\big(-\langle \mathbf{w}_{-y,r}^{(\bar{T}_k - 1)}, -y\mathbf{u} \rangle\big) \cdot \left(1 - \frac{2\eta\|\mathbf{u}\|_2^2}{m} \cdot \big(1 - yf(\mathbf{x}; \mathbf{W}^{(\bar{T}_k - 1)})\big)\right)^2$$

$$- \frac{1}{m} \sum_{r \in [m]} \sigma\big(-\langle \mathbf{w}_{-y,r}^{(\bar{T}_k - 1)}, -y\mathbf{v} \rangle\big) \cdot \left(1 - \frac{2\eta\|\mathbf{v}\|_2^2}{m} \cdot \big(1 - yf(\mathbf{x}; \mathbf{W}^{(\bar{T}_k - 1)})\big)\right)^2$$

$$\leq \frac{1}{m} \sum_{r \in [m]} \left(\sigma\big(\langle \mathbf{w}_{y,r}^{(\bar{T}_k - 1)}, y\mathbf{u} \rangle\big) + \sigma\big(\langle \mathbf{w}_{y,r}^{(\bar{T}_k - 1)}, y\mathbf{v} \rangle\big)\right) \cdot \left(1 + \frac{2\eta\|\mathbf{u}\|_2^2}{m} \cdot \big(1 - yf(\mathbf{x}; \mathbf{W}^{(\bar{T}_k - 1)})\big)\right)^2$$

$$- \frac{1}{m} \sum_{r \in [m]} \left(\sigma\big(-\langle \mathbf{w}_{-y,r}^{(\bar{T}_k - 1)}, -y\mathbf{u} \rangle\big) + \sigma\big(-\langle \mathbf{w}_{-y,r}^{(\bar{T}_k - 1)}, -y\mathbf{v} \rangle\big)\right)$$

$$\cdot \left(1 - \frac{2\eta\|\mathbf{u}\|_2^2}{m} \cdot \big(1 - yf(\mathbf{x}; \mathbf{W}^{(\bar{T}_k - 1)})\big)\right)^2$$

$$= yf(\mathbf{x}; \mathbf{W}^{(\bar{T}_k - 1)}) \cdot \left(1 + \tilde{\eta}\big(1 - yf(\mathbf{x}; \mathbf{W}^{(\bar{T}_k - 1)})\big)\right)^2$$

$$+ \frac{1}{m} \sum_{r \in [m]} \left(\sigma\big(-\langle \mathbf{w}_{-y,r}^{(\bar{T}_k - 1)}, -y\mathbf{u} \rangle\big) + \sigma\big(-\langle \mathbf{w}_{-y,r}^{(\bar{T}_k - 1)}, -y\mathbf{v} \rangle\big)\right) \cdot \underbrace{\left(2\tilde{\eta}\big(1 - yf(\mathbf{x}; \mathbf{W}^{(\bar{T}_k - 1)})\big)\right)^2}_{<2\tilde{\eta}<2}$$

$$\leq yf(\mathbf{x}; \mathbf{W}^{(\bar{T}_k - 1)}) \cdot \left(1 + \tilde{\eta}\big(1 - yf(\mathbf{x}; \mathbf{W}^{(\bar{T}_k - 1)})\big)\right)^2 + 2E_{\bar{T}_k - 1}. \tag{53}$$

Thus by (53) we have proved the second conclusion of Proposition 19. In the following, we prove the first conclusion of Proposition 19.

By Assumption 12, we know that $yf(\mathbf{x}; \mathbf{W}^{(\bar{T}_k)}) > 1 + \delta$. The definition of $T_{(\mathbf{v})}$ along with with Inequality (53) imply that

$$h_{\tilde{\eta}}\big(f(\mathbf{x}; \mathbf{W}^{(\bar{T}_k - 1)})\big) = yf(\mathbf{x}; \mathbf{W}^{(\bar{T}_k - 1)}) \cdot \left(1 + \tilde{\eta}\big(1 - yf(\mathbf{x}; \mathbf{W}^{(\bar{T}_k - 1)})\big)\right)^2 \geq 1 + \delta - \delta = 1 \tag{54}$$

Now it suffices to consider the equation $h_{\tilde{\eta}} - 1 = 0$, and one can easily verify that it has three roots

$$z_1 = 1,$$

$$z_2 = \frac{\tilde{\eta} + 2 - \sqrt{\tilde{\eta}^2 + 4\tilde{\eta}}}{2\tilde{\eta}},$$

$$z_3 = \frac{\tilde{\eta} + 2 + \sqrt{\tilde{\eta}^2 + 4\tilde{\eta}}}{2\tilde{\eta}} > \frac{2\tilde{\eta} + 2}{2\tilde{\eta}} > 1.$$

And the second root $z_2 < 1$ if and only if $\tilde{\eta} > 1/2$.

Now if $0 < \tilde{\eta} \le 1/2$, then $z_1 \le z_2 < z_3$, and then $h(z) < 1$ for $z < z_1 = 1$. Therefore Inequality (54) implies that $yf(\mathbf{x}; \mathbf{W}^{(\bar{T}_k-1)}) \ge z_1 = 1$, and recursively implies that $yf(\mathbf{x}; \mathbf{W}^{(0)}) \ge 1$, which contradicts with the fact that $yf(\mathbf{x}; \mathbf{W}^{(0)}) \le 2\sqrt{2\log(16m/p)} \cdot \sigma_0 \|\mathbf{u}\|_2$. So in order for $\bar{T}_1 < +\infty$, we must have $\tilde{\eta} > 1/2$. In conclusion, one necessary condition for the stable oscillation Assumption 12 is that $\tilde{\eta} > 1/2$, and therefore

$$yf(\mathbf{x}; \mathbf{W}^{(\bar{T}-1)}) \ge z_2 = \frac{\tilde{\eta} + 2 - \sqrt{\tilde{\eta}^2 + 4\tilde{\eta}}}{2\tilde{\eta}}.$$

This finishes the proof of Proposition 19. □

## C.7 DISCUSSION: NECESSARY CONDITION FOR $\delta$-OSCILLATION

Inequality (24) provides an upper bound involving $\tilde{\eta}$, which should be compatible with Assumption 12, hence

$$\left(\tilde{\eta}/2 + \sqrt{\tilde{\eta}^2/4 + \tilde{\eta}}\right)^2 > 1 + \delta \quad \Leftrightarrow \quad \tilde{\eta} > (1 + \delta^{-1})\left(\sqrt{1 + \delta} - 1\right).$$

One can verify with software that RHS is monotonically increasing in $\delta \in [0, 1]$ with minimal value $0.5$, when $\tilde{\eta} = 0$, which is in line with the weakest oscillation condition discovered in the last part. And the maximal value taken at $\delta = 1$ is less than $0.83$.

On the other hand, Inequality (21) should also be compatible with the Assumption 4, which indicates

$$1 - \delta > yf(\mathbf{x}; \mathbf{W}^{(\bar{T}_k-1)})) > \frac{2 + \tilde{\eta} - \sqrt{\tilde{\eta}^2 + 4\tilde{\eta}}}{2\tilde{\eta}}.$$

The readers can see that it is equivalent to $\tilde{\eta} > \delta^{-1}((1 - \delta)^{-1/2} - 1)$. Furthermore, we have that $\delta^{-1}((1 - \delta)^{-1/2} - 1) > (1 + \delta^{-1})(\sqrt{1 + \delta} - 1)$ thus it is a stronger requirement on $\eta$.

## D SINGLE TRAINING DATA CASE: SMALL LEARNING RATE REGIME

This section focuses on training our model with single noiseless data point $(\mathbf{x}, y)$, where $\mathbf{x} = (y\mathbf{u}, y\mathbf{v})$ contains two signal patches with $\mathbf{u}$ much stronger than $\mathbf{v}$. Therefore, the whole objective can be rearranged by

$$L(\mathbf{W}) = \frac{1}{2}\left(f(\mathbf{x}; \mathbf{W}) - y\right)^2 = \frac{1}{2}\left(\sum_{j \in \{\pm 1\}} \frac{j}{m} \sum_{r \in [m]} \sigma(\langle \mathbf{w}_{j,r}, y\mathbf{u}\rangle) + \sigma(\langle \mathbf{w}_{j,r}, y\mathbf{v}\rangle) - y\right)^2.$$

In this simplified setting, each weight vector is updated by

$$\mathbf{w}_{j,r}^{(t+1)} = \mathbf{w}_{j,r}^{(t)} - \frac{jy\eta}{m} \cdot \left(f(\mathbf{x}; \mathbf{W}^{(t)}) - y\right) \cdot \left(\sigma'(\langle \mathbf{w}_{j,r}^{(t)}, y\mathbf{u}\rangle)\mathbf{u} + \sigma'(\langle \mathbf{w}_{j,r}^{(t)}, y\mathbf{v}\rangle)\mathbf{v}\right).$$

Then we can directly obtain the following updating rules of the inner products,

$$\langle \mathbf{w}_{j,r}^{(t+1)}, \mathbf{u}\rangle = \langle \mathbf{w}_{j,r}^{(t)}, \mathbf{u}\rangle - \frac{jy\eta}{m} \cdot \left(f(\mathbf{x}; \mathbf{W}^{(t)}) - y\right) \cdot \sigma'(\langle \mathbf{w}_{j,r}^{(t)}, y\mathbf{u}\rangle) \cdot \|\mathbf{u}\|_2^2,$$

$$\langle \mathbf{w}_{j,r}^{(t+1)}, \mathbf{v}\rangle = \langle \mathbf{w}_{j,r}^{(t)}, \mathbf{u}\rangle - \frac{jy\eta}{m} \cdot \left(f(\mathbf{x}; \mathbf{W}^{(t)}) - y\right) \cdot \sigma'(\langle \mathbf{w}_{j,r}^{(t)}, y\mathbf{v}\rangle) \cdot \|\mathbf{v}\|_2^2,$$

since $\mathbf{u}$ and $\mathbf{v}$ are assumed to be orthogonal in our data generation model. In this section, we also denote the fitting residual at iteration $t$ as $\ell^{(t)} = f(\mathbf{x}; \mathbf{W}^{(t)}) - y$ for convenience.

To better prepare for the analysis of this section, we single out the following concentration results from Appendix B which provides a high-probability bound on the initialization $\mathbf{W}^{(0)}$.

**Lemma 21** (Initialization). *Suppose that $d = \Omega(\log(m/p))$ and $m = \Omega(\log(1/p))$. Then with probability at least $1 - p$, there holds*

$$\sigma_0 \|\mathbf{u}\|/2 \leq \max_{r \in [m]} \langle \mathbf{w}_{j,r}^{(t)}, j\mathbf{u} \rangle \leq \sqrt{2\log(16m/p)}\sigma_0\|\mathbf{u}\|,$$

$$\sigma_0 \|\mathbf{v}\|/2 \leq \max_{r \in [m]} \langle \mathbf{w}_{j,r}^{(t)}, j\mathbf{v} \rangle \leq \sqrt{2\log(16m/p)}\sigma_0\|\mathbf{v}\|,$$

*for all $j \in \{\pm 1\}$.*

The result of this section relies on the following conditions on the data model and the initialization.

**Assumption 22** (Conditions on hyperparameters). *Suppose that the following holds:*

1. *The weight initialization scale $\sigma_0 = \widetilde{\Theta}(\|\mathbf{u}\|_2^{-1})$;*

2. *The signal strength $\|\mathbf{u}\|_2 > \widetilde{\Omega}(m^2) \cdot \|\mathbf{v}\|_2$;*

3. *The dimension $d$ satisfies $d = \Omega(\mathrm{polylog}(m))$.*

**Theorem 23** (Restatement of Proposition 1). *Under Assumption 22, choosing the learning rate $\eta \leq m/6\|\mathbf{u}\|_2^2$ small enough and $\epsilon = 0.01$, then with probability at least $1 - p = 1 - 1/\mathrm{poly}(d)$, there exist*

$$T^\dagger = \frac{m}{\eta(1-\tau)\|\mathbf{u}\|_2^2} \log\left(\frac{2\iota}{\sigma_0\|\mathbf{u}\|_2}\right), \quad T = T^\dagger + \left\lfloor \frac{Cm^3}{2\eta\epsilon\|\mathbf{u}\|^2} \right\rfloor,$$

*with $\tau$, $\iota$ defined in (55), (56), such that: (i) the average loss over iterations $[T^\dagger, T]$ decreased to $2\epsilon$, i.e.*

$$\frac{1}{T - T^\dagger + 1} \sum_{s=T^\dagger}^{T} L(\mathbf{W}^{(s)}) \leq 2\epsilon,$$

*(ii) the model does not learn weak signal $\mathbf{v}$ well enough, compared to initialization, i.e.*

$$\max_{j \in \{\pm 1\}, r \in [m]} \left| \langle \mathbf{w}_{j,r}^{(t)}, \mathbf{v} \rangle \right| \leq 2\sqrt{2\log(16m/p)} \cdot \sigma_0\|\mathbf{v}\|_2.$$

In the small learning rate regime, the dynamics go through two stages, in which the strong signal $\mathbf{u}$ will be firstly learned exponentially fast, and subsequently fully fit the given data point $(\mathbf{x}, y)$ therefore stabilizing the training process. The following lemma plays an important role in the exponentially increasing stage, for which we single it out here.

**Lemma 24** (Derivative lower bound). *For any $0 < \tau < 1$ to be tuned later, suppose at some time $t$ there holds*

$$\max_{r \in [m], j \in \{\pm 1\}} \left\{ \left| \langle \mathbf{w}_{j,r}^{(t)}, \mathbf{u} \rangle \right|, \left| \langle \mathbf{w}_{j,r}^{(t)}, \mathbf{v} \rangle \right| \right\} \leq \sqrt{\frac{\tau}{2}},$$

*then we can lower bound the fitting residual by $-y\ell^{(t)} \geq 1 - \tau$.*

*Proof of Lemma 24.* Plug into the CNN model definition (4), we have that

$$-y\ell^{(t)} = 1 - F_y(\mathbf{x}; \mathbf{W}^{(t)}) + F_{-y}(\mathbf{x}; \mathbf{W}^{(t)}) \geq 1 - F_y(\mathbf{x}; \mathbf{W}^{(t)}).$$

We can upper bound $F_y(\mathbf{x}; \mathbf{W}^{(t)})$ further by

$$F_y(\mathbf{x}; \mathbf{W}^{(t)}) = \frac{1}{m} \sum_{r \in [m]} \sigma(\langle \mathbf{w}_{y,r}^{(t)}, y\mathbf{u} \rangle) + \sigma(\langle \mathbf{w}_{y,r}^{(t)}, y\mathbf{v} \rangle)$$

$$\leq \max_{r \in [m]} \left\{ \langle \mathbf{w}_{y,r}^{(t)}, y\mathbf{u} \rangle^2 + \langle \mathbf{w}_{y,r}^{(t)}, y\mathbf{v} \rangle^2 \right\}$$

$$\leq \tau.$$

Then it follows that $-y\ell^{(t)} \geq 1 - \tau$. $\qquad \square$

### D.1 STAGE 1. EXPONENTIAL GROWTH

We will mainly track the maximal inner product between $\mathbf{w}$ and the signal vectors $\mathbf{v}$ and $\mathbf{u}$, i.e.,

$$\Psi^{(t)} = \max_{j,r} \left| \langle \mathbf{w}_{j,r}^{(t)}, \mathbf{v} \rangle \right|, \quad \Phi^{(t)} = \max_{j,r} \left| \langle \mathbf{w}_{j,r}^{(t)}, \mathbf{u} \rangle \right|.$$

In the following, we would take

$$\tau = \max \left\{ 2\sigma_0 \|\mathbf{u}\|_2 \left( 2\log(16m/p) \right)^{1/2 - \|\mathbf{v}\|_2^2/4\|\mathbf{u}\|_2^2}, 1 - \frac{6\sqrt{2}\|\mathbf{v}\|_2^2}{\|\mathbf{u}\|_2^2 \log(2/\sqrt{2\log(16m/p)})} \right\} \tag{55}$$

$$\iota = \sigma_0 \|\mathbf{u}\|_2 \cdot \exp \left\{ \frac{1-\tau}{6} \log \left( \frac{\sqrt{\tau/2}}{\sigma_0 \|\mathbf{u}\|_2 \sqrt{2\log(16m/p)}} \right) \right\}. \tag{56}$$

By the conditions in Assumption 22 on $\|\mathbf{v}\|_2^2/\|\mathbf{u}\|_2^2$, we find $\tau, \iota$ both constants in $(0, 1)$.

**Lemma 25** (First stage: one training data case). *Under the same conditions as Theorem 23, there exists time*

$$T^\dagger = \frac{m}{\eta(1-\tau)\|\mathbf{u}\|_2^2} \log \left( \frac{2\iota}{\sigma_0 \|\mathbf{u}\|_2} \right),$$

*such that: (i) the model learns the strong signal to a constant level, i.e.,*

$$\max_{r \in [m]} \langle \mathbf{w}_{y,r}^{(T^\dagger)}, y\mathbf{u} \rangle \geq \iota,$$

*(ii) compared to the random initialization, the model does not learn weak signal that much, i.e.,*

$$\max_{j \in \{\pm 1\}, r \in [m]} \left| \langle \mathbf{w}_{j,r}^{(T^\dagger)}, \mathbf{v} \rangle \right| \leq 2\sqrt{2\log(16m/p)} \cdot \sigma_0 \|\mathbf{v}\|_2.$$

*Proof of Lemma 25.* Firstly, we would find $\{\Psi^{(t)}, \Phi^{(t)}\}_{t \geq 0}$ having an exponentially growing upper bound. Recursively, we would have that

$$\Psi^{(t+1)} \leq \Psi^{(t)} + \max_{j \in \{\pm 1\}, r \in [m]} \left| \frac{jy\eta}{m} \cdot \left( f(\mathbf{x}; \mathbf{W}^{(t)}) - y \right) \cdot \sigma'(\langle \mathbf{w}_{j,r}^{(t)}, y\mathbf{v} \rangle) \cdot \|\mathbf{v}\|_2^2 \right|$$

$$= \Psi^{(t)} + \frac{\eta}{m} \cdot \left| \ell^{(t)} \right| \cdot \|\mathbf{v}\|_2^2 \cdot \max_{j \in \{\pm 1\}, r \in [m]} \sigma'(\langle \mathbf{w}_{j,r}^{(t)}, y\mathbf{v} \rangle)$$

$$\leq \Psi^{(t)} + \frac{2\eta}{m} \cdot \left| \ell^{(t)} \right| \cdot \|\mathbf{v}\|_2^2 \cdot \Psi^{(t)}$$

$$\leq \exp \left( \frac{6\eta\|\mathbf{v}\|_2^2}{m} \right) \cdot \Psi^{(t)}.$$

Therefore, we have that

$$\Psi^{(t)} \leq \exp \left( \frac{6\eta\|\mathbf{v}\|_2^2 t}{m} \right) \cdot \Psi^{(0)} \leq \exp \left( \frac{6\eta\|\mathbf{v}\|_2^2 t}{m} \right) \cdot \sqrt{2\log(16m/p)} \cdot \sigma_0 \|\mathbf{v}\|_2. \tag{57}$$

It follows by the same argument that that

$$\Phi^{(t)} \leq \exp \left( \frac{6\eta\|\mathbf{u}\|_2^2 t}{m} \right) \Phi^{(0)} \leq \exp \left( \frac{6\eta\|\mathbf{u}\|_2^2 t}{m} \right) \cdot \sqrt{2\log(16m/p)} \cdot \sigma_0 \|\mathbf{u}\|_2. \tag{58}$$

Note that the growing rates of these two bounds differ a lot due to the different magnitudes of $\|\mathbf{u}\|_2$ and $\|\mathbf{v}\|_2$. Our subsequent analysis illustrates that $\Phi^{(t)}$ can grow into a constant-level magnitude since the strong signal $\mathbf{u}$ is significant enough. Now we can track how well our model learns $\mathbf{u}$ by

$$A^{(t)} = \max_{r \in [m]} \langle \mathbf{w}_{y,r}^{(t)}, y\mathbf{u} \rangle.$$

By the definition, $A^{(t)} \leq \Phi^{(t)}$ also admits an exponentially growing upper bound. For a certain $\tau \in (0, 1)$, due to the previous upper bounds (57) and (58), $\max\{\Phi^{(t)}, \Psi^{(t)}\} \leq \sqrt{\tau/2}$ remains true at least until

$$T_1 = \frac{m}{6\eta\|\mathbf{u}\|^2} \log\left(\frac{\sqrt{\tau/2}}{\sigma_0\|\mathbf{u}\|_2\sqrt{2\log(16m/p)}}\right). \tag{59}$$

Consequently, until at least $T_1$, we can use Lemma 24 to conclude that $-y\ell^{(t)} \geq 1 - \tau$, which enables lower bounding $A^{(t)}$. Specifically, start with the updating rule

$$\langle \mathbf{w}_{y,r}^{(t+1)}, y\mathbf{u}\rangle = \langle \mathbf{w}_{y,r}^{(t)}, y\mathbf{u}\rangle + \frac{\eta}{m} \cdot \left(-y\ell^{(t)}\right) \cdot \sigma'(\langle \mathbf{w}_{y,r}^{(t)}, y\mathbf{u}\rangle) \cdot \|\mathbf{u}\|_2^2$$

$$\geq \langle \mathbf{w}_{y,r}^{(t)}, y\mathbf{u}\rangle + \frac{2\eta(1-\tau)\|\mathbf{u}\|_2^2}{m} \cdot \max_{r\in[m]}\left\{\langle \mathbf{w}_{y,r}^{(t)}, y\mathbf{u}\rangle, 0\right\},$$

and take maximum over $r \in [m]$ to see that

$$A^{(t+1)} \geq A^{(t)} + \frac{2\eta(1-\tau)\|\mathbf{u}\|_2^2}{m} \cdot A^{(t)} \geq \exp\left(\frac{\eta(1-\tau)\|\mathbf{u}\|_2^2}{m}\right) \cdot A^{(t)},$$

where the last equality is by $1 + z \geq \exp(z/2)$ for any $0 \leq z \leq 2$. Consequently, we would have

$$A^{(t)} \geq \exp\left(\frac{\eta(1-\tau)\|\mathbf{u}\|_2^2 t}{m}\right) \cdot A^{(0)} \geq \exp\left(\frac{\eta(1-\tau)\|\mathbf{u}\|_2^2 t}{m}\right) \cdot \sigma_0\|\mathbf{u}\|_2/2, \tag{60}$$

at least until $t \leq T_1$ defined in (59). Then we define another time

$$T_2 = \frac{m}{\eta(1-\tau)\|\mathbf{u}\|_2^2} \log\left(\frac{2\iota}{\sigma_0\|\mathbf{u}\|_2}\right) \leq T_1,$$

where the inequality is due to the scaling of $\iota$ upon $\tau$. Plugging $T_2$ into the exponential lower bound (60), we can conclude that

$$\Phi^{(T_2)} \geq A^{(T_2)} \geq \iota,$$

which already grows up to a constant level magnitude by the time $T_2$. Lastly, we plug the definition of $T_2$ to upper bound $\Psi^{(T_2)}$ as

$$\Psi^{(T_2)} \leq \exp\left(\frac{6\|\mathbf{v}\|_2^2}{(1-\tau)\|\mathbf{u}\|_2^2} \log\left(\frac{2\iota}{\sigma_0\|\mathbf{u}\|_2}\right)\right) \cdot \sqrt{2\log(16m/p)} \cdot \sigma_0\|\mathbf{v}\|_2 \leq 2\sqrt{2\log(16m/p)} \cdot \sigma_0\|\mathbf{v}\|_2.$$

In conclusion, by taking $T^\dagger = T_2$, this lemma is completely proved. $\qquad\square$

## D.2 STAGE 2. STABILIZED CONVERGENCE

In the second stage, our lemmas would suggest that before the model really learns the weak signal $\mathbf{v}$ (i.e. before $\max_{j,r} |\langle \mathbf{w}_{j,r}^{(t)}, \mathbf{v}\rangle|$ breaks the $\widetilde{\mathcal{O}}(\sigma_0\|\mathbf{v}\|_2)$ upper bound), the model already fits the given data point by exploiting the strong signal $\mathbf{u}$ and decreasing the loss to $\epsilon$.

**Lemma 26** (Second stage: one training data case). *Under the same conditions as Theorem 23, for any $\epsilon = 0.01$, there exists time*

$$T = T^\dagger + \left\lfloor \frac{Cm^3}{2\eta\epsilon\|\mathbf{u}\|_2^2} \right\rfloor$$

*such that: (i) the average loss over iterations within this stage has decreased to $2\epsilon$, i.e.,*

$$\frac{1}{T - T^\dagger + 1} \sum_{s=T^\dagger}^{T} L(\mathbf{W}^{(s)}) \leq 2\epsilon,$$

*(ii) throughout the training dynamics $0 \leq t \leq T$, there holds*

$$\max_{j\in\{\pm 1\}, r\in[m]} |\langle \mathbf{w}_{j,r}^{(t)}, \mathbf{v}\rangle| \leq 2\sqrt{2\log(16m/p)}\sigma_0\|\mathbf{v}\|_2.$$

The proof of Lemma 26 relies on the following three lemmas (Lemmas 27, 28, and 29). We present these lemmas with proofs and then combine them to give the proof of Lemma 26.

Firstly, we identify when the upper bound on $\langle \mathbf{w}_{j,r}^{(t)}, \mathbf{v} \rangle$ breaks and find that the conclusions of Lemma 25 still holds before that time.

**Lemma 27.** *Under the same conditions in Theorem 23, take $\eta \leq m/6\|\mathbf{u}\|_2^2$. There exists a time*

$$T^{\ddagger} = \frac{m}{6\eta\|\mathbf{v}\|_2^2} \log(2) \geq T^{\dagger}$$

*such that*

$$\max_{r \in [m]} \langle \mathbf{w}_{y,r}^{(t)}, y\mathbf{u} \rangle \geq \iota/2, \quad \max_{j \in \{\pm 1\}, r \in [m]} \left| \langle \mathbf{w}_{j,r}^{(t)}, \mathbf{v} \rangle \right| \leq 2\sqrt{2 \log(16m/p)} \cdot \sigma_0 \|\mathbf{v}\|_2.$$

*hold for any $T^{\dagger} \leq t \leq T^{\ddagger}$.*

*Proof of Lemma 27.* Firstly, we need to adopt the exponential upper bound derived in proving Lemma 25,

$$\Psi^{(t)} \leq \exp\left(\frac{6\eta\|\mathbf{v}\|_2^2 t}{m}\right) \cdot \Psi^{(0)} \leq \exp\left(\frac{6\eta\|\mathbf{v}\|_2^2 t}{m}\right) \cdot \sqrt{2 \log(16m/p)} \cdot \sigma_0 \|\mathbf{v}\|_2.$$

Then we naturally find that before $T^{\ddagger}$, it would always hold that

$$\max_{j \in \{\pm 1\}, r \in [m]} \left| \langle \mathbf{w}_{j,r}^{(t)}, \mathbf{v} \rangle \right| \leq 2\sqrt{2 \log(16m/p)} \cdot \sigma_0 \|\mathbf{v}\|_2.$$

Due to the conditions on $\|\mathbf{u}\|_2^2/\|\mathbf{v}\|_2^2$, $T^{\ddagger}$ is found to be much larger than $T^{\dagger}$. Then we proceed by induction to prove the other assertion. At time $t = T^{\dagger}$, the lower bound $\max_r \langle \mathbf{w}_{y,r}^{(t)}, y\mathbf{u} \rangle \geq \iota/2$ holds as a consequence of the previous lemma. Suppose it holds until time $t$. We restate the updating rule by

$$\langle \mathbf{w}_{y,r}^{(t+1)}, y\mathbf{u} \rangle = \langle \mathbf{w}_{y,r}^{(t)}, y\mathbf{u} \rangle + \frac{\eta}{m} \cdot \left(1 - yf(\mathbf{x}; \mathbf{W}^{(t)})\right) \cdot \sigma'(\langle \mathbf{w}_{y,r}^{(t)}, y\mathbf{u} \rangle) \cdot \|\mathbf{u}\|_2^2,$$

from which we find $\max_{r \in [m]} \langle \mathbf{w}_{y,r}^{(t+1)}, y\mathbf{u} \rangle \geq \max_{r \in [m]} \langle \mathbf{w}_{y,r}^{(t)}, y\mathbf{u} \rangle$ must hold when $yf(\mathbf{x}; \mathbf{W}^{(t)}) \leq 1$. Otherwise, once $yf(\mathbf{x}; \mathbf{W}^{(t)}) > 1$, it immediately follows that

$$1 < yf(\mathbf{x}; \mathbf{W}^{(t)}) = F_y(\mathbf{x}; \mathbf{W}^{(t)}) - F_{-y}(\mathbf{x}; \mathbf{W}^{(t)})$$

$$\leq F_y(\mathbf{x}; \mathbf{W}^{(t)}) = \frac{1}{m} \sum_{r \in [m]} \sigma(\langle \mathbf{w}_{y,r}^{(t)}, y\mathbf{u} \rangle) + \sigma(\langle \mathbf{w}_{y,r}^{(t)}, y\mathbf{v} \rangle)$$

$$\leq \max_{r \in [m]} \langle \mathbf{w}_{y,r}^{(t)}, y\mathbf{u} \rangle^2 + 2 \log(16m/p) \cdot \sigma_0^2 \|\mathbf{v}\|_2^2.$$

Consequently, for the specific neuron $r^* = \operatorname{argmax}_{r \in [m]} \langle \mathbf{w}_{y,r}^{(t)}, y\mathbf{u} \rangle$, there holds

$$\langle \mathbf{w}_{y,r^*}^{(t+1)}, y\mathbf{u} \rangle \geq \langle \mathbf{w}_{y,r^*}^{(t)}, y\mathbf{u} \rangle - \frac{3\eta}{m} \cdot \langle \mathbf{w}_{y,r^*}^{(t)}, y\mathbf{u} \rangle \cdot \|\mathbf{u}\|_2^2$$

$$\geq \left(1 - 2 \log(16m/p) \cdot \sigma_0^2 \|\mathbf{v}\|_2^2\right) \cdot \left(1 - \frac{3\eta}{m} \cdot \|\mathbf{u}\|_2^2\right)$$

$$\geq \frac{\iota}{2},$$

where the last inequality is enabled by taking $\eta \leq m/6\|\mathbf{u}\|_2^2$ and $\sigma_0 \leq \sqrt{1-\iota}/(2 \log(16m/p)\|\mathbf{v}\|_2)$. Thus by induction, we have finished the proof of Lemma 27. $\square$

Our subsequently analysis confirms that even before $T^{\ddagger}$, the model can already fit the given data point by exploiting $\mathbf{u}$. For the given $0 < \epsilon < 1$, define a reference point $\mathbf{W}^*$ as

$$\mathbf{w}_{j,r}^* = \frac{4m(1+\epsilon)}{\iota} \cdot \frac{j\mathbf{u}}{\|\mathbf{u}\|_2^2}, \quad j \in \{\pm 1\}, r \in [m]. \tag{61}$$

**Lemma 28.** *Under the same condition as the previous lemma, for all $T^\dagger \leq t \leq T^\ddagger$, there holds that*

$$y\langle \nabla f(\mathbf{x}; \mathbf{W}^{(t)}), \mathbf{W}^* \rangle \geq 2 \cdot (1 + \epsilon).$$

*Proof of Lemma 28.* Recall the definition of the CNN in (4) and that $\mathbf{u} \perp \mathbf{v}$, so we have

$$
\begin{aligned}
y\langle \nabla f(\mathbf{x}; \mathbf{W}^{(t)}), \mathbf{W}^* \rangle &= \frac{1}{m} \sum_{j \in \{\pm 1\}, r \in [m]} \sigma'(\langle \mathbf{w}_j^{(t)}, y\mathbf{u} \rangle) \cdot \langle \mathbf{w}_{j,r}^*, y\mathbf{u} \rangle \\
&= \sum_{j \in \{\pm 1\}, r \in [m]} \sigma'(\langle \mathbf{w}_{j,r}^{(t)}, y\mathbf{u} \rangle) \cdot \frac{4(1 + \epsilon)}{\iota} \\
&\geq \max_{r \in [m]} \langle \mathbf{w}_{y,r}^{(t)}, y\mathbf{u} \rangle \cdot \frac{4(1 + \epsilon)}{\iota} \\
&\geq 2 \cdot (1 + \epsilon),
\end{aligned}
$$

where the last inequality is by $\max_{r \in [m]} \langle \mathbf{w}_{y,r}^{(t)}, y\mathbf{u} \rangle \geq \iota/2$ as shown by the previous lemma. $\square$

**Lemma 29.** *Continued from the previous setting, we know that for $T^\dagger \leq t \leq T^\ddagger$, it holds that*

$$\|\mathbf{W}^{(t)} - \mathbf{W}^*\|_F^2 - \|\mathbf{W}^{(t+1)} - \mathbf{W}^*\|_F^2 \geq 2\eta L(\mathbf{W}^{(t)}) - 2\eta\epsilon^2.$$

*Proof of Lemma 29.* Firstly we expand the difference by

$$\|\mathbf{W}^{(t)} - \mathbf{W}^*\|_F^2 - \|\mathbf{W}^{(t+1)} - \mathbf{W}^*\|_F^2 = 2\eta \cdot \langle \nabla L(\mathbf{W}^{(t)}), \mathbf{W}^{(t)} - \mathbf{W}^* \rangle - \eta^2 \cdot \|\nabla L(\mathbf{W}^{(t)})\|_F^2 \tag{62}$$

With one data point, $\nabla L(\mathbf{W}^{(t)}) = \ell^{(t)} \nabla f(\mathbf{x}; \mathbf{W}^{(t)})$ admits a simplified expression, where $\ell^{(t)} = f(\mathbf{W}^{(t)}, \mathbf{x}) - y$ denotes the fitting residual. Since the neural network $f(\mathbf{W}, \mathbf{x})$ is 2-homogeneous in $\mathbf{W}$ due to the activation function $\sigma(z) = \max\{z, 0\}^2$, we can have

$$\langle \nabla f(\mathbf{x}; \mathbf{W}^{(t)}), \mathbf{W}^{(t)} \rangle = 2f(\mathbf{x}; \mathbf{W}^{(t)}).$$

Stack these observations into the first term of previous difference expansion to obtain that

$$
\begin{aligned}
\langle \nabla L(\mathbf{W}^{(t)}), \mathbf{W}^{(t)} - \mathbf{W}^* \rangle &= \ell^{(t)} \cdot \langle \nabla f(\mathbf{x}; \mathbf{W}^{(t)}), \mathbf{W}^{(t)} - \mathbf{W}^* \rangle \\
&= \ell^{(t)} \cdot \left( 2f(\mathbf{x}; \mathbf{W}^{(t)}) - \langle \nabla f(\mathbf{x}; \mathbf{W}^{(t)}), \mathbf{W}^* \rangle \right) \\
&= 2\ell^{(t)} \cdot \left( f(\mathbf{x}; \mathbf{W}^{(t)}) - y \right) + \ell^{(t)} \cdot y \cdot \left( 2 - y\langle \nabla f(\mathbf{x}; \mathbf{W}^{(t)}), \mathbf{W}^* \rangle \right).
\end{aligned}
$$

Note that the first term is exactly $4L(\mathbf{W}^{(t)})$. As for the second term, we need to plug in Lemma 28 to see $2 - y\langle \nabla f(\mathbf{x}; \mathbf{W}^{(t)}), \mathbf{W}^* \rangle \leq -2\epsilon < 0$, so that

$$\left| \ell^{(t)} \cdot y \cdot \left( 2 - y\langle \nabla f(\mathbf{x}; \mathbf{W}^{(t)}), \mathbf{W}^* \rangle \right) \right| \leq \frac{1}{2}\left( \ell^{(t)} \right)^2 + 2\epsilon^2 = L(\mathbf{W}^{(t)}) + 2\epsilon^2.$$

As a result, we would know $\langle \nabla L(\mathbf{W}^{(t)}), \mathbf{W}^{(t)} - \mathbf{W}^* \rangle \geq 3L(\mathbf{W}^{(t)}) - 2\epsilon^2$. Next, an upper bound on the second order term $\eta^2 \cdot \|\nabla L(\mathbf{W}^{(t)})\|_F^2$ is given by

$$
\begin{aligned}
\eta^2 \cdot \|\nabla L(\mathbf{W}^{(t)})\|_F^2 &= \eta^2 \cdot \left( \ell^{(t)} \right)^2 \cdot \left( \|\mathbf{u}\|_2^2 \sum_{j \in \{\pm 1\}, r \in [m]} \sigma'(\langle \mathbf{w}_{j,r}^{(t)}, y\mathbf{u} \rangle) + \|\mathbf{v}\|_2^2 \sum_{j \in \{\pm 1\}, r \in [m]} \sigma'(\langle \mathbf{w}_{j,r}^{(t)}, y\mathbf{v} \rangle) \right) \\
&\leq \mathcal{O}\left( \max\{\|\mathbf{u}\|_2^2, \|\mathbf{v}\|_2^2\} \right) \cdot \eta^2 \cdot L(\mathbf{W}^{(t)}),
\end{aligned}
$$

since the dynamics of the inner products $\langle \mathbf{w}_{j,r}^{(t)}, y\mathbf{u} \rangle, \langle \mathbf{w}_{j,r}^{(t)}, y\mathbf{v} \rangle$ are well bounded by $\mathcal{O}(1)$ throughout the time we are considering. By scaling $\eta\mathcal{O}(\max\{\|\mathbf{u}\|^2, \|\mathbf{v}\|^2\}) \leq 1$, we would know that $\eta^2 \|\nabla L(\mathbf{W}^{(t)})\|_F^2 \leq \eta L(\mathbf{W}^{(t)})$. Eventually, continued from (62), we can completely prove this lemma. $\square$

Equipped with Lemmas 27, 28, and 29, we are ready to prove the main lemma for the second stage.

*Proof of Lemma 26.* Continued from Lemma 29, for any $t \geq T^\dagger$, it holds that

$$\frac{1}{t - T^\dagger + 1} \sum_{s=T^\dagger}^{t} L(\mathbf{W}^{(s)}) \leq \frac{\|\mathbf{W}^{(T^\dagger)} - \mathbf{W}^*\|_F^2}{2\eta(t - T^\dagger + 1)} + \epsilon^2.$$

Before proceeding to scale time $t$, it would be helpful to decompose $\|\mathbf{W}^{(T^\dagger)} - \mathbf{W}^*\|_F^2$ and to have an upper bound on this term,

$$\|\mathbf{W}^{(T^\dagger)} - \mathbf{W}^*\|_F^2$$

$$= \sum_{j \in \{\pm 1\}, r \in [m]} \frac{\langle \mathbf{w}_{j,r}^{(T^\dagger)} - \mathbf{w}_{j,r}^*, \mathbf{u} \rangle^2}{\|\mathbf{u}\|_2^2} + \frac{\langle \mathbf{w}_{j,r}^{(T^\dagger)} - \mathbf{w}_{j,r}^*, \mathbf{v} \rangle^2}{\|\mathbf{v}\|_2^2} + \left\| \left( \mathbf{I}_d - \frac{\mathbf{v}\mathbf{v}^\top}{\|\mathbf{v}\|_2^2} - \frac{\mathbf{u}\mathbf{u}^\top}{\|\mathbf{u}\|_2^2} \right) (\mathbf{w}_{j,r}^{(T^\dagger)} - \mathbf{w}_{j,r}^*) \right\|_2^2$$

$$\leq \sum_{j \in \{\pm 1\}, r \in [m]} \frac{2\langle \mathbf{w}_{j,r}^{(T^\dagger)}, \mathbf{u} \rangle^2 + 2\langle \mathbf{w}_{j,r}^*, \mathbf{u} \rangle^2}{\|\mathbf{u}\|_2^2} + \frac{\langle \mathbf{w}_{j,r}^{(T^\dagger)}, \mathbf{v} \rangle^2}{\|\mathbf{v}\|_2^2} + \left\| \left( \mathbf{I}_d - \frac{\mathbf{v}\mathbf{v}^\top}{\|\mathbf{v}\|_2^2} - \frac{\mathbf{u}\mathbf{u}^\top}{\|\mathbf{u}\|_2^2} \right) \mathbf{w}_{j,r}^{(0)} \right\|_2^2,$$

$$(63)$$

where we exploit the fact that $\mathbf{w}^*$ is parallel to $\mathbf{u}$ by Lemma (61), and the gradient steps only updates $\mathbf{w}$ along the directions of $\mathbf{u}, \mathbf{v}$. Recall that by Lemma 27,

$$\max_{j \in \pm 1, r \in [m]} \langle \mathbf{w}_{j,r}^{(T^\dagger)}, j\mathbf{u} \rangle = \Omega(1), \quad \max_{j \in \pm 1, r \in [m]} \left| \langle \mathbf{w}_{j,r}^{(T^\dagger)}, \mathbf{v} \rangle \right| = \widetilde{\mathcal{O}}(\sigma_0 \|\mathbf{v}\|_2),$$

and also that $\|\mathbf{w}_{j,r}^{(0)}\| = \widetilde{\mathcal{O}}(\sigma_0 \sqrt{d})$, the leading term in (63) would be $\sum_{j \in \{\pm 1\}, r \in [m]} \langle \mathbf{w}_{j,r}^*, \mathbf{u} \rangle^2 / \|\mathbf{u}\|_2^2$. Therefore, we conclude that $\|\mathbf{W}^{(T^\dagger)} - \mathbf{W}^*\|_F^2 \leq Cm^3 / \|\mathbf{u}\|_2^2$ for some constant $C > 0$. As a result, the average loss after iterations $T^\dagger$ can be bounded by

$$\frac{1}{t - T^\dagger + 1} \sum_{s=T^\dagger}^{t} L(\mathbf{W}^{(s)}) \leq \frac{Cm^3}{2\eta \|\mathbf{u}\|_2^2 (t - T^\dagger + 1)} + \epsilon^2, \quad \forall t^\dagger \leq t \leq T^\ddagger.$$

Then we choose time $T = T^\dagger + \lfloor Cm^3 / (2\eta\epsilon\|\mathbf{u}\|_2^2) \rfloor$ as stated in Lemma 26. Since $\|\mathbf{u}\|_2^2 / \|\mathbf{v}\|_2^2 \geq \widetilde{\Omega}(m^2)$ by Assumption 22, we can verify that $T \leq T^\ddagger$. In conclusion, the final output would be

$$\frac{1}{T - T^\dagger + 1} \sum_{s=T^\dagger}^{T} L(\mathbf{W}^{(s)}) \leq \epsilon + \epsilon^2 \leq 2\epsilon.$$

This finishes the proof of Lemma 26. $\qquad \square$

Combine Lemmas 25 and 26 to obtain the full version of Theorem 23.

# E    PROOFS FOR MAIN THEORETICAL RESULTS (SECTION 4)

In this section, we give a detailed proof for our main theoretical results for the multiple training data case, i.e., Theorem 5. The proofs follow the similar idea as the proofs for single training data case (Appendix C). The readers interested in the proofs are encouraged to first go through Appendix C to get the idea of the core steps. In Appendix E.1, we give a preliminary analysis of the SGD training dynamics. In Appendix E.2, we give an overview of the proofs with our fundamental reasoning towards weak signal learning. In Appendix E.3, we give the proof of Theorem 5. We prove other lemmas in subsequent sections.

## E.1    PRELIMINARY ANALYSIS

Recall that $\mathcal{W}$ is the index set of training data points which lack the strong feature patch. By Equation (3), the CNN weights are updated according to

$$\mathbf{w}_{j,r}^{(t+1)} = \mathbf{w}_{j,r}^{(t)} - \frac{j\eta}{m} \cdot \left( f(\mathbf{x}_{i_t}; \mathbf{W}^{(t)}) - y_{i_t} \right) \cdot \left( \sigma'(\langle \mathbf{w}_{j,r}^{(t)}, y_{i_t}\mathbf{u} \rangle) \cdot y_{i_t}\mathbf{u} \cdot \mathbf{1}\{i_t \notin \mathcal{W}\} \right.$$

$$\left. + \sigma'(\langle \mathbf{w}_{j,r}^{(t)}, y_{i_t}\mathbf{v} \rangle) \cdot y_{i_t}\mathbf{v} + \sigma'(\langle \mathbf{w}_{j,r}^{(t)} \cdot \boldsymbol{\xi}_{i_t} \rangle) \cdot \boldsymbol{\xi}_{i_t} + \sigma'(\langle \mathbf{w}_{j,r}^{(t)} \cdot \widetilde{\boldsymbol{\xi}}_{i_t} \rangle) \cdot \widetilde{\boldsymbol{\xi}}_{i_t} \cdot \mathbf{1}\{i_t \in \mathcal{W}\} \right).$$

$$(64)$$

Also, recall that the correct index sets for the strong and weak signal patches are defined as

$$\mathcal{U}_{j,+}^{(t)} := \left\{ r \in [m] : \langle \mathbf{w}_{j,r}^{(t)}, j\mathbf{u} \rangle \geq 0 \right\}, \quad \mathcal{V}_{j,+}^{(t)} := \left\{ r \in [m] : \langle \mathbf{w}_{j,r}^{(t)}, j\mathbf{v} \rangle \geq 0 \right\},$$

By the CNN expression (1), for each $j \in \{\pm 1\}$, the inner products that matter are

1. *positive neurons:* $\langle \mathbf{w}_{j,r}^{(t)}, j\mathbf{u} \rangle$ for $r \in \mathcal{U}_{j,+}^{(t)}$, $\langle \mathbf{w}_{j,r}^{(t)}, j\mathbf{v} \rangle$ for $r \in \mathcal{V}_{j,+}^{(t)}$;

2. *negative neurons:* $\langle \mathbf{w}_{-j,r}^{(t)}, j\mathbf{u} \rangle$ for $r \notin \mathcal{U}_{-j,+}^{(t)}$, $\langle \mathbf{w}_{-j,r}^{(t)}, j\mathbf{v} \rangle$ for $r \notin \mathcal{V}_{-j,+}^{(t)}$.

By (64), the update formula of these inner products of interests are given by

$$\langle \mathbf{w}_{j,r}^{(t+1)}, j\mathbf{u} \rangle = \langle \mathbf{w}_{j,r}^{(t)}, j\mathbf{u} \rangle + \frac{\eta \|\mathbf{u}\|_2^2}{m} \cdot \left(1 - y_{i_t} f(\mathbf{x}_{i_t}; \mathbf{W}^{(t)})\right) \cdot \sigma'(\langle \mathbf{w}_{j,r}^{(t)}, j\mathbf{u} \rangle j y_{i_t}) \cdot \mathbf{1}\{i_t \notin \mathcal{W}\} \tag{65}$$

$$\langle \mathbf{w}_{j,r}^{(t+1)}, j\mathbf{v} \rangle = \langle \mathbf{w}_{j,r}^{(t)}, j\mathbf{v} \rangle + \frac{\eta \|\mathbf{v}\|_2^2}{m} \cdot \left(1 - y_{i_t} f(\mathbf{x}_{i_t}; \mathbf{W}^{(t)})\right) \cdot \sigma'(\langle \mathbf{w}_{j,r}^{(t)}, j\mathbf{v} \rangle j y_{i_t}), \tag{66}$$

Also by (64), the update formula of the inner products with noise vectors are given by

$$\langle \mathbf{w}_{j,r}^{(t+1)}, \boldsymbol{\xi}_i \rangle = \langle \mathbf{w}_{j,r}^{(t)}, \boldsymbol{\xi}_i \rangle + \frac{\eta \cdot j y_{i_t}}{m} \cdot \left(1 - y_{i_t} f(\mathbf{x}_{i_t}; \mathbf{W}^{(t)})\right) \cdot \left( \sigma'(\langle \mathbf{w}_{j,r}^{(t)}, \boldsymbol{\xi}_{i_t} \rangle) \cdot \langle \boldsymbol{\xi}_{i_t}, \boldsymbol{\xi}_i \rangle \right.$$
$$\left. + \sigma'(\langle \mathbf{w}_{j,r}^{(t)}, \widetilde{\boldsymbol{\xi}}_{i_t} \rangle) \cdot \langle \widetilde{\boldsymbol{\xi}}_{i_t}, \boldsymbol{\xi}_i \rangle \cdot \mathbf{1}\{i_t \in \mathcal{W}\} \right), \quad i \in [n], \tag{67}$$

$$\langle \mathbf{w}_{j,r}^{(t+1)}, \widetilde{\boldsymbol{\xi}}_i \rangle = \langle \mathbf{w}_{j,r}^{(t)}, \widetilde{\boldsymbol{\xi}}_i \rangle + \frac{\eta \cdot j y_{i_t}}{m} \cdot \left(1 - y_{i_t} f(\mathbf{x}_{i_t}; \mathbf{W}^{(t)})\right) \cdot \left( \sigma'(\langle \mathbf{w}_{j,r}^{(t)}, \boldsymbol{\xi}_{i_t} \rangle) \cdot \langle \boldsymbol{\xi}_{i_t}, \widetilde{\boldsymbol{\xi}}_i \rangle \right.$$
$$\left. + \sigma'(\langle \mathbf{w}_{j,r}^{(t)}, \widetilde{\boldsymbol{\xi}}_{i_t} \rangle) \cdot \langle \widetilde{\boldsymbol{\xi}}_{i_t}, \widetilde{\boldsymbol{\xi}}_i \rangle \cdot \mathbf{1}\{i_t \in \mathcal{W}\} \right), \quad i \in \mathcal{W}.$$

At first glance, the update formulas given above seem intangible. The following proposition indicates that we can separate the neurons into two parts, with each individual part learning one kind of sample independently. For simplicity, we let $\tilde{\eta} := 2\eta \|\mathbf{u}\|_2^2/m$ and $\alpha = \|\mathbf{v}\|_2^2/\|\mathbf{u}\|_2^2$ and $\tilde{f}_t = y_{i_t} f(\mathbf{x}_{i_t}; \mathbf{W}^{(t)})$. For any $j \in \{\pm 1\}$ and $s \geq 0$, we define effective running time for learning label-$j$-samples as

$$t_j(s) = \min \left\{ t \in \mathbb{N} : t > t_j(s-1), \ y_{i_t} = j \right\}, \tag{68}$$

with $t_j(0) = \min\{t \in \mathbb{N} : y_{i_t} = j\}$.

**Proposition 30.** *Suppose that the sign stability condition holds before some $T_{\text{sign}}$, i.e., $\mathcal{U}_{\pm j, \pm}^{(t)} = \mathcal{U}_{\pm j, \pm}^{(0)} := \mathcal{U}_{\pm j, \pm}$ and $\mathcal{V}_{\pm j, \pm}^{(t)} = \mathcal{V}_{\pm j, \pm}^{(0)} := \mathcal{V}_{\pm j, \pm}$ for $t \leq T_{\text{sign}}$. Then for $s$ such that $t_j(s) \leq T_{\text{sign}}$, it holds that*

$$\langle \mathbf{w}_{j,r}^{(t_j(s+1))}, j\mathbf{u} \rangle = \langle \mathbf{w}_{j,r}^{(t_j(s))}, j\mathbf{u} \rangle \cdot \left(1 + \tilde{\eta}(1 - \tilde{f}_{t_j(s)}) \cdot \mathbf{1}\{i_{t_j(s)} \in \mathcal{W}\}\right), \quad \forall r \in \mathcal{U}_{j,+}; \tag{69}$$

$$\langle \mathbf{w}_{-j,r}^{(t_j(s+1))}, -j\mathbf{u} \rangle = \langle \mathbf{w}_{-j,r}^{(t_j(s))}, -j\mathbf{u} \rangle \cdot \left(1 - \tilde{\eta}(1 - \tilde{f}_{t_j(s)}) \cdot \mathbf{1}\{i_{t_j(s)} \in \mathcal{W}\}\right), \quad \forall r \in \mathcal{U}_{-j,-};$$

$$\langle \mathbf{w}_{j,r}^{(t_j(s+1))}, j\mathbf{v} \rangle = \langle \mathbf{w}_{j,r}^{(t_j(s))}, j\mathbf{v} \rangle \cdot \left(1 + \alpha \tilde{\eta}(1 - \tilde{f}_{t_j(s)})\right), \quad \forall r \in \mathcal{V}_{j,+};$$

$$\langle \mathbf{w}_{-j,r}^{(t_j(s+1))}, -j\mathbf{v} \rangle = \langle \mathbf{w}_{-j,r}^{(t_j(s))}, -j\mathbf{v} \rangle \cdot \left(1 - \alpha \tilde{\eta}(1 - \tilde{f}_{t_j(s)})\right), \quad \forall r \in \mathcal{V}_{-j,-}.$$

*Moreover, for every $t \in (t_j(s), t_j(s+1)]$, it holds that*

$$\langle \mathbf{w}_{j,r}^{(t)}, j\mathbf{u} \rangle = \langle \mathbf{w}_{j,r}^{(t_j(s)+1)}, j\mathbf{u} \rangle, \quad \forall r \in \mathcal{U}_{j,+};$$

$$\langle \mathbf{w}_{-j,r}^{(t)}, -j\mathbf{u} \rangle = \langle \mathbf{w}_{-j,r}^{(t_j(s)+1)}, -j\mathbf{u} \rangle, \quad \forall r \in \mathcal{U}_{-j,-};$$

$$\langle \mathbf{w}_{j,r}^{(t)}, j\mathbf{v} \rangle = \langle \mathbf{w}_{j,r}^{(t_j(s)+1)}, j\mathbf{v} \rangle, \quad \forall r \in \mathcal{V}_{j,+};$$

$$\langle \mathbf{w}_{-j,r}^{(t)}, -j\mathbf{v} \rangle = \langle \mathbf{w}_{-j,r}^{(t_j(s)+1)}, -j\mathbf{v} \rangle, \quad \forall r \in \mathcal{V}_{-j,-}.$$

*Proof of Proposition 30.* See Appendix E.6.1 for a detailed proof. □

This proposition helps to break down the dynamics of multiple data training into two folds. To see this, it suffices to note that the following components

$$\tilde{f}_{t_j(s)}, \quad \left\{\langle \mathbf{w}_{j,r}^{(t)}, j\mathbf{u} \rangle\right\}_{r \in \mathcal{U}_{j,+}}, \quad \left\{\langle \mathbf{w}_{-j,r}^{(t)}, -j\mathbf{u} \rangle\right\}_{r \in \mathcal{U}_{-j,-}}, \quad \left\{\langle \mathbf{w}_{j,r}^{(t)}, j\mathbf{v} \rangle\right\}_{r \in \mathcal{V}_{j,+}}, \quad \left\{\langle \mathbf{w}_{-j,r}^{(t)}, -j\mathbf{v} \rangle\right\}_{r \in \mathcal{V}_{-j,+}}$$

are independent with $\{\tilde{f}_t\}_{i_t = -j}$ and the rest inner products associated with $-j$.

### E.2 Overview of Analysis

The roadmap towards proving our main theorem shares nearly the same logic with the proof of single data setup (Appendix C). Basically, as long as the weak signal component and the noise component are not learned in the sense that these inner products remain negligible, we can prove that the inner products associated with the strong signal would dominate and the oscillation would accumulate at a linear rate. This further gives Lemma 32 which shows the CNN would learn effectively learn the weak signal $\langle \mathbf{w}_{j,r}^{(t)}, j\mathbf{v}\rangle$. Meanwhile, we can prove that the influences from the negative part of weak signal learning $\langle \mathbf{w}_{-j,r}^{(t)}, j\mathbf{v}\rangle$ and the noise memorization $\langle \mathbf{w}_{\pm 1,r}^{(t)}, \boldsymbol{\xi}_i\rangle$, $\langle \mathbf{w}_{\pm 1,r}^{(t)}, \widetilde{\boldsymbol{\xi}}_i\rangle$ can be well controlled (Propositions 33). Putting all together, we can prove the main result Theorem 5.

To be formal, we define two important stopping times as follows:

$$T_{(\mathbf{v})}^j = \min_{t \geq 0}\left\{t : \frac{1}{m}\sum_{r \in [m]}\sigma(\langle \mathbf{w}_{j,r}^{(t)}, j\mathbf{v}\rangle) \geq \delta/2\right\}, \tag{70}$$

$$T_{(\boldsymbol{\xi})} = \min_{t \geq 0}\left\{t : \max_{r \in [m], j \in \{\pm 1\}}\left\{\max_{i \in [n]}\left|\langle \mathbf{w}_{j,r}^{(t)}, \boldsymbol{\xi}_i\rangle\right|, \max_{i \in \mathcal{W}}\left|\langle \mathbf{w}_{j,r}^{(t)}, \widetilde{\boldsymbol{\xi}}_i\rangle\right|\right\} \geq \delta/4\right\}. \tag{71}$$

We recap that $\mathcal{W}$ denotes the index set of weak data, and $\widetilde{\boldsymbol{\xi}}_i$ denotes the Gaussian noise appearing on the lacking strong signal patch for those weak data. Also we note that $T_{(\mathbf{v})}^j, T_{(\boldsymbol{\xi})} \leq +\infty$, where the equal sign is attainable. We then define

$$T_{\max}^j = \min\left\{T_{(\mathbf{v})}^j, T_{(\boldsymbol{\xi})}\right\}.$$

In the first place, following the same arguments as in the single data setup (Appendix C), we can derive the boundedness and sign stability results before time $T_{\max}^j$.

**Lemma 31** (Boundedness and sign stability). *Under Assumptions 3 and 4, for fixed $j \in \{\pm 1\}$, the followings hold with probability at least $1 - p = 1 - 1/\mathrm{poly}(d)$:*

1. *it holds that $\mathcal{U}_{j,+}^{(t)} = \mathcal{U}_{j,+}^{(0)} \neq \emptyset$ and $\mathcal{V}_{j,+}^{(t)} = \mathcal{V}_{j,+}^{(0)} \neq \emptyset$ for any $t \in [0, T_{\max}^j]$. Hence the superscript $(t)$ can be dropped;*

2. *for any $t \in [0, T_{\max}^j]$, we have that*
$$\max_{r \in [m]}\langle \mathbf{w}_{j,r}^{(t)}, j\mathbf{u}\rangle \leq 1.5 \cdot (1.05\beta_{\mathbf{u},j}^* m)^{1/2}, \tag{72}$$
   *where $\beta_{\mathbf{u},j}^*$ is defined in Lemma 15;*

3. *for any $t \in [0, T_{\max}^j]$ it holds that*
$$\min_{r \in [m]}\langle \mathbf{w}_{-j,r}^{(t)}, -j\mathbf{u}\rangle \geq -\sqrt{2\log(16m/p)} \cdot \sigma_0\|\mathbf{u}\|_2, \tag{73}$$
$$\min_{r \in [m]}\langle \mathbf{w}_{-j,r}^{(t)}, -j\mathbf{v}\rangle \geq -\sqrt{2\log(16m/p)} \cdot \sigma_0\|\mathbf{v}\|_2; \tag{74}$$

4. *for any $t \in [0, T_{\max}^j]$ such that $y_{i_t} = j$, it holds that*
$$\left|1 - y_{i_t}f(\mathbf{x}_{i_t}; \mathbf{W}^{(t)})\right| \leq 2.$$

*Proof of Lemma 31.* See Appendix E.4 for a detailed proof. □

This boundedness and sign stability result further implies weak signal learning with an exponential rate, which is formally presented as follows.

**Lemma 32** (Weak signal learning). *Under Assumptions 3 and 4, with probability at least $1 - 1/\mathrm{poly}(d)$, it holds that for any $j \in \{\pm 1\}$ and $0 \leq t_1 \leq t_2 \leq T_{\max}^j$ that*

$$\sum_{\substack{s=t_1 \\ y_{i_s}=j}}^{t_2}\left(1 - y_{i_s}f(\mathbf{x}_{i_s}; \mathbf{W}^{(s)})\right) \geq \frac{\delta}{16} \cdot \left(1 - (1.05 - \delta/4)^{\frac{1}{2}}\right) \cdot (t_2 - t_1 + 1) - \frac{m(1.05)^{\frac{1}{2}}}{2\eta\|\mathbf{u}\|_2^2(1.05 - \delta/4)^{\frac{1}{2}}},$$

*where $\delta$ is specified in Assumption 4. Moreover, for $r \in \mathcal{V}_{j,+}^{(0)}$, we have that*

$$\langle \mathbf{w}_{j,r}^{(t_2+1)}, j\mathbf{v} \rangle \geq \langle \mathbf{w}_{j,r}^{(t_1)}, j\mathbf{v} \rangle \cdot \exp \left\{ \frac{\eta \|\mathbf{v}\|_2^2}{32m} \cdot \left( \delta - \delta(1.05 - \delta/4)^{\frac{1}{2}} \right) \cdot (t_2 - t_1 + 1) - \frac{\|\mathbf{v}\|_2^2 (1.05)^{\frac{1}{2}}}{\|\mathbf{u}\|_2^2 (1.05 - \delta/4)^{\frac{1}{2}}} \right\}.$$

(75)

*Proof of Lemma 32.* See Appendix E.5 for a detailed proof. $\square$

The last component to complete our proof of main theorem is controlling noise memorization during the training process. For simplicity, we define the maximum absolute value of the noise inner products over data as

$$\Upsilon^{(t)} = \max_{r \in [m], j \in \{\pm 1\}} \left\{ \max_{i \in [n]} \left| \langle \mathbf{w}_{j,r}^{(t)}, \boldsymbol{\xi}_i \rangle \right|, \max_{i \in \mathcal{W}} \left| \langle \mathbf{w}_{j,r}^{(t)}, \widetilde{\boldsymbol{\xi}}_i \rangle \right| \right\}.$$

We have the following proposition to control the growth of $\Upsilon^{(t)}$.

**Proposition 33** (Noise memorization). *Under Assumptions 3 and 4, then with probability at least $1 - 1/\mathrm{poly}(d)$, it holds for any $0 \leq t_1 \leq \min_{j \in \{\pm 1\}}\{T_{\max}^j\} - \widetilde{T}_{(\boldsymbol{\xi})}$ and $j \in \{\pm 1\}$ that*

$$\Upsilon^{(t)} \leq \Upsilon^{(t_1)} \cdot (1 + \epsilon), \quad \forall r \in [m], \quad \forall t_1 \leq t \leq t_1 + \widetilde{T}_{(\boldsymbol{\xi})}$$

*where $\widetilde{T}_{(\boldsymbol{\xi})} = \widetilde{\Theta}(mn \cdot \eta^{-1} \cdot \epsilon \cdot (1 + \epsilon)^{-1} \cdot (\sigma_p^2 d)^{-1})$ for any $\epsilon > 0$.*

*Proof of Proposition 33.* See Appendix E.6.3 for a detailed proof. $\square$

### E.3 PROOF OF THEOREM 5

With Lemma 31 and 32 and Proposition 33, we are ready to prove Theorem 5.

*Proof of Theorem 5.* For the $j = \mathrm{argmin}_{j' \in \{\pm 1\}}\{T_{\max}^{j'}\}$, we are going to prove that $T_{\max}^j$ is bounded by a polynomial time by using contradiction. Specifically, we prove that

$$T_{\max}^j \leq T_{j,0} := \frac{32m}{\eta \|\mathbf{v}\|_2^2} \cdot \left( \delta - \delta(1.05 - \delta/4)^{\frac{1}{2}} \right)^{-1} \cdot \left\{ \log \left( \frac{2\sqrt{m}\delta}{\sigma_0 \|\mathbf{v}\|_2} \right) + 1.5 \cdot \frac{\|\mathbf{v}\|_2^2}{\|\mathbf{u}\|_2^2} \cdot \sqrt{\frac{1.05}{1.05 - \delta/4}} \right\}.$$

Suppose that the result fails, then by definition we have that

$$T_{\max}^j = T_{(\mathbf{v})}^j \wedge T_{(\boldsymbol{\xi})} > T_{j,0}.$$

Then Lemma 31 and Lemma 32 hold on $[0, T_{j,0}]$. By applying the lower bound in Inequality (75) as well as Lemma 10, we have that

$$\max_{r \in [m]} \langle \mathbf{w}_{j,r}^{(T_{j,0})}, j\mathbf{v} \rangle \geq \max_{r \in [m]} \langle \mathbf{w}_{j,r}^{(0)}, j\mathbf{v} \rangle \cdot \exp \left\{ \frac{\eta \|\mathbf{v}\|_2^2}{32m} \cdot \left( \delta - \delta(1.05 - \delta/4)^{\frac{1}{2}} \right) \cdot T_{j,0} - 1.5 \cdot \frac{\|\mathbf{v}\|_2^2}{\|\mathbf{u}\|_2^2} \cdot \sqrt{\frac{1.05}{1.05 - \delta/4}} \right\}$$

$$\geq \frac{1}{2} \sigma_0 \|\mathbf{v}\|_2 \cdot \frac{2\sqrt{m}\delta}{\sigma_0 \|\mathbf{v}\|_2}$$

$$= \sqrt{m}\delta.$$

This leads to the following,

$$\frac{1}{m} \sum_{r \in [m]} \sigma(\langle \mathbf{w}_{j,r}^{(T_{j,0})}, j\mathbf{v} \rangle) \geq \frac{1}{m} \cdot \max_{r \in [m]} (\langle \mathbf{w}_{j,r}^{(T_{j,0})}, j\mathbf{v} \rangle)^2 \geq \delta,$$

which clearly contradicts the definition of $T_{(\mathbf{v})}^j$ in (70), hence it must be that $T_{\max}^j \leq T_{j,0} < +\infty$.

In the following, we prove that $T_{\max}^j = T_{(\mathbf{v})}^j < T_{(\boldsymbol{\xi})}$, for which our conclusion directly follows due to (70). Again we prove by contradiction. Suppose that $T_{\max}^j = T_{(\boldsymbol{\xi})} < T_{(\mathbf{v})}^j$, then $\Upsilon^{(t)}$ would reach $\delta/4$ for time less than $T_{j,0}$ due to definition of $T_{(\boldsymbol{\xi})}$ in (71). But by Proposition 33 with $\epsilon = 1$, we know $\Upsilon^{(t)}$ takes at least

$$K\widetilde{T}_{(\boldsymbol{\xi})} := \widetilde{\Theta}\left(\frac{mn\sigma_p^2 d}{\eta} \cdot \log\left(\frac{\delta}{\sigma_0 \sigma_p \sqrt{d}}\right)\right)$$

steps to reach $\delta/4$, where $\widetilde{T}_{(\boldsymbol{\xi})}$ is defined in Proposition 33. Then due to the scaling of the signal strength in Assumption 3, we can also find that $K\widetilde{T}_{(\boldsymbol{\xi})} > T_{j,0}$, which is a contradiction with the definition of $T_{(\boldsymbol{\xi})}$. Therefore, we must have that $T_{\max}^j = T_{(\mathbf{v})}^j$, which means that at $t^\star = T_{\max}^j$,

$$\frac{1}{m}\sum_{r\in[m]} \sigma(\langle \mathbf{w}_{j,r}^{(t^\star)}, j\mathbf{v}\rangle) \geq \frac{\delta}{2}.$$

In the meanwhile, Proposition 31 guarantees that at time $t^\star = T_{\max}^j$,

$$\frac{1}{m}\sum_{r\in[m]} \sigma(\langle \mathbf{w}_{-j,r}^{(t^\star)}, j\mathbf{v}\rangle) \leq \frac{\delta}{4}.$$

Thus, we conclude that at time $t^\star$,

$$\frac{1}{m}\sum_{r\in[m]} \sigma(\langle \mathbf{w}_{j,r}^{(t^\star)}, j\mathbf{v}\rangle) - \frac{1}{m}\sum_{r\in[m]} \sigma(\langle \mathbf{w}_{-j,r}^{(t^\star)}, j\mathbf{v}\rangle) \geq \frac{\delta}{4}.$$

This finishes the proof of Theorem 5. $\qquad\square$

### E.4 PROOF OF LEMMA 31

*Proof of Lemma 31.* Recall that $T_{\max}^j = T_{(\mathbf{v})}^j \wedge T_{(\boldsymbol{\xi})}$, where

$$T_{(\mathbf{v})}^j = \min_{t\geq 0}\left\{t : \frac{1}{m}\sum_{r\in[m]} \sigma(\langle \mathbf{w}_{j,r}^{(t)}, j\mathbf{v}\rangle) \geq \delta/2\right\},$$

$$T_{(\boldsymbol{\xi})} = \min_{t\geq 0}\left\{t : \max_{j\in\{\pm 1\}, r\in[m], i\in[n]}\left|\langle \mathbf{w}_{j,r}^{(t)}, \boldsymbol{\xi}_i\rangle\right| \vee \max_{j\in\{\pm 1\}, r\in[m], i\in\mathcal{W}}\left|\langle \mathbf{w}_{j,r}^{(t)}, \widetilde{\boldsymbol{\xi}}_i\rangle\right| \geq \delta/4\right\}.$$

We now define some notations to simplify our presentation. Recall that $\tilde{f} = y_{i_t} f(\mathbf{x}_{i_t}; \mathbf{W}^{(t)})$, $\tilde{\eta} = 2\eta\|\mathbf{u}\|_2^2/m$ and $\alpha = \|\mathbf{v}\|_2^2/\|\mathbf{u}\|_2^2$. For fixed $j$, we define

$$\bar{S}_{j,k} := \min\left\{s\in\mathbb{N} : s > \bar{S}_{j,k-1} \text{ such that } \tilde{f}_{t_j(s)} \geq 1 \text{ and } \tilde{f}_{t_j(\max\{s'<s:i_{t_j(s')}\notin\mathcal{W}\})} < 1\right\}, (76)$$

with $\bar{S}_{j,0} = 0$, and

$$\underline{S}_{j,k} := \min\left\{s\in\mathbb{N} : s > \underline{S}_{j,k-1} \text{ such that } i_{t_j(s)} \notin \mathcal{W}, \tilde{f}_{t_j(s)} < 1 \text{ and } \tilde{f}_{t_j(\max\{s'<s:i_{t_j(s')}\notin\mathcal{W}\})} \geq 1\right\}, \tag{77}$$

with $\underline{S}_{j,0} = 0$. From the definition of $T_{\max}^j$, we know that $i_{t_j(\bar{S}_{j,k})} \notin \mathcal{W}$ for $k$ such that $t_j(\bar{S}_{j,k}) \leq T_{\max}^j$, since otherwise $\tilde{f}_{t_j(\bar{S}_{j,k})} < \delta < 1$. Moreover we define

$$\mathcal{U}_{j,+} := \left\{r\in[m] : \langle \mathbf{w}_{j,r}^{(t_j(0))}, j\mathbf{u}\rangle > 0\right\}, \quad \mathcal{U}_{j,-} := \left\{r\in[m] : \langle \mathbf{w}_{j,r}^{(t_j(0))}, j\mathbf{u}\rangle \leq 0\right\},$$

and $\mathcal{V}_{j,\pm}$ are defined analogously.

The following analysis is nearly the same as the proof of one data case in the sense that each step of the proof is organized exactly the same to the proof of one data case. Therefore we suggest readers to refer to the roadmap provided in Appendix C.1 frequently for better understanding.

We note that for $t \leq T^j_{(\mathbf{v})} \wedge T_{(\boldsymbol{\xi})}$, which is the time scale we are mainly focusing on, it holds that

$$\max_{j \in \{\pm 1\}, r \in [m], i \in [n], i' \in \mathcal{W}} \left| \langle \mathbf{w}^{(t)}_{j,r}, \boldsymbol{\xi}_i \rangle \right| \vee \left| \langle \mathbf{w}^{(t)}_{j,r}, \widetilde{\boldsymbol{\xi}}_{i'} \rangle \right| \leq \frac{\delta}{4}, \quad \frac{1}{m} \sum_{r \in [m]} \sigma \big( \langle \mathbf{w}^{(t)}_{j,r}, j\mathbf{v} \rangle \big) < \frac{\delta}{2}.$$

Now we start presenting the proof of boundedness and sign stability with step-by-step analysis. The $j$ is arbitrary but fixed throughout analysis. In view of the effective running time for learning label-$j$-samples defined in (68), (76), and (77), the proofs for a certain $j$ directly do induction over $s$.

**Step 1: Pre-$\bar{S}_{j,1}$ analysis.** Clearly at $s = 0$, the lower bounds in Inequalities (73) and (74) are both guaranteed by Lemma 10. Also we know that $\tilde{f}_{t_j(0)} \ll 1$ and so $\bar{S}_{j,1} \geq 1$. The initialization also guarantees that the upper bound in Inequality (72) holds at $s = 0$.

For $s \in [0, \bar{S}_{j,1})$, the definition of $\bar{S}_{j,1}$ indicates that $\tilde{f}_{t_j(s)} \leq 1$. Therefore, from Proposition 30 we can see that the $\langle \mathbf{w}^{t_j(s)}_{j,r}, j\mathbf{u} \rangle$, $r \in \mathcal{U}_{j,+}$ and $\langle \mathbf{w}^{t_j(s)}_{-j,r}, -j\mathbf{u} \rangle$, $r \in \mathcal{U}_{-j,-}$ are non-decreasing in $s$ during this stage. One naturally infers that for all $r \in \mathcal{U}_{-j,-}$, it holds that

$$\langle \mathbf{w}^{(t_j(\bar{S}_{j,1}))}_{-j,r}, -j\mathbf{u} \rangle \geq \langle \mathbf{w}^{(t_j(\bar{S}_{j,1}-1))}_{-j,r}, -j\mathbf{u} \rangle \geq \langle \mathbf{w}^{(t_j(0))}_{-j,r}, -j\mathbf{u} \rangle \geq -\sqrt{2\log(16m/p)} \cdot \sigma_0 \|\mathbf{u}\|_2.$$

Same can be verified for $\langle \mathbf{w}^{(t_j(s))}_{-j,r}, -j\mathbf{v} \rangle$ with $r \in \mathcal{V}_{-j,-}$. Hence the lower bounds in Inequality (73) and (74) are extended to $s \in [0, \bar{S}_{j,k}]$. Also, for these inner products, the sign stability holds on $[0, \bar{S}_{j,1}]$, since $\tilde{\eta} < 4/5$ and thus we have

$$1 \pm \frac{2\eta \|\mathbf{u}\|_2^2}{m} \cdot (1 - \tilde{f}_{t_j(s)}) \geq 1 - \tilde{\eta}\big(1 + o(1)\big) \geq 0,$$

$$1 \pm \frac{2\eta \|\mathbf{v}\|_2^2}{m} \cdot (1 - \tilde{f}_{t_j(s)}) \geq 1 - \alpha\tilde{\eta}\big(1 + o(1)\big) \geq 0.$$

Then we turn to upper bound $\langle \mathbf{w}^{(t_j(s))}_{j,r}, j\mathbf{u} \rangle$ for $s \in [0, \bar{S}_{j,1})$. Note that, for $s$ such that $i_{t_j(s)} \notin \mathcal{W}$, the definition of $T^j_{\max}$ implies that

$$1 - \delta \geq \tilde{f}_{t_j(s)} \geq \frac{1}{m} \sum_{r \in [m]} \sigma\big( \langle \mathbf{w}^{(t_j(s))}_{j,r}, j\mathbf{u} \rangle \big) + \frac{1}{m} \sum_{r \in [m]} \sigma\big( \langle \mathbf{w}^{(t_j(s))}_{j,r}, j\mathbf{v} \rangle \big)$$

$$- \frac{1}{m} \sum_{r \in [m]} \sigma\big( -\langle \mathbf{w}^{(t_j(s))}_{-j,r}, -j\mathbf{u} \rangle \big) - \frac{1}{m} \sum_{r \in [m]} \sigma\big( -\langle \mathbf{w}^{(t_j(s))}_{-j,r}, -j\mathbf{v} \rangle \big)$$

$$- \left| \frac{1}{m} \sum_{r \in [m]} \sigma\big( \langle \mathbf{w}^{(t_j(s))}_{j,r}, \boldsymbol{\xi}_i \rangle \big) + \frac{1}{m} \sum_{r \in [m]} \sigma\big( \langle \mathbf{w}^{(t_j(s))}_{-j,r}, \boldsymbol{\xi}_i \rangle \big) \right|$$

$$\geq \frac{1}{m} \sum_{r \in [m]} \sigma\big( \langle \mathbf{w}^{(t_j(s))}_{j,r}, j\mathbf{u} \rangle \big) - 2 \times o(1) - 2 \times \delta/4.$$

As a consequence,

$$\frac{1}{m} \sum_{r \in [m]} \sigma\big( \langle \mathbf{w}^{(t_j(s))}_{j,r}, j\mathbf{u} \rangle \big) \leq 1 - \delta/2 + o(1) \leq 1.05.$$

Thanks to the local sign stability, Lemma 15 implies that

$$\max_{r \in [m]} \langle \mathbf{w}^{(t_j(s))}_{j,r}, j\mathbf{u} \rangle \leq (1.05 \beta^*_{\mathbf{u},j} m)^{1/2}.$$

If otherwise $s \in [0, \bar{S}_{j,k-1})$ such that $i_{t_j(s)} \in \mathcal{W}$, then we choose $\tilde{s} = \min\{s' > s : i_{t_j(s)} \notin \mathcal{W}\}$. It holds that either $\tilde{s} < \bar{S}_{j,1}$, or $\tilde{s} = \bar{S}_{j,1}$. For the previous case, similar argument implies that

$$\langle \mathbf{w}^{(t_j(s))}_{j,r}, j\mathbf{u} \rangle \leq \langle \mathbf{w}^{(t_j(\tilde{s}))}_{j,r}, j\mathbf{u} \rangle \leq (1.05 \beta^*_{\mathbf{u},j} m)^{1/2}. \tag{78}$$

The latter case reduces to establish the upper bound for $t_j(\bar{S}_{j,1})$, which is derived in the next step.

**Step 2.1: Bounding** $\langle \mathbf{w}_{j,r}^{(t_j(s))}, j\mathbf{u} \rangle$ **for** $s \in [\bar{S}_{j,1}, \underline{S}_{j,1})$. Since the weak data does not contributes to learning strong signal, as indicated in Proposition 30, we know that

$$\langle \mathbf{w}_{j,r}^{(t_j(\bar{S}_{j,1}))}, j\mathbf{u} \rangle = \langle \mathbf{w}_{j,r}^{(t_j(\tilde{S}_{j,1}))}, j\mathbf{u} \rangle \cdot \big( 1 + \tilde{\eta}(1 - \tilde{f}_{t_j(\tilde{S}_{j,1})})\big),$$

where $\tilde{S}_{j,1} = \max\{s \leq \tilde{S}_{j,1} - 1 : i_{t_j(s)} \notin \mathcal{W}\}$.

We begin with a proposition that is parallel to Proposition 19.

**Proposition 34.** *For simplicity, we assume that $i_{t_j(\bar{S}_{j,k}-1)} \notin \mathcal{W}$ and $\tilde{S}_{j,k} = \bar{S}_{j,k} - 1$. Otherwise we can find the last step before $\bar{S}_{j,k}$ such that $i_{t_j(s)} \notin \mathcal{W}$ and leverage the previous observation in Equation* (E.4). *Suppose that*

$$E_{t_j(\bar{S}_{j,k}-1)} := \frac{1}{m} \sum_{r \in [m]} \sigma\big( - \langle \mathbf{w}_{-j,r}^{(t_j(\bar{S}_{j,k}-1))}, -j\mathbf{u} \rangle \big) + \sigma\big( - \langle \mathbf{w}_{j,r}^{(t_j(\bar{S}_{j,k}-1))}, -j\mathbf{v} \rangle \big) \leq \delta/4,$$

$$\Upsilon^{(t_j(\bar{S}_{j,k}))} := \max_{j \in \{\pm 1\}, r \in [m], i \in [n]} \Big| \langle \mathbf{w}_{j,r}^{(t_j(\bar{S}_{j,k}))}, \boldsymbol{\xi}_i \rangle \Big| \vee \max_{j \in \{\pm 1\}, r \in [m], i \in \mathcal{W}} \Big| \langle \mathbf{w}_{j,r}^{(t_j(\bar{S}_{j,k}))}, \widetilde{\boldsymbol{\xi}}_i \rangle \Big| \leq \delta/4,$$

*then we have that*

$$\tilde{f}_{t_j(\bar{S}_{j,k}-1)} \geq \frac{\tilde{\eta} + 2 - \sqrt{\tilde{\eta}^2 + 4\tilde{\eta}}}{2\tilde{\eta}}, \tag{79}$$

*and that*

$$\tilde{f}_{t_j(\bar{S}_{j,k})} \leq \Big( 1 + \tilde{\eta}\big(1 - \tilde{f}_{t_j(\bar{S}_{j,k}-1)}\big)\Big)^2 + 2E_{t_j(\bar{S}_{j,k}-1)} + \Upsilon^{(t_j(\bar{S}_{j,k}))^2}$$

$$\leq \big( \tilde{\eta}/2 + \sqrt{\tilde{\eta}^2/4 + \tilde{\eta}} \big)^2 + 2E_{t_j(\bar{S}_{j,k}-1)} + \Upsilon^{(t_j(\bar{S}_{j,k}))^2}.$$

*Proof of Proposition 34.* See Appendix E.6.4 for a detailed proof. □

With this proposition, we can derive an upper bound on $\langle \mathbf{w}_{j,r}^{(t_j(\bar{S}_{j,1}))}, j\mathbf{u} \rangle$. One step gradient (69) implies that, for $r \in \mathcal{U}_{j,+}$, it holds that

$$\langle \mathbf{w}_{j,r}^{(t_j(\bar{S}_{j,1}))}, j\mathbf{u} \rangle = \langle \mathbf{w}_{j,r}^{(t_j(\bar{S}_{j,1}-1))}, j\mathbf{u} \rangle \cdot \Big( 1 + \tilde{\eta}\big(1 - \tilde{f}_{t_j(\bar{S}_{j,1}-1)}\big)\Big)$$

$$\leq \langle \mathbf{w}_{j,r}^{(t_j(\bar{S}_{j,1}-1))}, j\mathbf{u} \rangle \cdot \left( 1 + \tilde{\eta}\left( 1 - \frac{\tilde{\eta} + 2 - \sqrt{\tilde{\eta}^2 + 4\tilde{\eta}}}{2\tilde{\eta}} \right) \right)$$

$$\leq 1.5 \cdot (1.05\beta_{\mathbf{u},j}^* m)^{1/2}.$$

Here the first inequality comes from Inequality (79). Note that the upper bound on $\langle \mathbf{w}_{j,r}^{(t_j(\bar{S}_{j,1}-1))}, j\mathbf{u} \rangle$ has been derived in Inequality (78) in **Step 1.**. Taking $\tilde{\eta} = 4/5$ easily implies the second inequality above.

This upper bound continues to hold for $\langle \mathbf{w}_{j,r}^{(t_j(s))}, j\mathbf{u} \rangle$ with $s \in [\bar{S}_{j,1}, \underline{S}_{j,1})$ because of monotonicity. We consider the lower bound for $\langle \mathbf{w}_{j,r}^{(t_j(s))}, j\mathbf{u} \rangle$ with $s \in [\bar{S}_{j,1}, \underline{S}_{j,1})$ and $i_{t_j(s)} \notin \mathcal{W}$. For these $s$, we have that

$$1 + \delta < \tilde{f}_{t_j(s)} \leq \frac{1}{m} \sum_{r \in [m]} \sigma\big(\langle \mathbf{w}_{j,r}^{(t_j(s))}, j\mathbf{u} \rangle\big) + \frac{1}{m} \sum_{r \in [m]} \sigma\big(\langle \mathbf{w}_{j,r}^{(t_j(s))}, j\mathbf{v} \rangle\big) + \text{(negative part)} + \text{(noise part)}$$

$$\leq \frac{1}{m} \sum_{r \in [m]} \sigma\big(\langle \mathbf{w}_{j,r}^{(t_j(s))}, j\mathbf{u} \rangle\big) + \delta/2 + \delta/4.$$

Combining with Lemma 15, we obtain that

$$\max_{r \in [m]} \langle \mathbf{w}_{j,r}^{(t_j(s))}, j\mathbf{u} \rangle \geq (\beta_{\mathbf{u},j}^* m)^{1/2},$$

for $s \in [\bar{S}_{j,1}, \underline{S}_{j,1})$ and $i_{t_j(s)} \notin \mathcal{W}$.

**Step 2.2: Lower bounding** $\max_r \langle \mathbf{w}_{y,r}^{(t_j(S_{j,k}))}, y\mathbf{u} \rangle$. Same to the proof of Proposition 34, we can assume that $i_{t_j(\underline{S}_{j,1}-1)} \notin \mathcal{W}$, without loss of generality.

Note that, while $\tilde{f}_{t_j(s)}$ is not monotonically changing in $s \in [\bar{S}_{j,1}, \underline{S}_{j,1}]$ in the presence of noise components, the inner products are still changing monotonically, as indicated in Proposition 30. Therefore, we can leverage the control over noise components to derive an approximate monotonic property. Specifically, the inner products

$$\sigma\big(\langle \mathbf{w}_{j,r}^{(t_j(s))}, j\mathbf{u} \rangle\big), \quad -\sigma\big(\langle \mathbf{w}_{-j,r}^{(t_j(s))}, j\mathbf{u} \rangle\big), \quad \sigma\big(\langle \mathbf{w}_{j,r}^{(t_j(s))}, j\mathbf{v} \rangle\big), \quad -\sigma\big(\langle \mathbf{w}_{-j,r}^{(t_j(s))}, j\mathbf{v} \rangle\big),$$

are decreasing in $s \in [\bar{S}_{j,1}, \underline{S}_{j,1}]$. From Inequality (94) in the proof of Proposition 34, we know that

$$\tilde{f}_{t_j(\underline{S}_{j,1}-1)} \le \tilde{f}_{t_j(\bar{S}_{j,1})} - \frac{1}{m} \sum_{r \in [m]} \sigma\big(\langle \mathbf{w}_{j,r}^{(t_j(\bar{S}_{j,1}))}, \boldsymbol{\xi}_{i_{t_j(\bar{S}_{j,1})}} \rangle\big) - \sigma\big(\langle \mathbf{w}_{-j,r}^{(t_j(\bar{S}_{j,1}))}, \boldsymbol{\xi}_{i_{t_j(\bar{S}_{j,1})}} \rangle\big)$$

$$+ \frac{1}{m} \sum_{r \in [m]} \sigma\big(\langle \mathbf{w}_{j,r}^{(t_j(\underline{S}_{j,1}-1))}, \boldsymbol{\xi}_{i_{t_j(\underline{S}_{j,1}-1)}} \rangle\big) - \sigma\big(\langle \mathbf{w}_{-j,r}^{(t_j(\underline{S}_{j,1}-1))}, \boldsymbol{\xi}_{i_{t_j(\underline{S}_{j,1}-1)}} \rangle\big)$$

$$\le \big(\tilde{\eta}/2 + \sqrt{\tilde{\eta}^2/4 + \tilde{\eta}}\big)^2 + 2E_{T_j(\bar{S}_{j,1}-1)} + \Upsilon^{(t_j(\underline{S}_{j,1}))^2}. \tag{80}$$

Recall that in **Step 1.**, we proved that $E_{j(\bar{S}_{j,1}-1)} \le 2(\sqrt{2\log(16m/p)} \cdot \sigma_0 \|\mathbf{u}\|_2)^2 = o(1)$. Thus from doing one-step gradient descent (69), we know that for $r \in \mathcal{U}_{j,+}$, it holds that

$$\langle \mathbf{w}_{j,r}^{(t_j(\underline{S}_{j,1}))}, j\mathbf{u} \rangle = \langle \mathbf{w}_{j,r}^{(t_j(\underline{S}_{j,1}-1))}, j\mathbf{u} \rangle \cdot \Big(1 + \tilde{\eta}\big(1 - \tilde{f}_{t_j(\underline{S}_{j,1}-1)}\big)\Big)$$

$$\ge \langle \mathbf{w}_{j,r}^{(t_j(\underline{S}_{j,1}-1))}, j\mathbf{u} \rangle \cdot \Big(1 + \tilde{\eta}\big(1 - (\tilde{\eta}/2 + \sqrt{\tilde{\eta}^2/4 + \tilde{\eta}})^2\big) - 0.1\Big)$$

$$\ge 0.1 \cdot (\beta_{\mathbf{u},j}^* m)^{1/2}. \tag{81}$$

Here the first inequality is by (80), $E_{j(\bar{S}_{j,1}-1)} = o(1)$ and the definition of $T_{(\boldsymbol{\xi})}$, and the second inequality is a consequence of Inequality (26). This positive constant-level lower bound guarantees that our $\tilde{\eta}$ choice is sufficient for the sign stability to hold for $s \in [\bar{S}_{j,1}, \underline{S}_{j,2}]$, as shown in the next step.

**Step 2.3: Lower bounding** $\langle \mathbf{w}_{-j,r}^{(t_j(\underline{S}_{-j,1}))}, -j\mathbf{u} \rangle$ **and** $\langle \mathbf{w}_{-j,r}^{(t_j(\underline{S}_{j,1}))}, -j\mathbf{v} \rangle$. It suffices to consider $r \in \mathcal{V}_{-j,-}$ and $r \in \mathcal{U}_{-j,-}$. Inequality (81) indicates that

$$1 \ll \frac{\langle \mathbf{w}_{j,r}^{(t_j(\underline{S}_{j,1}))}, j\mathbf{u} \rangle}{\langle \mathbf{w}_{j,r}^{(t_j(0))}, j\mathbf{u} \rangle} = \prod_{s=0, i_{t_j(s)} \notin \mathcal{W}}^{\underline{S}_{j,1}-1} \Big(1 + \tilde{\eta}\big(1 - \tilde{f}_{t_j(s)}\big)\Big) \le \exp\left\{ \tilde{\eta} \sum_{s=0, i_{t_j(s)} \notin \mathcal{W}}^{\underline{S}_{j,1}-1} \big(1 - \tilde{f}_{t_j(s)}\big) \right\}.$$

Therefore,

$$\sum_{s=0, i_{t_j(s)} \in \mathcal{W}}^{\underline{S}_{j,1}-1} \big(1 - \tilde{f}_{t_j(s)}\big) > 0.$$

Now we can prove the lower bound. For $r \in \mathcal{U}_{-j,-}$, we have that

$$\langle \mathbf{w}_{-j,r}^{(t_j(\underline{S}_{j,1}))}, -j\mathbf{u} \rangle = \langle \mathbf{w}_{-j,r}^{(t_j(0))}, -j\mathbf{u} \rangle \cdot \prod_{s=0, i_{t_j(s)} \notin \mathcal{W}}^{\underline{S}_{j,1}-1} \Big(1 - \tilde{\eta}\big(1 - \tilde{f}_{t_j(s)}\big)\Big)$$

$$\ge \langle \mathbf{w}_{-j,r}^{(t_j(0))}, -j\mathbf{u} \rangle \cdot \exp\left\{ -\tilde{\eta} \sum_{s=0, i_{t_j(s)} \notin \mathcal{W}}^{\underline{S}_{j,1}-1} \big(1 - \tilde{f}_{t_j(s)}\big) \right\}$$

$$\ge \langle \mathbf{w}_{-j,r}^{(t_j(0))}, -j\mathbf{u} \rangle \ge -\sqrt{2\log(16m/p)} \cdot \sigma_0 \|\mathbf{u}\|_2. \tag{82}$$

On the other hand, for $s \in [0, \underline{S}_{j,1} - 1]$ such that $i_{t_j(s)} \in \mathcal{W}$, condition $t \leq T_{\max}^j = T_{(\mathbf{v})}^j \wedge T_{(\boldsymbol{\xi})}$ guarantees that

$$
\begin{aligned}
\tilde{f}_{t_j(s)} = \frac{1}{m} \sum_{r \in [m]} &\sigma\big(\langle \mathbf{w}_{j,r}^{(t_j(s))}, j\mathbf{v}\rangle\big) - \sigma\big(-\langle \mathbf{w}_{-j,r}^{(t_j(s))}, -j\mathbf{v}\rangle\big) \\
&+ \frac{1}{m} \sum_{r \in [m]} \sigma\big(\langle \mathbf{w}_{j,r}^{(t_j(s))}, \boldsymbol{\xi}_{i_{t_j(s)}}\rangle\big) - \sigma\big(\langle \mathbf{w}_{-j,r}^{(t_j(s))}, \boldsymbol{\xi}_{i_{t_j(s)}}\rangle\big) \\
&+ \frac{1}{m} \sum_{r \in [m]} \sigma\big(\langle \mathbf{w}_{j,r}^{(t_j(s))}, \widetilde{\boldsymbol{\xi}}_{i_{t_j(s)}}\rangle\big) - \sigma\big(\langle \mathbf{w}_{-j,r}^{(t_j(s))}, \widetilde{\boldsymbol{\xi}}_{i_{t_j(s)}}\rangle\big) \\
&\leq \delta/2 + 2 \times \delta/4 \\
&\leq 1.
\end{aligned}
$$

Hence we derive that

$$
\sum_{s \in [0, \underline{S}_{j,1}-1]} \big(1 - \tilde{f}_{t_j(s)}\big) = \sum_{\substack{s \in [0, \underline{S}_{j,1}-1] \\ i_{t_j(s)} \in \mathcal{W}}} \big(1 - \tilde{f}_{t_j(s)}\big) + \sum_{\substack{s \in [0, \underline{S}_{j,1}-1] \\ i_{t_j(s)} \notin \mathcal{W}}} \big(1 - \tilde{f}_{t_j(s)}\big) \geq 0.
$$

And in consequence, for $r \in \mathcal{V}_{j,-}$ it holds that

$$
\begin{aligned}
\langle \mathbf{w}_{-j,r}^{(t_j(S_{j,1}))}, -j\mathbf{v}\rangle &= \langle \mathbf{w}_{-j,r}^{(t_j(0))}, -j\mathbf{v}\rangle \cdot \prod_{s=0}^{\underline{S}_{j,1}-1} \Big(1 - \alpha\tilde{\eta}\big(1 - \tilde{f}_{t_j(s)}\big)\Big) \\
&\geq \langle \mathbf{w}_{-j,r}^{(t_j(0))}, -j\mathbf{v}\rangle \cdot \exp\left\{-\alpha\tilde{\eta} \sum_{s=0}^{\underline{S}_{j,1}-1} \big(1 - \tilde{f}_{t_j(s)}\big)\right\} \\
&\geq \langle \mathbf{w}_{-j,r}^{(t_j(0))}, -j\mathbf{v}\rangle \geq -\sqrt{2\log(16m/p)} \cdot \sigma_0 \|\mathbf{v}\|_2. \quad (83)
\end{aligned}
$$

The monotonicity again extends lower bounds in Inequalities (83) and (82) to $s \in [\bar{S}_{j,1}, \bar{S}_{j,2}]$. And the sign stability naturally holds on this interval.

**Step 2.4 & Finalizing.** Thanks to the sign stability proved in the last step, we can now apply Lemma 15 to derive that the upper bound in Inequality (72) continues to hold for all $s \in [\underline{S}_{j,1}, \bar{S}_{j,2}]$, with exactly the same argument as the **Step 1.**. With an inductive argument that exactly repeats the argument above, we can infer that all the results in Lemma 31 holds for $t_j(s)$ with $s \in [\bar{S}_{j,1}, \bar{S}_{j,2}]$. Proposition 30 implies that for any $t \in \mathbb{N}$ we can find $s_t \in \mathbb{N}$ such that $t_j(s_t) \leq t < t_j(s_t + 1)$ and $\langle \mathbf{w}_{j,r}^{(t)}, j\mathbf{u}\rangle = \langle \mathbf{w}_{j,r}^{(t_j(s_t))}, j\mathbf{u}\rangle$ (so are all other inner products), therefore all the results in Lemma 31 hold for $t \leq T_{\max}^j$. $\qquad\square$

### E.5 PROOF OF LEMMA 32

*Proof of Lemma 32.* For any $0 \leq t_1 \leq t_2 \leq T_{\max}^j$, $j \in \{\pm 1\}$, and $r \in \mathcal{U}_{j,+}^{(t)}$, by (65),

$$
\begin{aligned}
\langle \mathbf{w}_{j,r}^{(t_2+1)}, j\mathbf{u}\rangle = \langle \mathbf{w}_{j,r}^{(t_1)}, j\mathbf{u}\rangle &+ \frac{2\eta\|\mathbf{u}\|_2^2}{m} \cdot \sum_{\substack{t_1 \leq s \leq t_2 \\ y_{i_s}=j, i_s \notin \mathcal{W} \\ y_{i_s} f(\mathbf{x}_{i_s}; \mathbf{W}^{(s)}) > 1}} \big(1 - y_{i_s} f(\mathbf{x}_{i_s}; \mathbf{W}^{(s)})\big) \cdot \langle \mathbf{w}_{j,r}^{(s)}, j\mathbf{u}\rangle \\
&+ \frac{2\eta\|\mathbf{u}\|_2^2}{m} \cdot \sum_{\substack{t_1 \leq s \leq t_2 \\ y_{i_s}=j, i_s \notin \mathcal{W} \\ y_{i_s} f(\mathbf{x}_{i_s}; \mathbf{W}^{(s)}) < 1}} \big(1 - y_{i_s} f(\mathbf{x}_{i_s}; \mathbf{W}^{(s)})\big) \cdot \langle \mathbf{w}_{j,r}^{(s)}, j\mathbf{u}\rangle,
\end{aligned}
$$

Note that for $0 \leq t_1 \leq t_2 \leq T_{\max}^j$, we can apply the conclusions of Lemma 31. Specifically, we consider the maximal neuron $r^* = \operatorname{argmax}_{r \in [m]} \langle \mathbf{w}_{j,r}^{(t)}, j\mathbf{u} \rangle$, and

$$(1.05)^{\frac{1}{2}} \cdot (\beta_{\mathbf{u},j}^* m)^{\frac{1}{2}} \geq \left| \langle \mathbf{w}_{j,r^*}^{(t_2+1)}, j\mathbf{u} \rangle - \langle \mathbf{w}_{j,r^*}^{(t_1)}, j\mathbf{u} \rangle \right| \tag{84}$$

$$= \frac{2\eta \|\mathbf{u}\|_2^2}{m} \cdot \left| \sum_{\substack{t_1 \leq s \leq t_2 \\ y_{i_s}=j, i_s \notin \mathcal{W} \\ y_{i_s} f(\mathbf{x}_{i_s}; \mathbf{W}^{(s)}) > 1}} \left(1 - y_{i_s} f(\mathbf{x}_{i_s}; \mathbf{W}^{(s)})\right) \cdot \underbrace{\langle \mathbf{w}_{j,r^*}^{(s)}, j\mathbf{u} \rangle}_{>(\beta_{\mathbf{u},j}^* m)^{\frac{1}{2}}} \right.$$

$$\left. - \sum_{\substack{t_1 \leq s \leq t_2 \\ y_{i_s}=j, i_s \notin \mathcal{W} \\ y_{i_s} f(\mathbf{x}_{i_s}; \mathbf{W}^{(s)}) < 1}} \left(1 - y_{i_s} f(\mathbf{x}_{i_s}; \mathbf{W}^{(s)})\right) \cdot \underbrace{\langle \mathbf{w}_{j,r^*}^{(s)}, j\mathbf{u} \rangle}_{<(1.05-\delta/4)^{\frac{1}{2}} \cdot (\beta_{\mathbf{u},j}^* m)^{\frac{1}{2}}} \right|,$$

where the red remarks follow from Lemma 15. Rearranging terms, we conclude from (84) that

$$\sum_{\substack{t_1 \leq s \leq t_2 \\ y_{i_s}=j, i_s \notin \mathcal{W} \\ y_{i_s} f(\mathbf{x}_{i_s}; \mathbf{W}^{(s)}) < 1}} \left(1 - y_{i_s} f(\mathbf{x}_{i_s}; \mathbf{W}^{(s)})\right) \tag{85}$$

$$\geq (1.05 - \delta/4)^{-\frac{1}{2}} \cdot \sum_{\substack{t_1 \leq s \leq t_2 \\ y_{i_s}=j, i_s \notin \mathcal{W} \\ y_{i_s} f(\mathbf{x}_{i_s}; \mathbf{W}^{(s)}) > 1}} \left(y_{i_s} f(\mathbf{x}_{i_s}; \mathbf{W}^{(s)}) - 1\right) - \frac{m(1.05)^{\frac{1}{2}}}{2\eta \|\mathbf{u}\|_2^2 (1.05 - \delta/4)^{\frac{1}{2}}},$$

Under Assumption 3, with probability at least $1 - 1/\operatorname{poly}(d)$, it holds that

$$\left| \{ t_1 \leq s \leq t_2 : y_{i_s} = j, i_s \notin \mathcal{W} \} \right| \geq \frac{1}{4} \cdot (t_2 - t_1 + 1), \tag{86}$$

Combining (85) and (86), we can finally prove that

$$\sum_{\substack{s=t_1 \\ y_{i_s}=j, i_s \notin \mathcal{W}}}^{t_2} 1 - y_{i_s} f(\mathbf{x}_{i_s}; \mathbf{W}^{(s)}) \geq \frac{\delta}{8} \cdot \left(1 - (1.05 - \delta/4)^{\frac{1}{2}}\right) \cdot (t_2 - t_1 + 1) - \frac{m(1.05)^{\frac{1}{2}}}{2\eta \|\mathbf{u}\|_2^2 (1.05 - \delta/4)^{\frac{1}{2}}}.$$

$$\tag{87}$$

Finally, for the weak data $i_s \in \mathcal{W}$, under Assumption 3, with probability at least $1 - 1/\operatorname{poly}(d)$,

$$\left| \{ t_1 \leq s \leq t_2 : y_{i_s} = j, i_s \in \mathcal{W} \} \right| \leq 2\rho \cdot (t_2 - t_1 + 1) \leq \frac{\delta}{32} \cdot \left(1 - (1.05 - \delta/4)^{\frac{1}{2}}\right) \cdot (t_2 - t_1 + 1),$$

where the second inequality follows from the condition on $\rho$ by Assumption 3. By Lemma 31, we have that $|1 - y_{i_s} f(\mathbf{x}_{i_s}; \mathbf{W}^{(s)})| \leq 2$ for $0 \leq s \leq T_{\max}^j$ and $y_s = j$. Therefore, we have that

$$\sum_{\substack{s=t_1 \\ y_{i_s}=j, i_s \in \mathcal{W}}}^{t_2} 1 - y_{i_s} f(\mathbf{x}_{i_s}; \mathbf{W}^{(s)}) \geq -\frac{\delta}{16} \cdot \left(1 - (1.05 - \delta/4)^{\frac{1}{2}}\right) \cdot (t_2 - t_1 + 1). \tag{88}$$

Combining (87) and (88), we can conclude that

$$\sum_{\substack{s=t_1 \\ y_{i_s}=j}}^{t_2} 1 - y_{i_s} f(\mathbf{x}_{i_s}; \mathbf{W}^{(s)}) \geq \frac{\delta}{16} \cdot \left(1 - (1.05 - \delta/4)^{\frac{1}{2}}\right) \cdot (t_2 - t_1 + 1) - \frac{m(1.05)^{\frac{1}{2}}}{2\eta \|\mathbf{u}\|_2^2 (1.05 - \delta/4)^{\frac{1}{2}}}.$$

This finishes the proof of the first part in Lemma 32. Now we consider the second part. For simplicity, we denote by $\alpha = \|\mathbf{v}\|_2^2 / \|\mathbf{u}\|_2^2$ and $\tilde{\eta} = 2\|\mathbf{u}\|_2^2 / m$. Consider that for any $0 \leq t \leq T_{\max}^j$,

$j \in \{\pm 1\}$, and $r \in \mathcal{V}_{j,+}^{(t)}$, due to Lemma 31, the **v**-sign stability condition is true on $[0, T_{\max}^j]$. In view of (66), this means that

$$1 + \alpha\tilde{\eta} \cdot \left(1 - y_{i_s} f(\mathbf{x}_{i_s}; \mathbf{W}^{(s)})\right) > 0, \quad \forall 0 \le s \le T_{\max}^j \text{ s.t. } y_{i_s} = j.$$

Then for any $t_1 \le t \le T_{\max}^j$, $t_1 \le s \le t$, and $r \in \mathcal{C}_{j,+}^{(t)}$, since $-2 \le 1 - y_{i_s} f(\mathbf{x}_{i_s}; \mathbf{W}^{(t)}) \le 2$ due to Lemma 31, we can lower bound the relative increment as

$$\sum_{\substack{s=t_1 \\ y_{i_s}=j}}^{t} \log\left(1 + \alpha\tilde{\eta}\left(1 - y_{i_s} f(\mathbf{x}_{i_s}; \mathbf{W}^{(s)})\right)\right) = \sum_{\substack{s=t_1 \\ y_{i_s}=j}}^{t} \int_0^\alpha \frac{\tilde{\eta}\left(1 - y_{i_s} f(\mathbf{x}_{i_s}; \mathbf{W}^{(s)})\right)}{1 + \tilde{\eta} z\left(1 - y_{i_s} f(\mathbf{x}_{i_s}; \mathbf{W}^{(s)})\right)} dz$$

$$= \sum_{\substack{s=t_1 \\ y_{i_s}=j}}^{t} \int_0^\alpha \frac{\tilde{\eta}\left(\left(1 - y_{i_s} f(\mathbf{x}_{i_s}; \mathbf{W}^{(s)})\right) + 2\right)}{1 + \tilde{\eta} z\left(1 - y_{i_s} f(\mathbf{x}_{i_s}; \mathbf{W}^{(s)})\right)} dz - \sum_{\substack{s=t_1 \\ y_{i_s}=j}}^{t} \int_0^\alpha \frac{2\tilde{\eta}}{1 + \tilde{\eta} z\left(1 - y_{i_s} f(\mathbf{x}_{i_s}; \mathbf{W}^{(s)})\right)} dz$$

$$\ge \sum_{\substack{s=t_1 \\ y_{i_s}=j}}^{t} \int_0^\alpha \frac{\tilde{\eta}\left(\left(1 - y_{i_s} f(\mathbf{x}_{i_s}; \mathbf{W}^{(s)})\right) + 2\right)}{1 + 2\tilde{\eta} z} dz - \sum_{\substack{s=t_1 \\ y_{i_s}=j}}^{t} \int_0^\alpha \frac{2\tilde{\eta}}{1 - 2\tilde{\eta} z} dz$$

$$\ge \int_0^\alpha \frac{\tilde{\eta}\left(\sum_{s=t_1, y_{i_s}=j}^{t} \left(1 - y_{i_s} f(\mathbf{x}_{i_s}; \mathbf{W}^{(t)})\right) + 2N_j(t_0, t)\right)}{1 + 2\tilde{\eta} z} dz - \int_0^\alpha \frac{2\tilde{\eta} N_j(t_1, t)}{1 - 2\tilde{\eta} z} dz, \quad (89)$$

where for simplicity we denote $N_j(t_1, t) = |\{t_1 \le s \le t : y_{i_s} = j\}|$. Now we can use Proposition 32 to lower bound the right hand side of (89), Denoting $\epsilon = \delta \cdot (1 - (1.05 - \delta/4)^{\frac{1}{2}})/16$, we have that

$$\sum_{\substack{s=t_1 \\ y_{i_s}=j}}^{t} \log\left\{1 + \alpha\tilde{\eta}\left(1 - y_{i_s} f(\mathbf{x}_{i_s}; \mathbf{W}^{(s)})\right)\right\}$$

$$\ge \int_0^\alpha \frac{\tilde{\eta}\left(\epsilon(t - t_1 + 1) - \Delta + 2N_j(t_1, t)\right)}{1 + 2\tilde{\eta} z} dz - \int_0^\alpha \frac{2\tilde{\eta} N_j(t_1, t)}{1 - 2\tilde{\eta}} dz$$

$$\ge \left(\frac{\epsilon}{2}(t - t_1 + 1) - \frac{\Delta}{2} + N_j(t_1, t)\right) \cdot \log(1 + 2\alpha\tilde{\eta}) + N_j(t_1, t) \cdot \log(1 - 2\alpha\tilde{\eta})$$

$$\ge \left(\frac{\epsilon}{2}(t - t_1 + 1) - \frac{\Delta}{2}\right) \cdot \log(1 + 2\alpha\tilde{\eta}) + N_j(t_1, t) \cdot \log\left\{(1 + 2\alpha\tilde{\eta}) \cdot (1 - 2\alpha\tilde{\eta})\right\},$$

where $\Delta$ is defined in Lemma 32. Moreover since for our choice of $\alpha\tilde{\eta} \ll 1$ in Assumption 3,

$$\log(1 + 2\alpha\tilde{\eta}) \ge \alpha\tilde{\eta}, \quad \log\left\{(1 + 2\alpha\tilde{\eta}) \cdot (1 - 2\alpha\tilde{\eta})\right\} = \log\left\{1 - 4\alpha^2\tilde{\eta}^2\right\} \ge -2\alpha^2\tilde{\eta}^2,$$

and using the fact that with probability at least $1 - 1/\text{poly}(d)$ it holds that $N_j(t_1, t) \le (t - t_1 + 1)/2$, we finally have that

$$\sum_{\substack{s=t_1 \\ y_{i_s}=j}}^{t} \log\left\{1 + \alpha\tilde{\eta}\left(1 - y_{i_s} f(\mathbf{x}_{i_s}; \mathbf{W}^{(s)})\right)\right\} \ge \frac{\alpha\tilde{\eta}}{2} \cdot (\epsilon - 2\alpha\tilde{\eta}) \cdot (t_1 - t_1 + 1) - \Delta \cdot \log(1 + 2\alpha\tilde{\eta})$$

$$\ge \frac{1}{4}\alpha\tilde{\eta} \cdot \epsilon \cdot (t_1 - t_1 + 1) - 2\alpha\tilde{\eta} \cdot \Delta.$$

Finally, using (66) again, we can lower bound our target as

$$\langle \mathbf{w}_{j,r}^{(t+1)}, j\mathbf{v} \rangle = \langle \mathbf{w}_{j,r}^{(t_1)}, j\mathbf{v} \rangle \cdot \prod_{\substack{s=t_1 \\ y_{i_s}=j}}^{t} \Big( 1 + \alpha\tilde{\eta}\big(1 - y_{i_s} f(\mathbf{x}_{i_s}; \mathbf{W}^{(s)})\big) \Big)$$

$$= \langle \mathbf{w}_{j,r}^{(t_1)}, j\mathbf{v} \rangle \cdot \exp\left\{ \sum_{\substack{s=t_1 \\ y_{i_s}=j}}^{t} \log\Big\{ 1 + \alpha\tilde{\eta}\big(1 - y_{i_s} f(\mathbf{x}_{i_s}; \mathbf{W}^{(s)})\big) \Big\} \right\}$$

$$\geq \langle \mathbf{w}_{j,r}^{(t_1)}, j\mathbf{v} \rangle \cdot \exp\left\{ \frac{1}{4}\alpha\tilde{\eta} \cdot \epsilon \cdot (t_1 - t_1 + 1) - 2\alpha\tilde{\eta} \cdot \Delta \right\}.$$

Plugging in the definition of $\epsilon$, $\Delta$, $\alpha$, and $\tilde{\eta}$, we can arrive at

$$\langle \mathbf{w}_{j,r}^{(t+1)}, j\mathbf{v} \rangle \geq \langle \mathbf{w}_{j,r}^{(t_1)}, j\mathbf{v} \rangle \cdot \exp\left\{ \frac{\eta\|\mathbf{v}\|_2^2}{32m} \cdot \big(\delta - \delta(1.05 - \delta/4)^{\frac{1}{2}}\big) \cdot (t - t_1 + 1) - \frac{\|\mathbf{v}\|_2^2(1.05)^{\frac{1}{2}}}{\|\mathbf{u}\|_2^2(1.05 - \delta/4)^{\frac{1}{2}}} \right\}.$$

This finishes the proof of Lemma 32. $\qquad\square$

### E.6 Proof of Technical Results

#### E.6.1 Proof of Proposition 30

*Proof of Proposition 30.* From Equation (65), for $r \in \mathcal{U}_{j,+}$ and $t' \in (t_j(s), t_j(s+1)]$, we can infer that

$$\langle \mathbf{w}_{j,r}^{(t')}, j\mathbf{u} \rangle = \langle \mathbf{w}_{j,r}^{(t_j(s))}, j\mathbf{u} \rangle + \sum_{t=t_j(s)}^{t'-1} \frac{\eta\|\mathbf{u}\|_2^2}{m} \cdot \big(1 - \tilde{f}_t\big) \cdot \sigma'\big(\langle \mathbf{w}_{j,r}^{(t)}, j\mathbf{u} \rangle \cdot jy_{i_t}\big) \cdot \mathbf{1}\{i_t \in \mathcal{W}\}.$$

The definition of $t_j(s)$ implies that for $t \in (t_j(s), t_j(s+1))$, $y_{i_t}j = -1$. On the other hand $y_{i_{t_j(s)}}j = 1$, hence we obtain that

$$\langle \mathbf{w}_{j,r}^{(t')}, j\mathbf{u} \rangle = \langle \mathbf{w}_{j,r}^{(t_j(s))}, j\mathbf{u} \rangle + \frac{2\eta\|\mathbf{u}\|_2^2}{m} \cdot \big(1 - \tilde{f}_{t_j(s)}\big) \cdot \langle \mathbf{w}_{j,r}^{(t)}, j\mathbf{u} \rangle \cdot \mathbf{1}\{i_{t_j(s)} \in \mathcal{W}\}.$$

Since the sign stability holds, this multiplication by a non-negative factor does not change the sign of the inner products. Therefore we have that

$$\langle \mathbf{w}_{j,r}^{(t_j(s+1))}, j\mathbf{u} \rangle = \langle \mathbf{w}_{j,r}^{(t_j(s+1)-1)}, j\mathbf{u} \rangle = \cdots = \langle \mathbf{w}_{j,r}^{(t_j(s)+1)}, j\mathbf{u} \rangle$$

$$= \langle \mathbf{w}_{j,r}^{(t_j(s))}, j\mathbf{u} \rangle \cdot \big(1 + \tilde{\eta} \cdot (1 - \tilde{f}_{t_j(s)})\big).$$

which concludes our result for $\langle \mathbf{w}_{j,r}^{(t)}, j\mathbf{u} \rangle$, $r \in \mathcal{U}_{j,+}$. Other result can be proved analogously and are omitted here. $\qquad\square$

#### E.6.2 Proof of Lemma 15 for Multiple Data Setting

*Proof of Lemma 15 (multiple data setting).* In the multiple data setting, the (positive) inner products only changes at the steps where the corresponding data label aligns with the directions of the neurons (i.e., $j = \pm 1$). Define

$$\beta_{\mathbf{u},j}^{*,(t_1)} = \frac{\max_{r \in [m]} \sigma(\langle \mathbf{w}_{j,r}^{(t_1)}, j\mathbf{u} \rangle)}{\sum_{r \in [m]} \sigma(\langle \mathbf{w}_{j,r}^{(t_1)}, j\mathbf{u} \rangle)}.$$

Again, the local sign stability assumption ensures that each inner product grows proportionally and the superscript $(t)$ in the neuron index sets can be dropped, with

$$\frac{1}{m}\sum_{r\in[m]}\sigma\big(\langle\mathbf{w}_{j,r}^{(t)},j\mathbf{u}\rangle\big) = \frac{1}{m}\sum_{r\in\mathcal{U}_{j,+}}\sigma\big(\langle\mathbf{w}_{j,r}^{(t_1)},j\mathbf{u}\rangle\big)\cdot\prod_{\substack{t'\in[t_1,t-1]:\\ y_{i_t}=j,\, i_t\notin\mathcal{W}}}\left(1+\frac{2\eta\|\mathbf{u}\|_2^2}{m}\cdot\big(1-y_{i_{t'}}f(\mathbf{x}_{i_{t'}};\mathbf{W}^{(t')})\big)\right)^2$$

$$= \frac{\max_{r\in[m]}\sigma\big(\langle\mathbf{w}_{j,r}^{(t_1)},j\mathbf{u}\rangle\big)}{m\beta_{\mathbf{u},j}^{*,(t_1)}}\cdot\prod_{\substack{t'\in[t_1,t-1]:\\ y_{i_{t'}}=j,\, i_t\notin\mathcal{W}}}\left(1+\frac{2\eta\|\mathbf{u}\|_2^2}{m}\cdot\big(1-y_{i_{t'}}f(\mathbf{x}_{i_{t'}};\mathbf{W}^{(t')})\big)\right)^2$$

$$= \frac{\sigma\big(\max_{r\in[m]}\langle\mathbf{w}_{j,r}^{(t)},j\mathbf{u}\rangle\big)}{m\beta_{\mathbf{u},j}^{*,(t_1)}}$$

Here the first and the third equality is true because Equation (8) implies that all the positive $\langle\mathbf{w}_{y,r}^{(t)},y\mathbf{u}\rangle$ iterates by sequentially multiplying the same factor

$$1+\frac{2\eta\|\mathbf{u}\|_2^2}{m}\cdot\big(1-yf(\mathbf{x};\mathbf{W}^{(t')})\big).$$

The second equality comes from the definition of $\beta_{\mathbf{u}}^{*,(t_1)}$ in Lemma 15.

Therefore, $m^{-1}\cdot\sum_{r\in[m]}\sigma(\langle\mathbf{w}_{j,r}^{(t)},j\mathbf{u}\rangle) > c$ implies that $\sigma(\max_{r\in[m]}\langle\mathbf{w}_{y,r}^{(t)},y\mathbf{u}\rangle) > \beta_{\mathbf{u},j}^{*,(t_1)}mc$ and the desired lower bound follows. The upper bound can be proved analogously. $\qquad\square$

### E.6.3 Proof of Proposition 33

*Proof of Proposition 33.* We prove the result by induction. For the step $t = t_1$, the result holds trivially. Suppose that this result holds for each step $t_1,\cdots,t$, then by (67), we have that

$$\langle\mathbf{w}_{j,r}^{(t+1)},\boldsymbol{\xi}_i\rangle = \langle\mathbf{w}_{j,r}^{(t_1)},\boldsymbol{\xi}_i\rangle + \sum_{s=t_1}^{t}\frac{\eta\cdot jy_{i_s}}{m}\cdot\big(1-y_{i_s}f(\mathbf{x}_{i_s};\mathbf{W}^{(s)})\big)\cdot\Big(\sigma'(\langle\mathbf{w}_{j,r}^{(s)}\cdot\boldsymbol{\xi}_{i_s}\rangle)\cdot\langle\boldsymbol{\xi}_{i_s},\boldsymbol{\xi}_i\rangle$$

$$+\,\sigma'(\langle\mathbf{w}_{j,r}^{(s)}\cdot\widetilde{\boldsymbol{\xi}}_{i_s}\rangle)\cdot\langle\widetilde{\boldsymbol{\xi}}_{i_s},\boldsymbol{\xi}_i\rangle\cdot\mathbf{1}\{i_s\in\mathcal{W}\}\Big)$$

$$=\langle\mathbf{w}_{j,r}^{(t_1)},\boldsymbol{\xi}_i\rangle + \sum_{\substack{s=t_1\\ i_s=i}}^{t}\frac{\eta\|\boldsymbol{\xi}_i\|_2^2\cdot jy_{i_s}}{m}\cdot\big(1-y_if(\mathbf{x}_i;\mathbf{W}^{(s)})\big)\cdot\sigma'(\langle\mathbf{w}_{j,r}^{(s)}\cdot\boldsymbol{\xi}_i\rangle)$$

$$+\sum_{\substack{s=t_1\\ i_s\neq i}}^{t}\frac{\eta\|\boldsymbol{\xi}_i\|_2^2\cdot jy_{i_s}}{m}\cdot\big(1-y_{i_s}f(\mathbf{x}_{i_s};\mathbf{W}^{(s)})\big)\cdot\Big(\sigma'(\langle\mathbf{w}_{j,r}^{(s)}\cdot\boldsymbol{\xi}_{i_s}\rangle)\cdot\frac{\langle\boldsymbol{\xi}_{i_s},\boldsymbol{\xi}_i\rangle}{\|\boldsymbol{\xi}_i\|_2^2}$$

$$+\,\sigma'(\langle\mathbf{w}_{j,r}^{(s)}\cdot\widetilde{\boldsymbol{\xi}}_{i_s}\rangle)\cdot\frac{\langle\widetilde{\boldsymbol{\xi}}_{i_s},\boldsymbol{\xi}_i\rangle}{\|\boldsymbol{\xi}_i\|_2^2}\cdot\mathbf{1}\{i_s\in\mathcal{W}\}\Big)$$

Taking absolute value, we have that

$$
\left| \langle \mathbf{w}_{j,r}^{(t+1)}, \boldsymbol{\xi}_i \rangle \right| \leq \left| \langle \mathbf{w}_{j,r}^{(t_1)}, \boldsymbol{\xi}_i \rangle \right| + \left| \sum_{\substack{s=t_1 \\ i_s=i}}^{t} \frac{\eta \|\boldsymbol{\xi}_i\|_2^2 \cdot j y_{i_s}}{m} \cdot \left( 1 - y_i f(\mathbf{x}_i; \mathbf{W}^{(s)}) \right) \cdot \sigma'(\langle \mathbf{w}_{j,r}^{(s)} \cdot \boldsymbol{\xi}_i \rangle) \right|
$$

$$
+ \left| \sum_{\substack{s=t_1 \\ i_s \neq i}}^{t} \frac{\eta \|\boldsymbol{\xi}_i\|_2^2 \cdot j y_{i_s}}{m} \cdot \left( 1 - y_{i_s} f(\mathbf{x}_{i_s}; \mathbf{W}^{(s)}) \right) \right.
$$

$$
\left. \cdot \left( \sigma'(\langle \mathbf{w}_{j,r}^{(s)} \cdot \boldsymbol{\xi}_{i_s} \rangle) \cdot \frac{\langle \boldsymbol{\xi}_{i_s}, \boldsymbol{\xi}_i \rangle}{\|\boldsymbol{\xi}_i\|_2^2} + \sigma'(\langle \mathbf{w}_{j,r}^{(s)} \cdot \widetilde{\boldsymbol{\xi}}_{i_s} \rangle) \cdot \frac{\langle \widetilde{\boldsymbol{\xi}}_{i_s}, \boldsymbol{\xi}_i \rangle}{\|\boldsymbol{\xi}_i\|_2^2} \cdot \mathbf{1}\{i_s \in \mathcal{W}\} \right) \right|
$$

Applying the definition of $\Upsilon^{(s)}$, we further obtain that

$$
\left| \langle \mathbf{w}_{j,r}^{(t+1)}, \boldsymbol{\xi}_i \rangle \right| \leq \Upsilon^{(t_1)} + \sum_{\substack{s=t_1 \\ i_s=i}}^{t} \frac{2\eta \|\boldsymbol{\xi}_i\|_2^2}{m} \cdot \left| 1 - y_i f(\mathbf{x}_i; \mathbf{W}^{(s)}) \right| \cdot \Upsilon^{(s)}
$$

$$
+ \sum_{\substack{s=t_1 \\ i_s \neq i}}^{t} \frac{2\eta \|\boldsymbol{\xi}_i\|_2^2}{m} \cdot \left| 1 - y_{i_s} f(\mathbf{x}_{i_s}; \mathbf{W}^{(s)}) \right| \cdot \Upsilon^{(s)} \cdot \left( \frac{|\langle \boldsymbol{\xi}_{i_s}, \boldsymbol{\xi}_i \rangle|}{\|\boldsymbol{\xi}_i\|_2^2} + \frac{|\langle \widetilde{\boldsymbol{\xi}}_{i_s}, \boldsymbol{\xi}_i \rangle|}{\|\boldsymbol{\xi}_i\|_2^2} \right) \tag{90}
$$

Now using Lemma 31, for $s \leq \min_{j \in \{\pm 1\}} \{T_{\max}^j\}$, we have that $|1 - y_{i_s} f(\mathbf{x}_{i_s}; \mathbf{W}^{(s)})| \leq 2$. Also, by Lemma 9, it holds that $\|\boldsymbol{\xi}_i\|_2^2 \leq 3\sigma_p^2 d/2$ and $|\langle \boldsymbol{\xi}, \boldsymbol{\xi}' \rangle|/\|\boldsymbol{\xi}\|_2^2 \leq \widetilde{\mathcal{O}}(d^{-1/2})$. Thus we can further upper bound (90) as

$$
\max_{i \in [n]} \left| \langle \mathbf{w}_{j,r}^{(t+1)}, \boldsymbol{\xi}_i \rangle \right| \leq \Upsilon^{(t_1)} + \frac{6\eta \sigma_p^2 d}{m} \cdot \left( \sum_{\substack{s=t_1 \\ i_s=i}}^{t} \Upsilon^{(s)} + \widetilde{\mathcal{O}}(d^{-1/2}) \cdot \sum_{\substack{s=t_1 \\ i_s \neq i}}^{t} \Upsilon^{(s)} \right) \tag{91}
$$

$$
= \Upsilon^{(t_1)} + \frac{6\eta \sigma_p^2 d}{m} \cdot \left( \sum_{\substack{s=t_1 \\ i_s=i}}^{t} \Upsilon^{(t_1)} + \widetilde{\mathcal{O}}(d^{-1/2}) \cdot \sum_{\substack{s=t_1 \\ i_s \neq i}}^{t} \Upsilon^{(t_1)} \right)
$$

$$
+ \frac{6\eta \sigma_p^2 d}{m} \cdot \left( \sum_{\substack{s=t_1 \\ i_s=i}}^{t} \left( \Upsilon^{(t_1)} - \Upsilon^{(s)} \right) + \widetilde{\mathcal{O}}(d^{-1/2}) \cdot \sum_{\substack{s=t_1 \\ i_s \neq i}}^{t} \left( \Upsilon^{(t_1)} - \Upsilon^{(s)} \right) \right).
$$

By our induction, we have that $\Upsilon^{(s)} - \Upsilon^{(t_1)} \leq \Upsilon^{(t_1)} \cdot \epsilon$, for which we can further bound (91) as

$$
\max_{i \in [n]} \left| \langle \mathbf{w}_{j,r}^{(t+1)}, \boldsymbol{\xi}_i \rangle \right| \leq \Upsilon^{(t_1)} \cdot \left[ 1 + \frac{6\eta \sigma_p^2 d}{m} \cdot (1+\epsilon) \cdot \left( \sum_{\substack{s=t_1 \\ i_s=i}}^{t} 1 + \widetilde{\mathcal{O}}(d^{-1/2}) \cdot \sum_{\substack{s=t_1 \\ i_s \neq i}}^{t} 1 \right) \right]
$$

$$
\leq \Upsilon^{(t_1)} \cdot \left[ 1 + \frac{6\eta \sigma_p^2 d}{m} \cdot (1+\epsilon) \cdot \left( \frac{2(t-t_1+1)}{n} + \widetilde{\mathcal{O}}(d^{-1/2}) \cdot (t-t_1+1) \right) \right]
$$

$$
\leq \Upsilon^{(t_1)} \cdot \left[ 1 + \frac{18\eta \sigma_p^2 d}{mn} \cdot (1+\epsilon) \cdot (t-t_1+1) \right],
$$

where in the first inequality we have utilized the fact that $|\{t_1 \leq s \leq t : i_s = i\}| \leq 2(t-t_1+1)/n$, and in the last inequality we apply the condition that $d = \widetilde{\Omega}(n^2)$ by Assumption 3. Therefore, when $t \leq t_1 + \widetilde{T}_{(\boldsymbol{\xi})} - 1$ with $\widetilde{T}_{(\boldsymbol{\xi})} = \widetilde{\Theta}(mn \cdot \eta^{-1} \cdot \epsilon \cdot (1+\epsilon)^{-1} \cdot (\sigma_p^2 d)^{-1})$, it holds that

$$
\max_{i \in [n]} \left| \langle \mathbf{w}_{j,r}^{(t+1)}, \boldsymbol{\xi}_i \rangle \right| \leq \Upsilon^{(t_1)} \cdot (1+\epsilon). \tag{92}
$$

By using the same argument as proving (92), we can also show that for $t \leq t_1 + \widetilde{T}_{(\boldsymbol{\xi})} - 1$

$$\max_{i \in \mathcal{W}} \left| \langle \mathbf{w}_{j,r}^{(t+1)}, \widetilde{\boldsymbol{\xi}}_i \rangle \right| \leq \Upsilon^{(t_1)} \cdot (1 + \epsilon). \tag{93}$$

By combining (92) and (93), we can arrive at

$$\Upsilon^{(t+1)} = \max_{j \in \{\pm 1\}, r \in [m]} \left\{ \max_{i \in [n]} \left| \langle \mathbf{w}_{j,r}^{(t+1)}, \boldsymbol{\xi}_i \rangle \right|, \max_{i \in \mathcal{W}} \left| \langle \mathbf{w}_{j,r}^{(t+1)}, \widetilde{\boldsymbol{\xi}}_i \rangle \right| \right\} \leq \Upsilon^{(t_1)} \cdot (1 + \epsilon).$$

Thus we have proved our induction statement for step $t + 1$. Repeating the induction completes the proof of Proposition 33. $\qquad \square$

### E.6.4    PROOF OF PROPOSITION 34

*Proof of Proposition 34.* We expand $\tilde{f}_{t_j(\bar{S}_{j,k})}$ as follows.

$$\tilde{f}_{t_j(\bar{S}_{j,k})} \leq \frac{1}{m} \sum_{r \in [m]} \sigma \left( \langle \mathbf{w}_{j,r}^{(t_j(\bar{S}_{j,k}-1))}, j\mathbf{u} \rangle \right) \cdot \left( 1 + \tilde{\eta} \left( 1 - \tilde{f}_{t_j(\bar{S}_{j,k}-1)} \right) \right)^2$$

$$+ \frac{1}{m} \sum_{r \in [m]} \sigma \left( \langle \mathbf{w}_{j,r}^{(t_j(\bar{S}_{j,k}-1))}, j\mathbf{v} \rangle \right) \cdot \left( 1 + \alpha\tilde{\eta} \left( 1 - \tilde{f}_{t_j(\bar{S}_{j,k}-1)} \right) \right)^2$$

$$- \frac{1}{m} \sum_{r \in [m]} \sigma \left( - \langle \mathbf{w}_{-j,r}^{(t_j(\bar{S}_{j,k}-1))}, -j\mathbf{u} \rangle \right) \cdot \left( 1 - \tilde{\eta} \left( 1 - \tilde{f}_{t_j(\bar{S}_{j,k}-1)} \right) \right)^2$$

$$- \frac{1}{m} \sum_{r \in [m]} \sigma \left( - \langle \mathbf{w}_{j,r}^{(t_j(\bar{S}_{j,k}-1))}, -j\mathbf{v} \rangle \right) \cdot \left( 1 - \alpha\tilde{\eta} \left( 1 - \tilde{f}_{t_j(\bar{S}_{j,k}-1)} \right) \right)^2$$

$$+ \left| \frac{1}{m} \sum_{r \in [m]} \sigma \left( \langle \mathbf{w}_{j,r}^{(t_j(\bar{S}_{j,k}))}, \boldsymbol{\xi}_{i_{t_j(\bar{S}_{j,k})}} \rangle \right) - \frac{1}{m} \sum_{r \in [m]} \sigma \left( \langle \mathbf{w}_{-j,r}^{(t_j(\bar{S}_{j,k}))}, \boldsymbol{\xi}_{i_{t_j(\bar{S}_{j,k})}} \rangle \right) \right|$$

$$\leq \frac{1}{m} \sum_{r \in [m]} \left( \sigma \left( \langle \mathbf{w}_{j,r}^{(t_j(\bar{S}_{j,k}-1))}, j\mathbf{u} \rangle \right) + \sigma \left( \langle \mathbf{w}_{j,r}^{(t_j(\bar{S}_{j,k}-1))}, j\mathbf{v} \rangle \right) \right.$$

$$\left. - \sigma \left( - \langle \mathbf{w}_{-j,r}^{(t_j(\bar{S}_{j,k}-1))}, -j\mathbf{u} \rangle \right) - \sigma \left( - \langle \mathbf{w}_{j,r}^{(t_j(\bar{S}_{j,k}-1))}, -j\mathbf{v} \rangle \right) \right) \cdot \left( 1 + \tilde{\eta} \left( 1 - \tilde{f}_{t_j(\bar{S}_{j,k}-1)} \right) \right)^2$$

$$+ \frac{1}{m} \sum_{r \in [m]} \left( \sigma \left( - \langle \mathbf{w}_{-j,r}^{(t_j(\bar{S}_{j,k}-1))}, -j\mathbf{u} \rangle \right) + \sigma \left( - \langle \mathbf{w}_{j,r}^{(t_j(\bar{S}_{j,k}-1))}, -j\mathbf{v} \rangle \right) \right) \cdot \left( 2\tilde{\eta} \left( 1 - \tilde{f}_{t_j(\bar{S}_{j,k}-1)} \right) \right)$$

$$+ \Upsilon^{(t_j(\bar{S}_{j,k}))^2}$$

$$\leq \tilde{f}_{t_j(\bar{S}_{j,k}-1)} \cdot \left( 1 + \tilde{\eta} \left( 1 - \tilde{f}_{t_j(\bar{S}_{j,k}-1)} \right) \right)^2 + 2E_{t_j(\bar{S}_{j,k}-1)} + \Upsilon^{(t_j(\bar{S}_{j,k}))^2}. \tag{94}$$

By Assumption 12 we know that $\tilde{f}_{t_j(\bar{S}_{j,k})} > 1 + \delta$. From the discussion in the proof of Lemma 19, we know that once

$$2E_{t_j(\bar{S}_{j,k}-1)} + \Upsilon^{(t_j(\bar{S}_{j,k}))^2} \leq \delta,$$

we have that

$$\tilde{f}_{t_j(\bar{S}_{j,k}-1)} \geq \frac{\tilde{\eta} + 2 - \sqrt{\tilde{\eta}^2 + 4\tilde{\eta}}}{2\tilde{\eta}},$$

and that

$$\tilde{f}_{t_j(\bar{S}_{j,k})} \leq \left( 1 + \tilde{\eta}(1 - \tilde{f}_{t_j(\bar{S}_{j,k}-1)}) \right)^2 + E_{t_j(\bar{S}_{j,k}-1)} + \Upsilon^{(t_j(\bar{S}_{j,k}))^2}$$

$$\leq \left( \tilde{\eta}/2 + \sqrt{\tilde{\eta}^2/4 + \tilde{\eta}} \right)^2 + E_{t_j(\bar{S}_{j,k}-1)} + \Upsilon^{(t_j(\bar{S}_{j,k}))^2}.$$

This finishes the proof of Proposition 34 $\qquad \square$

# F MULTIPLE TRAINING DATA CASE: SMALL LEARNING RATE REGIME

This section focuses on the multiple training data setup with a small learning rate.

Recall that $\mathcal{W}$ is the index set of training data points which lack the strong feature patch. By (3), the CNN weights are updated according to

$$
\begin{aligned}
\mathbf{w}_{j,r}^{(t+1)} = \mathbf{w}_{j,r}^{(t)} - \frac{j\eta}{m} \cdot \big(f(\mathbf{x}_{i_t}; \mathbf{W}^{(t)}) - y_{i_t}\big) \cdot \Big(&\sigma'(\langle \mathbf{w}_{j,r}^{(t)}, y_{i_t}\mathbf{u}\rangle) \cdot y_{i_t}\mathbf{u} \cdot \mathbf{1}\{i_t \notin \mathcal{W}\} \\
&+ \sigma'(\langle \mathbf{w}_{j,r}^{(t)}, y_{i_t}\mathbf{v}\rangle) \cdot y_{i_t}\mathbf{v} + \sigma'(\langle \mathbf{w}_{j,r}^{(t)} \cdot \boldsymbol{\xi}_{i_t}\rangle) \cdot \boldsymbol{\xi}_{i_t} \\
&+ \sigma'(\langle \mathbf{w}_{j,r}^{(t)} \cdot \widetilde{\boldsymbol{\xi}}_{i_t}\rangle) \cdot \widetilde{\boldsymbol{\xi}}_{i_t}, \cdot\mathbf{1}\{i_t \in \mathcal{W}\}\Big).
\end{aligned}
\tag{95}
$$

Subsequently, by (95) the update formulas of those inner products of interests are given by

$$
\langle \mathbf{w}_{j,r}^{(t+1)}, j\mathbf{u}\rangle = \langle \mathbf{w}_{j,r}^{(t)}, j\mathbf{u}\rangle + \frac{\eta\|\mathbf{u}\|_2^2}{m} \cdot \big(1 - y_{i_t}f(\mathbf{x}_{i_t}; \mathbf{W}^{(t)})\big) \cdot \sigma'(\langle \mathbf{w}_{j,r}^{(t)}, j\mathbf{u}\rangle jy_{i_t}) \cdot \mathbf{1}\{i_t \notin \mathcal{W}\},
$$

$$
\langle \mathbf{w}_{j,r}^{(t+1)}, j\mathbf{v}\rangle = \langle \mathbf{w}_{j,r}^{(t)}, j\mathbf{v}\rangle + \frac{\eta\|\mathbf{v}\|_2^2}{m} \cdot \big(1 - y_{i_t}f(\mathbf{x}_{i_t}; \mathbf{W}^{(t)})\big) \cdot \sigma'(\langle \mathbf{w}_{j,r}^{(t)} \cdot j\mathbf{v}\rangle jy_{i_t}),
$$

and

$$
\begin{aligned}
\langle \mathbf{w}_{j,r}^{(t+1)}, \boldsymbol{\xi}_i\rangle = \langle \mathbf{w}_{j,r}^{(t)}, \boldsymbol{\xi}_i\rangle + \frac{\eta \cdot jy_{i_t}}{m} \cdot \big(1 - y_{i_t}f(\mathbf{x}_{i_t}; \mathbf{W}^{(t)})\big) \cdot \Big(&\sigma'(\langle \mathbf{w}_{j,r}^{(t)} \cdot \boldsymbol{\xi}_{i_t}\rangle) \cdot \langle \boldsymbol{\xi}_{i_t}, \boldsymbol{\xi}_i\rangle \\
&+ \sigma'(\langle \mathbf{w}_{j,r}^{(t)} \cdot \widetilde{\boldsymbol{\xi}}_{i_t}\rangle) \cdot \langle \widetilde{\boldsymbol{\xi}}_{i_t}, \boldsymbol{\xi}_i\rangle \cdot \mathbf{1}\{i_t \in \mathcal{W}\}\Big).
\end{aligned}
$$

$$
\begin{aligned}
\langle \mathbf{w}_{j,r}^{(t+1)}, \widetilde{\boldsymbol{\xi}}_i\rangle = \langle \mathbf{w}_{j,r}^{(t)}, \widetilde{\boldsymbol{\xi}}_i\rangle + \frac{\eta \cdot jy_{i_t}}{m} \cdot \big(1 - y_{i_t}f(\mathbf{x}_{i_t}; \mathbf{W}^{(t)})\big) \cdot \Big(&\sigma'(\langle \mathbf{w}_{j,r}^{(t)}, \boldsymbol{\xi}_{i_t}\rangle) \cdot \langle \boldsymbol{\xi}_{i_t}, \widetilde{\boldsymbol{\xi}}_i\rangle \\
&+ \sigma'(\langle \mathbf{w}_{j,r}^{(t)}, \widetilde{\boldsymbol{\xi}}_{i_t}\rangle) \cdot \langle \widetilde{\boldsymbol{\xi}}_{i_t}, \widetilde{\boldsymbol{\xi}}_i\rangle \cdot \mathbf{1}\{i_t \in \mathcal{W}\}\Big), \quad i \in \mathcal{W}.
\end{aligned}
$$

For convenience, we also write $\ell_i^{(t)} = f(\mathbf{x}_i; \mathbf{W}^{(t)}) - y_i$ as the fitting residual.

The result of this section relies on the following conditions on the data model and the initialization.

**Assumption 35** (Conditions on hyperparameters). *Suppose that the following holds. For some $\epsilon \in (0, 1)$,*

1. *The weight initialization scale $\sigma_0 = \widetilde{\Theta}(\|\mathbf{u}\|_2^{-1})$;*

2. *Strong signal strength $\|\mathbf{u}\|_2 > \widetilde{\Omega}(m/\sqrt{n\epsilon}) \cdot \sigma_p\sqrt{d}$ and weak signal strength $\sigma_p\sqrt{d} \geq \widetilde{\Omega}(m/\sqrt{n\epsilon}) \cdot \|\mathbf{v}\|_2$;*

3. *The dimension $d$ satisfies $d = \Omega(\mathrm{polylog}(m, n))$.*

**Theorem 36** (Restatement of Proposition 6). *Under Assumption 35, choosing the learning rate $\eta \leq m/6\|\mathbf{u}\|_2^2$ small enough and $\epsilon' \in (0, 1)$, then with high probability $1 - 1/\mathrm{poly}(d)$, there exist*

$$
T^\dagger = \frac{4m}{\eta(1-\tau)(1-\rho)\|\mathbf{u}\|_2^2} \log\left(\frac{2\iota}{\sigma_0\|\mathbf{u}\|_2}\right), \quad T = T^\dagger + \left\lfloor\frac{Cm^3}{2\eta\epsilon\|\mathbf{u}\|_2^2}\right\rfloor, \quad T' = T + \left\lfloor\frac{\left\|\mathbf{W}^{(T)} - \mathbf{W}^\star\right\|_F^2}{2\eta\epsilon}\right\rfloor,
$$

*with $\tau$, $\iota$ defined in (97), (98) such that: (i) the average loss on samples $\mathcal{W}^c$ decreases to $3\epsilon$ over iterations $[T^\dagger, T]$, i.e.*

$$
\frac{1}{2n}\sum_{i \in \mathcal{W}^c} \min_{T^\dagger \leq t \leq T}\left\{y_i - f(\mathbf{x}_i; \mathbf{W}^{(t)})\right\}^2 \leq 3\epsilon,
$$

*(ii) average loss on samples $\mathcal{W}$ decreases to $3\epsilon'$ over iterations $[T, T']$, i.e.*

$$
\frac{1}{2n}\sum_{i \in \mathcal{W}} \min_{T \leq t \leq T'}\left(y_i - f(\mathbf{x}_i; \mathbf{W}^{(t)})\right)^2 \leq 3\epsilon',
$$

*(iii) the model does not learn weak signal $\mathbf{v}$ well enough even until $T'$, compared to initialization, i.e.*

$$
\max_{j,r}\left|\langle \mathbf{w}_{j \in \{\pm 1\}, r \in [m]}^{(t)}, \mathbf{v}\rangle\right| \leq 2\sqrt{2\log(16m/p)} \cdot \sigma_0\|\mathbf{v}\|_2, \quad \forall t \leq T'.
$$

In the multiple training data small learning rate regime, the dynamics go through three stages, which we characterize in Appendices F.1, F.2, and F.3, respectively. The following lemma plays an important role in the exponentially increasing stage (stage 1), for which we single it out here.

**Lemma 37** (Derivative lower bound). *For any $0 < \tau < 1$ to be tuned later, suppose at some time $t$ there holds*

$$\max_{j \in \{\pm 1\}, r \in [m]} \left\{ \left| \langle \mathbf{w}_{j,r}^{(t)}, \mathbf{u} \rangle \right|, \left| \langle \mathbf{w}_{j,r}^{(t)}, \mathbf{v} \rangle \right|, \max_{i \in [n]} \left| \langle \mathbf{w}_{j,r}^{(t)}, \boldsymbol{\xi}_i \rangle \right|, \max_{i \in \mathcal{W}} \left| \langle \mathbf{w}_{j,r}^{(t)}, \widetilde{\boldsymbol{\xi}}_i \rangle \right| \right\} \le \sqrt{\frac{\tau}{3}},$$

*then we can lower bound the fitting residual $-y\ell_i^{(t)} \ge 1 - \tau$ for every $i \in [n]$.*

*Proof of Lemma 37.* Plug into the CNN model definition (1), we have that

$$-y\ell_i^{(t)} = 1 - F_y(\mathbf{x}_i; \mathbf{W}^{(t)}) + F_{-y}(\mathbf{x}_i; \mathbf{W}^{(t)}) \ge 1 - F_y(\mathbf{x}_i; \mathbf{W}^{(t)}).$$

If $i \in \mathcal{W}^c$, we can upper bound $F_y(\mathbf{x}_i; \mathbf{W}^{(t)})$ further by

$$F_y(\mathbf{x}_i; \mathbf{W}^{(t)}) = \frac{1}{m} \sum_{r \in [m]} \sigma(\langle \mathbf{w}_{y,r}^{(t)}, y\mathbf{u} \rangle) + \sigma(\langle \mathbf{w}_{y,r}^{(t)}, y\mathbf{v} \rangle) + \sigma(\langle \mathbf{w}_{y,r}^{(t)}, \boldsymbol{\xi}_i \rangle)$$

$$\le \max_{r \in [m]} \left\{ \langle \mathbf{w}_{y,r}^{(t)}, y\mathbf{u} \rangle^2 + \langle \mathbf{w}_{y,r}^{(t)}, y\mathbf{v} \rangle^2 + \langle \mathbf{w}_{y,r}^{(t)}, \boldsymbol{\xi}_i \rangle^2 \right\} \le \tau.$$

Otherwise, if $i \in \mathcal{W}$, we also have

$$F_y(\mathbf{x}_i; \mathbf{W}^{(t)}) = \frac{1}{m} \sum_{r \in [m]} \sigma(\langle \mathbf{w}_{y,r}^{(t)}, \widetilde{\boldsymbol{\xi}}_i \rangle) + \sigma(\langle \mathbf{w}_{y,r}^{(t)}, y\mathbf{v} \rangle) + \sigma(\langle \mathbf{w}_{y,r}^{(t)}, \boldsymbol{\xi}_i \rangle)$$

$$\le \max_{r \in [m]} \left\{ \langle \mathbf{w}_{y,r}^{(t)}, \widetilde{\boldsymbol{\xi}}_i \rangle^2 + \langle \mathbf{w}_{y,r}^{(t)}, y\mathbf{v} \rangle^2 + \langle \mathbf{w}_{y,r}^{(t)}, \boldsymbol{\xi}_i \rangle^2 \right\} \le \tau.$$

Then it follows that $-y\ell^{(t)} \ge 1 - \tau$. $\qquad\qquad\square$

### F.1  STAGE 1. LEARNING STRONG SIGNAL EXPONENTIALLY FAST

In this stage, we mainly track the maximal inner product between $\mathbf{w}$ and the signal vectors $\mathbf{v}$ and $\mathbf{u}$, with extra attention to the maximal inner product between $\mathbf{w}$ and the noise vectors.

$$\Psi^{(t)} = \max_{j \in \{\pm 1\}, r \in [m]} \left| \langle \mathbf{w}_{j,r}^{(t)}, \mathbf{v} \rangle \right|, \quad \Phi^{(t)} = \max_{j \in \{\pm 1\}, r \in [m]} \left| \langle \mathbf{w}_{j,r}^{(t)}, \mathbf{u} \rangle \right|,$$

$$\Gamma_i^{(t)} = \max_{j \in \{\pm 1\}, r \in [m]} \left| \langle \mathbf{w}_{j,r}^{(t)}, \boldsymbol{\xi}_i \rangle \right|, \quad i \in [n],$$

$$\widetilde{\Gamma}_i^{(t)} = \max_{j \in \{\pm 1\}, r \in [m]} \left| \langle \mathbf{w}_{j,r}^{(t)}, \widetilde{\boldsymbol{\xi}}_i \rangle \right|, \quad i \in \mathcal{W}.$$

**Lemma 38** (First stage: noise). *Under the same conditions as Theorem 36, ever since initialization, at least until time*

$$T_+ := \frac{nm}{3\eta(4 + \rho)\sigma_p^2 d},$$

*there still holds that*

$$\max_{i \in [n]} \Gamma_i^{(t)} \le \sigma_0 \sigma_p \sqrt{d}, \quad \max_{i \in \mathcal{W}} \widetilde{\Gamma}_i^{(t)} \le \sigma_0 \sigma_p \sqrt{d}. \tag{96}$$

*Proof of Lemma 38.* For those inner products with noise vectors, $\forall i \in [n]$, the updating rules become

$$\left| \langle \mathbf{w}_{j,r}^{(t+1)}, \boldsymbol{\xi}_i \rangle \right| \le \left| \langle \mathbf{w}_{j,r}^{(t)}, \boldsymbol{\xi}_i \rangle \right| + \frac{\eta}{m} \cdot \left| \ell_{i_t}^{(t)} \right| \cdot \left( \sigma'(\langle \mathbf{w}_{j,r}^{(t)} \cdot \boldsymbol{\xi}_{i_t} \rangle) \cdot |\langle \boldsymbol{\xi}_{i_t}, \boldsymbol{\xi}_i \rangle| + \sigma'(\langle \mathbf{w}_{j,r}^{(t)} \cdot \widetilde{\boldsymbol{\xi}}_{i_t} \rangle) \cdot \left| \langle \widetilde{\boldsymbol{\xi}}_{i_t}, \boldsymbol{\xi}_i \rangle \right| \cdot \mathbf{1}\{i_t \in \mathcal{W}\} \right)$$

$$\le \left| \langle \mathbf{w}_{j,r}^{(t)}, \boldsymbol{\xi}_i \rangle \right| + \frac{6\eta}{m} \cdot \left( \left| \langle \mathbf{w}_{j,r}^{(t)}, \boldsymbol{\xi}_{i_t} \rangle \right| \cdot |\langle \boldsymbol{\xi}_{i_t}, \boldsymbol{\xi}_i \rangle| + \left| \langle \mathbf{w}_{j,r}^{(t)}, \widetilde{\boldsymbol{\xi}}_{i_t} \rangle \right| \cdot \left| \langle \widetilde{\boldsymbol{\xi}}_{i_t}, \boldsymbol{\xi}_i \rangle \right| \cdot \mathbf{1}\{i_t \in \mathcal{W}\} \right).$$

By taking maximum over $r \in [m]$, we conclude that

$$\Gamma_i^{(t+1)} \leq \Gamma_i^{(t)} + \frac{6\eta}{m} \cdot \left(\Gamma_{i_t}^{(t)} \cdot |\langle \boldsymbol{\xi}_{i_t}, \boldsymbol{\xi}_i \rangle| + \widetilde{\Gamma}_{i_t}^{(t)} \cdot \left|\langle \widetilde{\boldsymbol{\xi}}_{i_t}, \boldsymbol{\xi}_i \rangle\right| \cdot \mathbf{1}\{i_t \in \mathcal{W}\}\right), \quad \forall i \in [n].$$

Similarly, we also have that

$$\widetilde{\Gamma}_i^{(t+1)} \leq \widetilde{\Gamma}_i^{(t)} + \frac{6\eta}{m} \cdot \left(\Gamma_{i_t}^{(t)} \cdot \left|\langle \boldsymbol{\xi}_{i_t}, \widetilde{\boldsymbol{\xi}}_i \rangle\right| + \widetilde{\Gamma}_{i_t}^{(t)} \cdot \left|\langle \widetilde{\boldsymbol{\xi}}_{i_t}, \widetilde{\boldsymbol{\xi}}_i \rangle\right| \cdot \mathbf{1}\{i_t \in \mathcal{W}\}\right), \quad \forall i \in \mathcal{W}.$$

We then use induction to rigorously prove our conclusions. Firstly, (96) holds at time $t = 0$. Now suppose that (96) holds until some $\widetilde{T} < T_+$. Fixing some $i \in [n]$,

$$\begin{aligned}
\Gamma_i^{(\widetilde{T}+1)} &\leq \frac{6\eta\sigma_0\sigma_p\sqrt{d}}{m} \cdot \sum_{t=0}^{\widetilde{T}} \left(|\langle \boldsymbol{\xi}_{i_t}, \boldsymbol{\xi}_i \rangle| + \left|\langle \widetilde{\boldsymbol{\xi}}_{i_t}, \boldsymbol{\xi}_i \rangle\right| \cdot \mathbf{1}\{i_t \in \mathcal{W}\}\right) \\
&\leq \frac{6\eta\sigma_0\sigma_p\sqrt{d}}{m} \cdot \left(\frac{3\widetilde{T}\sigma_p^2 d}{2n} + 2\widetilde{T} \cdot (1+\rho) \cdot \sigma_p^2 \sqrt{d\log(4n^2/p)}\right) \\
&\leq \frac{3\eta\sigma_0\sigma_p\sqrt{d}(4+\rho)\widetilde{T}\sigma_p^2 d}{nm} \\
&\leq \sigma_0\sigma_p\sqrt{d}.
\end{aligned}$$

The first inequality is by induction hypothesis. The second inequality is because that there are at most $\widetilde{T}/n$ many $i_t$'s would equal $i$ and at most $\rho\widetilde{T}$ many $i_t$'s would be in $\mathcal{W}$, and we also use Lemma 9 to control the correlations between noise vectors. The third inequality is by $d \geq 16n^2 \log(4n^2/p)$ for Assumption 35, while the last inequality is due to $\widetilde{T} < T_+$. Similarly, we can also control $\widetilde{\Gamma}_i^{(t+1)}$ for some fixed $i \in \mathcal{W}$ as

$$\begin{aligned}
\widetilde{\Gamma}_i^{(\widetilde{T}+1)} &\leq \frac{6\eta\sigma_0\sigma_p\sqrt{d}}{m} \cdot \sum_{t=0}^{\widetilde{T}} \left(\left|\langle \boldsymbol{\xi}_{i_t}, \widetilde{\boldsymbol{\xi}}_i \rangle\right| + \left|\langle \widetilde{\boldsymbol{\xi}}_{i_t}, \widetilde{\boldsymbol{\xi}}_i \rangle\right| \cdot \mathbf{1}\{i_t \in \mathcal{W}\}\right) \\
&\leq \frac{6\eta\sigma_0\sigma_p\sqrt{d}}{m} \left(\frac{3\widetilde{T}\sigma_p^2 d}{2n} + 2\widetilde{T} \cdot (1+\rho) \cdot \sigma_p^2 \sqrt{d\log(4n^2/p)}\right) \\
&\leq \sigma_0\sigma_p\sqrt{d},
\end{aligned}$$

where the second inequality is because there are at most $\widetilde{T}/n$ many $i_t$'s would equal $i \in \mathcal{W}$. In conclusion, (96) holds at least until $T_+$. $\qquad\square$

In the following, we would take

$$\tau = \max\left\{2\sigma_0\|\mathbf{u}\|_2 \left(2\log(16m/p)\right)^{1/2 - \|\mathbf{v}\|_2^2/4(\sigma_p^2 d)}, 1 - \frac{6\sqrt{2}\sigma_p^2 d}{\|\mathbf{u}\|_2^2 \log(2/\sqrt{2\log(16m/p)})}\right\} \quad (97)$$

$$\iota = 2\sigma_0\|\mathbf{u}\|_2 \cdot \exp\left\{\frac{(1-\tau)(1-\rho)\|\mathbf{u}\|_2^2}{3(4+\rho)\sigma_p^2 d}\right\}. \quad (98)$$

By the conditions in Assumption 35 on $\|\mathbf{v}\|_2^2/\|\mathbf{u}\|_2^2$, we find $\tau, \iota$ both constant in $(0, 1)$.

**Lemma 39** (First stage: signal). *Under the same conditions as Theorem 36, there exists time*

$$T^\dagger = \frac{4m}{\eta(1-\tau)(1-\rho)\|\mathbf{u}\|_2^2} \log\left(\frac{2\iota}{\sigma_0\|\mathbf{u}\|_2}\right),$$

*such that: (i) the model learns strong signal to a constant level,*

$$\max_{r \in [m]} \langle \mathbf{w}_{j,r}^{(T^\dagger)}, j\mathbf{u} \rangle \geq \iota, \quad \forall j \in \{\pm 1\},$$

*(ii) compared to the random initialization, the model does not learn weak signal that much, i.e.,*

$$\max_{j \in \{\pm 1\}, r \in [m]} \left|\langle \mathbf{w}_{j,r}^{(T^\dagger)}, \mathbf{v} \rangle\right| \leq 2\sqrt{2\log(16m/p)} \cdot \sigma_0\|\mathbf{v}\|_2$$

*Proof of Lemma 39.* Firstly, we would find $\{\Psi^{(t)}, \Phi^{(t)}\}_{t \geq 0}$ having an exponentially growing upper bound. Recursively, we would have that

$$
\begin{aligned}
\Psi^{(t+1)} &\leq \Psi^{(t)} + \max_{j \in \{\pm 1\}, r \in [m]} \left| \frac{jy\eta}{m} \cdot \left( f(\mathbf{x}_i; \mathbf{W}^{(t)}) - y \right) \cdot \sigma'(\langle \mathbf{w}_{j,r}^{(t)}, y\mathbf{v} \rangle) \cdot \|\mathbf{v}\|_2^2 \right| \\
&= \Psi^{(t)} + \frac{\eta}{m} \cdot \left| \ell_i^{(t)} \right| \cdot \|\mathbf{v}\|_2^2 \cdot \max_{j \in \{\pm 1\}, r \in [m]} \sigma'(\langle \mathbf{w}_{j,r}^{(t)}, y\mathbf{v} \rangle) \\
&\leq \Psi^{(t)} + \frac{2\eta}{m} \cdot \left| \ell_i^{(t)} \right| \cdot \|\mathbf{v}\|_2^2 \cdot \Psi^{(t)} \\
&\leq \exp\left( \frac{6\eta \|\mathbf{v}\|_2^2}{m} \right) \cdot \Psi^{(t)}.
\end{aligned}
$$

Therefore, we have that

$$
\Psi^{(t)} \leq \exp\left( \frac{6\eta \|\mathbf{v}\|_2^2 t}{m} \right) \cdot \Psi^{(0)} \leq \exp\left( \frac{6\eta \|\mathbf{v}\|_2^2 t}{m} \right) \cdot \sqrt{2 \log(16m/p)} \cdot \sigma_0 \|\mathbf{v}\|_2 \tag{99}
$$

It follows similarly that

$$
\Phi^{(t)} \leq \exp\left( \frac{6\eta(1-\rho) \|\mathbf{u}\|_2^2 t}{m} \right) \cdot \Phi^{(0)} \leq \exp\left( \frac{6\eta(1-\rho) \|\mathbf{u}\|_2^2 t}{m} \right) \cdot \sqrt{2 \log(16m/p)} \cdot \sigma_0 \|\mathbf{u}\|_2 \tag{100}
$$

The extra factor $1 - \rho$ appears because only a $1 - \rho$ proportion of data points would contain $\mathbf{u}$, and therefore contribute to evolution of $\Phi^{(t)}$. Note that growing rates of these two bounds differ a lot due to the different magnitudes of $(1-\rho)\|\mathbf{u}\|_2$ and $\|\mathbf{v}\|_2$.

Our subsequent analysis illustrates that $\Phi^{(t)}$ can grow into a constant-level magnitude since strong signal $\mathbf{u}$ is significant enough. We can track how well our model learns $\mathbf{u}$ by

$$
A_1^{(t)} = \max_{r \in [m], i_t \notin \mathcal{W}} \langle \mathbf{w}_{1,r}^{(t)}, \mathbf{u} \rangle, \quad A_{-1}^{(t)} = \max_{r \in [m], i_t \notin \mathcal{W}} \langle \mathbf{w}_{-1,r}^{(t)}, -\mathbf{u} \rangle.
$$

By definition, $A_1^{(t)}, A_{-1}^{(t)} \leq \Phi^{(t)}$ also admits an exponentially upper bound in (100). For a certain $\tau \in (0, 1)$, due to the exponential upper bounds (99) and (100), $\max\{\Phi^{(t)}, \Psi^{(t)}\} \leq \sqrt{\tau/3}$ is true at least until

$$
T_1 = \frac{m}{6\eta(1-\rho)\|\mathbf{u}\|_2^2} \log\left( \frac{\sqrt{\tau/2}}{\sigma_0 \|\mathbf{u}\|_2 \sqrt{2\log(16m/p)}} \right).
$$

Moreover, since we have $(1-\rho)\|\mathbf{u}\|_2^2 \gg \sigma_p^2 d/n$ by Assumption 35, we also know $T_1 \leq T_+$ where $T_+$ comes from Lemma 38. Therefore

$$
\Gamma_i^{(t)} \leq \sigma_0 \sigma_p \sqrt{d} \leq \sqrt{\tau/3}, \quad \widetilde{\Gamma}_i^{(t)} \leq \sigma_0 \sigma_p \sqrt{d} \leq \sqrt{\tau/3}, \quad \forall t \leq T_1
$$

Consequently, until at least time $T_1$, we can use Lemma 37 to conclude that $-y_{i_t} \ell_{i_t}^{(t)} \geq 1 - \tau$, which enables lower bounding $A^{(t)}$ in the following.

The $i_t$-th sample would be used to update parameters, according to our multi-pass SGD updates (3). If $i_t \in \mathcal{W}$, then $\langle \mathbf{w}_{j,r}^{(t+1)}, j\mathbf{u} \rangle = \langle \mathbf{w}_{j,r}^{(t)}, j\mathbf{u} \rangle$ holds for any $j \in \{\pm 1\}$ and $r \in [m]$. If $i_t \notin \mathcal{W}$ but $y_{i_t} = -1$, then $\max_r \langle \mathbf{w}_{1,r}^{(t+1)}, \mathbf{u} \rangle = \max_r \langle \mathbf{w}_{1,r}^{(t)}, \mathbf{u} \rangle \geq 0$ since that neuron will not be activated. Otherwise, only if $i_t \notin \mathcal{W}$ and $y_{i_t} = 1$, the updating rule becomes

$$
\begin{aligned}
\langle \mathbf{w}_{1,r}^{(t+1)}, \mathbf{u} \rangle &= \langle \mathbf{w}_{1,r}^{(t)}, \mathbf{u} \rangle + \frac{\eta}{m} \cdot \left( -y_{i_t} \ell_{i_t}^{(t)} \right) \cdot \sigma'(\langle \mathbf{w}_{1,r}^{(t)}, \mathbf{u} \rangle) \cdot \|\mathbf{u}\|_2^2 \\
&\geq \langle \mathbf{w}_{1,r}^{(t)}, \mathbf{u} \rangle + \frac{2\eta(1-\tau)\|\mathbf{u}\|_2^2}{m} \cdot \max\left\{ \langle \mathbf{w}_{1,r}^{(t)}, \mathbf{u} \rangle, 0 \right\}.
\end{aligned}
$$

Take maximum over $r \in [m]$ to see that

$$
A_1^{(t+1)} \geq A_1^{(t)} + \frac{2\eta(1-\tau)\|\mathbf{u}\|_2^2}{m} \cdot A_1^{(t)} \geq \exp\left( \frac{\eta(1-\tau)\|\mathbf{u}\|^2}{m} \right) \cdot A_1^{(t)},
$$

where the last equality is by $1 + z \geq \exp(z/2)$ for any $0 \leq z \leq 2$. Consequently, when $t$ is large (larger than $n$), we would have

$$
\begin{aligned}
A_1^{(t)} &\geq \exp\left(\frac{\eta(1-\tau)\|\mathbf{u}\|_2^2}{m} \cdot \sum_{t' \leq t} \mathbf{1}\{i_{t'} \notin \mathcal{W}, y_{i_t} = 1\}\right) \cdot A_1^{(0)} \\
&\geq \exp\left(\frac{\eta(1-\tau)\|\mathbf{u}\|_2^2 \cdot (1-\rho) \cdot t}{4m}\right) \cdot \sigma_0 \|\mathbf{u}\|_2/2,
\end{aligned} \tag{101}
$$

at least until step $t \leq T_1$. We use the fact that $\sum_{t' \leq t} \mathbf{1}\{i_{t'} \notin \mathcal{W}, y_{i_t} = 1\} \geq (1-\rho)t/4$ because the sample labels are balanced (Lemma 7) and $1 - \rho$ proportion of samples come with the strong signal. In the same manner, we would have that

$$
A_{-1}^{(t)} \geq \exp\left(\frac{\eta(1-\tau)\|\mathbf{u}\|_2^2(1-\rho)t}{4m}\right) \cdot \sigma_0 \|\mathbf{u}\|_2/2. \tag{102}
$$

Define the time when $A_{\pm 1}^{(t)}$ both break $\iota$,

$$
T_2 = \frac{4m}{\eta(1-\tau)(1-\rho)\|\mathbf{u}\|_2^2} \log\left(\frac{2\iota}{\sigma_0\|\mathbf{u}\|_2}\right) \leq T_1,
$$

where the inequality is due to the scaling of $\iota$ upon $\tau$. Moreover, we also need that $T_2 \leq T_+$, where $T_+$ is the time that $\langle \mathbf{w}_{j,r}^{(t)}, \widetilde{\boldsymbol{\xi}}_i \rangle, \langle \mathbf{w}_{j,r}^{(t)}, \boldsymbol{\xi}_i \rangle$ remains in $\mathcal{O}(\sigma_0 \sigma_p \sqrt{d})$. And this requirement is also achieved by the selection of $\tau$ and $\iota$. Plugging $T_2$ into the exponential lower bounds (101) and (102), we can conclude that

$$
\Phi^{(T_2)} \geq A_{\pm 1}^{(T_2)} \geq \iota,
$$

which already grows up to a constant level magnitude by the time $T_2$. Lastly, plug the definition of $T_2$ to upper bound $\psi^{(T_2)}$ as

$$
\Psi^{(T_2)} \leq \exp\left(\frac{24\|\mathbf{v}\|_2^2}{(1-\tau)(1-\rho)\|\mathbf{u}\|_2^2} \log\left(\frac{2\iota}{\sigma_0\|\mathbf{u}\|_2}\right)\right) \cdot \sqrt{2\log(16m/p)} \cdot \sigma_0\|\mathbf{v}\|_2 \leq 2\sqrt{2\log(16m/p)} \cdot \sigma_0\|\mathbf{v}\|_2.
$$

In conclusion, by taking $T^\dagger = T_2$, this lemma is completely proved. $\qquad\square$

## F.2 STAGE 2. EXPLOITING STRONG SIGNAL

In the second stage, our lemmas suggest that before the model really learns the weak signal $\mathbf{v}$ or memorizes any noise vector, the model already fits a proportion $1 - \rho$ of the data points (i.e., strong data) by exploiting strong signal $\mathbf{u}$.

**Lemma 40** (Second stage). *Under the same conditions as Theorem 36, there exists time*

$$
T = T^\dagger + \left\lfloor \frac{Cm^3}{2\eta\epsilon\|\mathbf{u}\|_2^2} \right\rfloor
$$

*such that: (i) the average loss over iterations within this stage has decreased to $2\epsilon$, i.e.,*

$$
\frac{1}{2n} \sum_{i \in \mathcal{W}^c} \min_{T^\dagger \leq t \leq T} \left\{ y_i - f(\boldsymbol{x}_i; \mathbf{W}^{(t)}) \right\}^2 \leq 3\epsilon,
$$

*(ii) all through the training dynamics $0 \leq t \leq T$, there holds that*

$$
\max_{j \in \{\pm 1\}, r \in [m]} \left| \langle \mathbf{w}_{j,r}^{(t)}, \mathbf{v} \rangle \right| \leq 2\sqrt{2\log(16m/p)}\sigma_0 \cdot \|\mathbf{v}\|_2,
$$

*(iii) all through the training dynamics $0 \leq t \leq T$, there holds that*

$$
\max_{j \in \{\pm 1\}, r \in [m], i \in [n]} \left| \langle \mathbf{w}_{j,r}^{(t)}, \boldsymbol{\xi}_i \rangle \right| \leq \sigma_0 \sigma_p \sqrt{d}, \qquad \max_{j \in \{\pm 1\}, r \in [m], i \in \mathcal{W}} \left| \langle \mathbf{w}_{j,r}^{(t)}, \widetilde{\boldsymbol{\xi}}_i \rangle \right| \leq \sigma_0 \sigma_p \sqrt{d}.
$$

In studying the second stage, we firstly identify when the upper bound on $\langle \mathbf{w}_{j,r}^{(t)}, \mathbf{v} \rangle$ breaks and find that the conclusions of Lemma 39 still holds before that time.

**Lemma 41.** *Under the same conditions as Theorem 36, take $\eta \leq m/6\|\mathbf{u}\|_2^2$. There exists a time*

$$T^{\ddagger} = \frac{m}{6\eta\|\mathbf{v}\|_2^2} \log \left( \frac{\sqrt{\tau/2}}{\sigma_0 \|\mathbf{v}\|_2 \sqrt{2\log(16m/p)}} \right) \geq T^{\dagger}$$

*such that* (96) *and*

$$\max_{j\in\{\pm 1\}, r\in[m]} \left| \langle \mathbf{w}_{j,r}^{(t)}, \mathbf{v} \rangle \right| \leq 2\sqrt{2\log(16m/p)}\sigma_0\|\mathbf{v}\|, \quad \max_{r\in[m]} \langle \mathbf{w}_{j,r}^{(t)}, j\mathbf{u} \rangle \geq \iota/2, \quad \forall j \in \{\pm 1\},$$

*hold for any $T^{\dagger} \leq t \leq T^{\ddagger}$.*

*Proof of Lemma 41.* Firstly, we need to adopt the exponential upper bound derived in proving Lemma 39,

$$\Psi^{(t)} \leq \exp\left(\frac{6\eta\|\mathbf{v}\|_2^2 t}{m}\right) \cdot \Psi^{(0)} \leq \exp\left(\frac{6\eta\|\mathbf{v}\|_2^2 t}{m}\right) \cdot \sqrt{2\log(16m/p)} \cdot \sigma_0\|\mathbf{v}\|_2.$$

Then we naturally find that before $T^{\ddagger}$, it would always hold that

$$\max_{j\in\{\pm 1\}, r\in[m]} \left| \langle \mathbf{w}_{j,r}^{(t)}, \mathbf{v} \rangle \right| \leq 2\sqrt{2\log(16m/p)} \cdot \sigma_0\|\mathbf{v}\|_2.$$

Due to the conditions on $\|\mathbf{u}\|_2^2/\|\mathbf{v}\|_2^2$, $T^{\ddagger}$ is found to be much larger than $T^{\dagger}$. Then we proceed to prove the other assertion by induction. At time $t = T^{\dagger}$, the lower bound $\max_{j,r} \langle \mathbf{w}_{j,r}^{(t)}, j\mathbf{u} \rangle \geq \iota/2$ holds as a consequence of the previous lemma. Suppose it holds until time $t$. If $i_t \in \mathcal{W}$, then $\langle \mathbf{w}_{j,r}^{(t+1)}, j\mathbf{u} \rangle = \langle \mathbf{w}_{j,r}^{(t)}, j\mathbf{u} \rangle$ holds for any $j \in \{\pm 1\}, r \in [m]$. If $i_t \notin \mathcal{W}$ but $y_{i_t} = -1$, then $\max_{r\in[m]} \langle \mathbf{w}_{1,r}^{(t+1)}, \mathbf{u} \rangle = \max_{r\in[m]} \langle \mathbf{w}_{1,r}^{(t)}, \mathbf{u} \rangle \geq 0$ since that neuron will not be activated. Otherwise, if $i_t \notin \mathcal{W}$ and $y_{i_t} = 1$, consider the updating rule

$$\langle \mathbf{w}_{1,r}^{(t+1)}, \mathbf{u} \rangle = \langle \mathbf{w}_{1,r}^{(t)}, \mathbf{u} \rangle + \frac{\eta}{m} \cdot \left(1 - y_{i_t} f(\mathbf{x}_{i_t}; \mathbf{W}^{(t)})\right) \cdot \sigma'(\langle \mathbf{w}_{1,r}^{(t)}, \mathbf{u} \rangle) \cdot \|\mathbf{u}\|_2^2,$$

from which we find $\max_{r\in[m], i_t\notin\mathcal{W}} \langle \mathbf{w}_{y_i,r}^{(t+1)}, y_i\mathbf{u} \rangle \geq \max_{r\in[m], i_t\notin\mathcal{W}} \langle \mathbf{w}_{y_i,r}^{(t)}, y_i\mathbf{u} \rangle$ must hold if $y_{i_t} f(\mathbf{x}_{i_t}; \mathbf{W}^{(t)}) \leq 1$. Otherwise, once $y_{i_t} f(\mathbf{x}_{i_t}; \mathbf{W}^{(t)}) > 1$, it immediately follows that

$$1 < y_{i_t} f(\mathbf{x}_{i_t}; \mathbf{W}^{(t)}) = F_{y_{i_t}}(\mathbf{x}_{i_t}; \mathbf{W}^{(t)}) - F_{-y_{i_t}}(\mathbf{x}_{i_t}; \mathbf{W}^{(t)})$$

$$\leq F_{y_{i_t}}(\mathbf{x}_{i_t}; \mathbf{W}^{(t)}) = \frac{1}{m} \sum_{r\in[m]} \sigma(\langle \mathbf{w}_{1,r}^{(t)}, y_{i_t}\mathbf{u} \rangle) + \sigma(\langle \mathbf{w}_{1,r}^{(t)}, y_{i_t}\mathbf{v} \rangle) + \sigma(\langle \mathbf{w}_{1,r}^{(t)}, \xi_{i_t} \rangle)$$

$$\leq \max_{r\in[m]} \langle \mathbf{w}_{1,r}^{(t)}, \mathbf{u} \rangle^2 + \sigma_0^2 \|\mathbf{v}\|^2 + \sigma_0^2 \sigma_p^2 d.$$

Consequently, for the specific neuron $r^* = \arg\max_{r\in[m]} \langle \mathbf{w}_{1,r}^{(t)}, \mathbf{u} \rangle^2$, there holds

$$\langle \mathbf{w}_{1,r^*}^{(t+1)}, \mathbf{u} \rangle \geq \langle \mathbf{w}_{1,r^*}^{(t)}, \mathbf{u} \rangle - \frac{3\eta}{m} \cdot \langle \mathbf{w}_{1,r^*}^{(t)}, \mathbf{u} \rangle \cdot \|\mathbf{u}\|_2^2$$

$$\geq \left(1 - 8\log(16m/p) \cdot \sigma_0^2\|\mathbf{v}\|_2^2 - \sigma_0^2\sigma_p^2 d\right) \cdot \left(1 - \frac{3\eta}{m} \cdot \|\mathbf{u}\|_2^2\right)$$

$$\geq \frac{\iota}{2},$$

where the last inequality is enabled by taking

$$\eta \leq \frac{m}{6\|\mathbf{u}\|_2^2}, \quad \sigma_0 \leq \sqrt{\frac{1-\iota}{8\log(16m/p) \cdot \|\mathbf{v}\|_2^2 + \sigma_p^2 d}}.$$

Therefore, we find that $\max_{r\in[m]} \langle \mathbf{w}_{1,r}^{(t+1)}, \mathbf{u} \rangle \geq \iota/2$ must hold no matter what $i_t$ is. In the same way, one can also obtain $\max_{r\in[m]} \langle \mathbf{w}_{-1,r}^{(t+1)}, -\mathbf{u} \rangle \geq \iota/2$. By induction, the induction proof is complete. $\square$

Our subsequently analysis confirms that even before $T^\ddagger$, the model can already fit those data points with strong signal by exploiting $\mathbf{u}$. For the given $0 < \epsilon < 1$, define a reference point $\mathbf{W}^*$ as

$$\mathbf{w}_{j,r}^* = \frac{4m(1+\epsilon)}{\iota} \cdot \frac{j\mathbf{u}}{\|\mathbf{u}\|_2^2}, \quad j \in \{\pm 1\}, r \in [m]. \tag{103}$$

**Lemma 42.** *Under the same condition as the previous lemma, for all $T^\dagger \le t \le T^\ddagger$, there holds*

$$y_i \langle \nabla f(\mathbf{x}_i; \mathbf{W}^{(t)}), \mathbf{W}^* \rangle \ge 2 \cdot (1+\epsilon)$$

*for any $i \notin \mathcal{W}$.*

*Proof of Lemma 42.* Recall that the definition of CNN in (1) and that $\mathbf{u} \perp \mathrm{span}(\mathbf{v}, \boldsymbol{\xi}_i)$. We have that

$$
\begin{aligned}
y_i \langle \nabla f(\mathbf{x}_i; \mathbf{W}^{(t)}), \mathbf{W}^* \rangle &= \frac{1}{m} \sum_{j \in \{\pm 1\}, r \in [m]} \sigma'(\langle \mathbf{w}_{j,r}^{(t)}, y_i \mathbf{u} \rangle) \cdot \langle \mathbf{w}_{j,r}^*, y_i \mathbf{u} \rangle \\
&= \sum_{j \in \{\pm 1\}, r \in [m]} \sigma'(\langle \mathbf{w}_{j,r}^{(t)}, y_i \mathbf{u} \rangle) \cdot \frac{4(1+\epsilon)}{\iota} \\
&\ge \max_{r \in [m]} \langle \mathbf{w}_{y_i, r}^{(t)}, y_i \mathbf{u} \rangle \cdot \frac{4(1+\epsilon)}{\iota} \\
&\ge 2 \cdot (1+\epsilon),
\end{aligned}
$$

where the last inequality is by $\max_r \langle \mathbf{w}_{j,r}^{(t)}, j\mathbf{u} \rangle \ge \iota/2$ for any $j \in \{\pm 1\}$ as shown by the previous lemma. $\square$

**Lemma 43.** *Continued from the previous setting, for $T^\dagger \le t \le T^\ddagger$, if $i_t \notin \mathcal{W}$, there holds*

$$\|\mathbf{W}^{(t)} - \mathbf{W}^*\|_F^2 - \|\mathbf{W}^{(t+1)} - \mathbf{W}^*\|_F^2 \ge 2\eta \big(f(\mathbf{x}_{i_t}; \mathbf{W}^{(t)}) - y_{i_t}\big)^2 - 2\eta \epsilon^2.$$

*Proof of Lemma 43.* Firstly we expand the difference by

$$
\begin{aligned}
&\|\mathbf{W}^{(t)} - \mathbf{W}^*\|_F^2 - \|\mathbf{W}^{(t+1)} - \mathbf{W}^*\|_F^2 \\
&= 2\eta \langle \ell_{i_t}^{(t)} \nabla f(\mathbf{x}_{i_t}; \mathbf{W}^{(t)}), \mathbf{W}^{(t)} - \mathbf{W}^* \rangle - \eta^2 \cdot \left|\ell_{i_t}^{(t)}\right|^2 \cdot \|\nabla f(\mathbf{x}_{i_t}; \mathbf{W}^{(t)})\|_F^2. \tag{104}
\end{aligned}
$$

Since the neural network $f(\mathbf{x}; \mathbf{W})$ is 2-homogeneous in $\mathbf{W}$ due to the activation function $\sigma(z) = \max\{z, 0\}^2$, we have that

$$\langle \nabla f(\mathbf{x}; \mathbf{W}^{(t)}), \mathbf{W}^{(t)} \rangle = 2f(\mathbf{x}; \mathbf{W}^{(t)}).$$

Stack these observations into the first term of previous difference expansion to obtain

$$
\begin{aligned}
\langle \ell_{i_t}^{(t)} \nabla f(\mathbf{x}_{i_t}; \mathbf{W}^{(t)}), \mathbf{W}^{(t)} - \mathbf{W}^* \rangle &= \ell_{i_t}^{(t)} \cdot \big(2f(\mathbf{x}_{i_t}; \mathbf{W}^{(t)}) - \langle \nabla f(\mathbf{x}_{i_t}; \mathbf{W}^{(t)}), \mathbf{W}^* \rangle\big) \\
&= 2\ell_{i_t}^{(t)} \cdot \big(f(\mathbf{x}_{i_t}; \mathbf{W}^{(t)}) - y_{i_t}\big) + \ell_{i_t}^{(t)} \cdot y_{i_t} \cdot \big(2 - y_{i_t} \langle \nabla f(\mathbf{x}_{i_t}; \mathbf{W}^{(t)}), \mathbf{W}^* \rangle\big).
\end{aligned}
$$

Note that the first term is exactly $2(f(\mathbf{x}_{i_t}; \mathbf{W}^{(t)}) - y_{i_t})^2$. As for the second term, since $i_t \notin \mathcal{W}$, we need to plug in Lemma 42 to see $2 - y_{i_t} \langle \nabla f(\mathbf{x}_{i_t}; \mathbf{W}^{(t)}), \mathbf{W}^* \rangle \le -2\epsilon < 0$, so that

$$\left| \ell_{i_t}^{(t)} \cdot y_{i_t} \cdot \big(2 - y_{i_t} \langle \nabla f(\mathbf{x}_{i_t}; \mathbf{W}^{(t)}), \mathbf{W}^* \rangle\big) \right| \le \frac{1}{2}\big(\ell_{i_t}^{(t)}\big)^2 + 2\epsilon^2.$$

As a result, we would know that

$$\langle \ell_{i_t}^{(t)} \nabla f(\mathbf{x}_{i_t}; \mathbf{W}^{(t)}), \mathbf{W}^{(t)} - \mathbf{W}^* \rangle \ge \frac{3}{2}(f(\mathbf{x}_{i_t}; \mathbf{W}^{(t)}) - y_{i_t})^2 - 2\epsilon^2.$$

Next, an upper bound on the second order term $\eta^2 \|\nabla L(\mathbf{W}^{(t)})\|_F^2$ is given by

$$\eta^2 \cdot \left|\ell_{i_t}^{(t)}\right|^2 \cdot \|\nabla f(\mathbf{x}_{i_t}; \mathbf{W}^{(t)})\|_F^2$$

$$= \eta^2 \cdot \ell_{i_t}^{(t)2} \cdot \left( \|\mathbf{u}\|_2^2 \cdot \sum_{j \in \{\pm 1\}, r \in [m]} \sigma'(\langle \mathbf{w}_{j,r}^{(t)}, y\mathbf{u}\rangle)^2 \right.$$

$$\left. + \|\mathbf{v}\|_2^2 \cdot \sum_{j \in \{\pm 1\}, r \in [m]} \sigma'(\langle \mathbf{w}_{j,r}^{(t)}, y\mathbf{v}\rangle)^2 + \|\boldsymbol{\xi}_{i_t}\|_2^2 \cdot \sum_{j \in \{\pm 1\}, r \in [m]} \sigma'(\langle \mathbf{w}_{j,r}^{(t)}, \boldsymbol{\xi}_{i_t}\rangle)^2 \right)$$

$$\leq \mathcal{O}(\max\{\|\mathbf{u}\|_2^2, \|\mathbf{v}\|_2^2, \|\boldsymbol{\xi}_{i_t}\|_2^2\}) \cdot \eta^2 \cdot \ell_{i_t}^{(t)2},$$

since the dynamics of inner products $\langle \mathbf{w}_{j,r}^{(t)}, y\mathbf{u}\rangle, \langle \mathbf{w}_{j,r}^{(t)}, y\mathbf{v}\rangle, \langle \mathbf{w}_{j,r}^{(t)}, \boldsymbol{\xi}_i\rangle$ are well bounded by $\mathcal{O}(1)$. Via scaling $\eta \cdot \mathcal{O}(\max\{\|\mathbf{u}\|_2^2, \|\mathbf{v}\|_2^2, \|\boldsymbol{\xi}_{i_t}\|_2^2\}) \leq 1$, we would know $\eta^2|\ell_{i_t}^{(t)}|^2\|\nabla f(\mathbf{x}_{i_t}; \mathbf{W}^{(t)})\|_F^2 \leq \eta\ell_{i_t}^{(t)2}$. Eventually, continued from (104), we can completely prove this lemma. $\qquad \square$

**Lemma 44.** *Continued from the previous setting, for $T^\dagger \leq t \leq T^\ddagger$, if $i_t \in \mathcal{W}$, there holds*

$$\|\mathbf{W}^{(t)} - \mathbf{W}^*\|_F^2 - \|\mathbf{W}^{(t+1)} - \mathbf{W}^*\|_F^2 \geq -C\eta\sigma_0^2 \cdot \left(\|\mathbf{v}\|_2^2 + \sigma_p^2 d\right). \qquad (105)$$

*Proof of Lemma 44.* Same as the last lemma, from the SGD setting, we have that

$$\|\mathbf{W}^{(t)} - \mathbf{W}^*\|_F^2 - \|\mathbf{W}^{(t+1)} - \mathbf{W}^*\|_F^2$$

$$= 2\eta\langle \ell_{i_t}^{(t)}\nabla f(\mathbf{x}_{i_t}; \mathbf{W}^{(t)}), \mathbf{W}^{(t)} - \mathbf{W}^*\rangle - \eta^2 \cdot \left|\ell_{i_t}^{(t)}\right|^2 \cdot \|\nabla f(\mathbf{x}_{i_t}; \mathbf{W}^{(t)})\|_F^2, \quad (106)$$

and from the 2-homogeneity, it follows that

$$\langle \nabla f(\mathbf{x}_{i_t}; \mathbf{W}^{(t)}), \mathbf{W}^{(t)}\rangle = 2f(\mathbf{x}_{i_t}; \mathbf{W}^{(t)}).$$

Since $i_t \in \mathcal{W}$, every $\nabla_{\mathbf{w}_{j,r}} f(\mathbf{x}_{i_t}; \mathbf{W}^{(t)})$ is in $\mathrm{span}(\mathbf{v}, \boldsymbol{\xi}_{i_t}, \widetilde{\boldsymbol{\xi}}_{i_t}) \perp \mathbf{u}$, so

$$\langle \nabla f(\mathbf{x}_{i_t}; \mathbf{W}^{(t)}), \mathbf{W}^*\rangle = 0.$$

As a result, the first term in (106) can be bounded by

$$\left|2\eta\langle \ell_{i_t}^{(t)}\nabla f(\mathbf{x}_{i_t}; \mathbf{W}^{(t)}), \mathbf{W}^{(t)} - \mathbf{W}^*\rangle\right|$$

$$= 4\eta \cdot \left|\left(f(\mathbf{x}_{i_t}; \mathbf{W}^{(t)}) - y_{i_t}\right) \cdot f(\mathbf{x}_{i_t}; \mathbf{W}^{(t)})\right|$$

$$\leq \frac{12\eta}{m} \cdot \left|\sum_{j \in \{\pm 1\}, r \in [m]} \sigma(\langle \mathbf{w}_{j,r}^{(t)}, \widetilde{\boldsymbol{\xi}}_{i_t}\rangle) + \sum_{j \in \{\pm 1\}, r \in [m]} \sigma(\langle \mathbf{w}_{j,r}^{(t)}, y_{i_t}\mathbf{v}\rangle) + \sum_{j \in \{\pm 1\}, r \in [m]} \sigma(\langle \mathbf{w}_{j,r}^{(t)}, \boldsymbol{\xi}_{i_t}\rangle)\right|$$

$$\leq \mathcal{O}\left(\eta\sigma_0^2\|\mathbf{v}\|_2^2 + \eta\sigma_0^2\sigma_p^2 d\right).$$

We can also deal with the second term in (106) by

$$\eta^2 \cdot \left|\ell_{i_t}^{(t)}\right|^2 \cdot \|\nabla f(\mathbf{x}_{i_t}; \mathbf{W}^{(t)})\|_F^2$$

$$\leq \eta^2 \cdot \ell_{i_t}^{(t)2} \cdot \left( \|\widetilde{\boldsymbol{\xi}}_{i_t}\|_2^2 \cdot \sum_{j \in \{\pm 1\}, r \in [m]} \sigma'(\langle \mathbf{w}_{j,r}^{(t)}, \widetilde{\boldsymbol{\xi}}_{i_t}\rangle)^2 \right.$$

$$\left. + \|\mathbf{v}\|_2^2 \cdot \sum_{j \in \{\pm 1\}, r \in [m]} \sigma'(\langle \mathbf{w}_{j,r}^{(t)}, y_{i_t}\mathbf{v}\rangle)^2 + \|\boldsymbol{\xi}_{i_t}\|_2^2 \cdot \sum_{j \in \{\pm 1\}, r \in [m]} \sigma'(\langle \mathbf{w}_{j,r}^{(t)}, \boldsymbol{\xi}_{i_t}\rangle)^2 \right)$$

$$\leq \mathcal{O}\left(\eta^2(\sigma_0^4\|\mathbf{v}\|_2^4 + \sigma_0^4\sigma_p^4 d^2)\right),$$

where the last inequality is due to $\ell_{i_t}^{(t)}$ being $\mathcal{O}(1)$. Since we already take $\eta \leq m/6\|\mathbf{u}\|_2^2$, the second term of (106) is ignorable compared to the first term of (106). Therefore, we can conclude (105). $\qquad \square$

Equipped with Lemmas 41, 42, 43, and 44, we are ready to prove the main lemma of second stage.

*Proof of Lemma 40.* Continued from Lemmas 43 and 44, for any $t \geq T^\dagger$, it holds that

$$\frac{1}{t - T^\dagger + 1} \sum_{s=T^\dagger}^{t} \mathbf{1}\{i_s \notin \mathcal{W}\} \cdot \frac{1}{2} \big(f(\mathbf{x}_{i_t}; \mathbf{W}^{(t)}) - y_{i_t}\big)^2$$

$$\leq \frac{\|\mathbf{W}^{(T^\dagger)} - \mathbf{W}^*\|_F^2}{2\eta(t - T^\dagger + 1)} + \epsilon^2(1 - \rho) + C\sigma_0^2(\|\mathbf{v}\|_2^2 + \sigma_p^2 d)\rho.$$

Before proceeding to scale time $t$, it is helpful to decompose $\|\mathbf{W}^{(T^\dagger)} - \mathbf{W}^*\|_F^2$ and have an upper bound,

$$\|\mathbf{W}^{(T^\dagger)} - \mathbf{W}^*\|_F^2 = \sum_{j \in \{\pm 1\}, r \in [m]} \frac{\langle \mathbf{w}_{j,r}^{(T^\dagger)} - \mathbf{w}_{j,r}^*, \mathbf{u}\rangle^2}{\|\mathbf{u}\|_2^2} + \frac{\langle \mathbf{w}_{j,r}^{(T^\dagger)} - \mathbf{w}_{j,r}^*, \mathbf{v}\rangle^2}{\|\mathbf{v}\|_2^2} + \left\|\mathbf{P}_{\boldsymbol{\xi}, \widetilde{\boldsymbol{\xi}}}(\mathbf{w}_{j,r}^{(T^\dagger)} - \mathbf{w}_{j,r}^*)\right\|_2^2$$

$$+ \left\|\left(\mathbf{I}_d - \frac{\mathbf{v}\mathbf{v}^\top}{\|\mathbf{v}\|_2^2} - \frac{\mathbf{u}\mathbf{u}^\top}{\|\mathbf{u}\|_2^2} - \mathbf{P}_{\boldsymbol{\xi}, \widetilde{\boldsymbol{\xi}}}\right)(\mathbf{w}_{j,r}^{(T^\dagger)} - \mathbf{w}_{j,r}^*)\right\|_2^2$$

$$\leq \sum_{j \in \{\pm 1\}, r \in [m]} \frac{2\langle \mathbf{w}_{j,r}^{(T^\dagger)}, \mathbf{u}\rangle^2 + 2\langle \mathbf{w}_{j,r}^*, \mathbf{u}\rangle^2}{\|\mathbf{u}\|_2^2} + \frac{\langle \mathbf{w}_{j,r}^{(T^\dagger)}, \mathbf{v}\rangle^2}{\|\mathbf{v}\|_2^2} + \left\|\mathbf{P}_{\boldsymbol{\xi}, \widetilde{\boldsymbol{\xi}}}\mathbf{w}_{j,r}^{(T^\dagger)}\right\|^2$$

$$+ \left\|\left(\mathbf{I}_d - \frac{\mathbf{v}\mathbf{v}^\top}{\|\mathbf{v}\|_2^2} - \frac{\mathbf{u}\mathbf{u}^\top}{\|\mathbf{u}\|_2^2} - \mathbf{P}_{\boldsymbol{\xi}, \widetilde{\boldsymbol{\xi}}}\right)(\mathbf{w}_{j,r}^{(T^\dagger)} - \mathbf{w}_{j,r}^*)\right\|_2^2, \tag{107}$$

where $\mathbf{P}_{\boldsymbol{\xi}, \widetilde{\boldsymbol{\xi}}}$ denotes the projection matrix onto linear space $\text{span}_{i \in [n], i' \in \mathcal{W}}(\boldsymbol{\xi}_i, \widetilde{\boldsymbol{\xi}}_{i'})$. In these derivations, we exploit the fact that $\mathbf{w}^*$ is parallel to $\mathbf{u}$, and the gradient steps only updates $\mathbf{w}$ along the directions of $\mathbf{u}, \mathbf{v}$. Recall that by Lemma 41,

$$\max_{j \in \{\pm 1\}, r \in [m]} \langle \mathbf{w}_{j,r}^{(T^\dagger)}, j\mathbf{u}\rangle = \Omega(1), \quad \max_{j \in \{\pm 1\}, r \in [m]} \left|\langle \mathbf{w}_{j,r}^{(T^\dagger)}, \mathbf{v}\rangle\right| = \widetilde{\mathcal{O}}(\sigma_0\|\mathbf{v}\|_2), \quad \|\mathbf{w}_{j,r}^{(0)}\|_2 = \widetilde{\mathcal{O}}(\sigma_0\sqrt{d}),$$

$$\max_{j \in \{\pm 1\}, r \in [m], i \in [n]} \left|\langle \mathbf{w}_{j,r}^{(T^\dagger)}, \boldsymbol{\xi}_i\rangle\right| = \widetilde{\mathcal{O}}(\sigma_0\sigma_p\sqrt{d}),$$

the leading term in (107) is $\sum_{j \in \{\pm 1\}, r \in [m]} \langle \mathbf{w}_{j,r}^*, \mathbf{u}\rangle^2 / \|\mathbf{u}\|_2^2$. Therefore, we would conclude that $\|\mathbf{W}^{(T^\dagger)} - \mathbf{W}^*\|_F^2 \leq Cm^3/\|\mathbf{u}\|_2^2$. As a result, the average loss after iterations $T^\dagger$ can be bounded by

$$\frac{1}{t - T^\dagger + 1} \sum_{s=T^\dagger}^{t} \mathbf{1}\{i_s \notin \mathcal{W}\} \cdot \frac{1}{2}\big(f(\mathbf{x}_{i_t}; \mathbf{W}^{(t)}) - y_{i_t}\big)^2$$

$$\leq \frac{Cm^3}{2\eta\|\mathbf{u}\|_2^2(t - T^\dagger + 1)} + \epsilon^2(1 - \rho) + C\sigma_0^2(\|\mathbf{v}\|_2^2 + \sigma_p^2 d)\rho.$$

Then choose $T = T^\dagger + \lfloor Cm^3/(2\eta\epsilon\|\mathbf{u}\|_2^2)\rfloor$ as stated in Lemma 40. Since $\|\mathbf{u}\|_2^2/\|\mathbf{v}\|_2^2 \geq \widetilde{\Omega}(m^2)$, we can verify that $T \leq T^\ddagger$ where $T^\ddagger$ is given in Lemma 41 until when the weak signal cannot be fully learned. Moreover, we also have $T \leq T_+$ where $T_+$ is given in Lemma 38 when the noise is not memorized.

In conclusion, via scaling $\sigma_0^2 \leq \epsilon/(C\rho(\|\mathbf{v}\|_2^2 + \sigma_p^2 d))$, the final output would be

$$\frac{1}{2n} \sum_{i \in \mathcal{W}^c} \min_{T^\dagger \leq t \leq T} \big\{y_i - f(\mathbf{x}_i; \mathbf{W}^{(t)})\big\}^2 \leq \frac{1}{t - T^\dagger + 1} \sum_{s=T^\dagger}^{t} \mathbf{1}\{i_s \notin \mathcal{W}\} \cdot \frac{1}{2}\big(f(\mathbf{x}_{i_t}; \mathbf{W}^{(t)}) - y_{i_t}\big)^2$$

$$\leq \epsilon + \epsilon^2(1 - \rho) + C\sigma_0^2(\|\mathbf{v}\|_2^2 + \sigma_p^2 d)\rho$$

$$\leq 3\epsilon,$$

ending the proof of Lemma 40. $\qquad\square$

### F.3 STAGE 3. MEMORIZING NOISE

After the second stage, the model already fits those data points with strong signal by exploiting $\mathbf{u}$. Subsequently, in the following third stage, the residual $\ell_i^{(t)}$ for $i \in \mathcal{W}^c$ would remain quite small, preventing the model from learning $\mathbf{u}$.

On the contrary, since $f(\mathbf{x}_i; \mathbf{W}^{(t)}) = \widetilde{\mathcal{O}}(\sigma_0^2)$ is still far from its label $y_i$ for each sample $i \in \mathcal{W}$ without the strong signal $\mathbf{u}$. Therefore, the weight vectors would still evolve in the directions perpendicular to $\mathbf{u}$. In Assumption 35, the ratio between the weak signal $\mathbf{v}$ and the typical noise norm $\sigma_p \sqrt{d}$ is scaled by

$$\frac{\|\mathbf{v}\|_2}{\sigma_p \sqrt{d}} \leq \widetilde{\mathcal{O}}\left(\frac{1}{\sqrt{n}}\right). \tag{108}$$

Therefore, the model will eventually interpolates the whole dataset by memorizing noise vectors $(\widetilde{\boldsymbol{\xi}}_i, \boldsymbol{\xi}_i), i \in \mathcal{W}$, Now we define a new reference point $\mathbf{W}^\star$ by

$$\mathbf{w}_{j,r}^\star = \mathbf{w}_{j,r}^* + \frac{4m(1+\epsilon')}{\iota} \cdot \left(\sum_{i \in \mathcal{W}} \mathbf{1}\{y_i = j\} \cdot \frac{\boldsymbol{\xi}_i}{\|\boldsymbol{\xi}_i\|_2} + \mathbf{1}\{y_i = j\} \cdot \frac{\widetilde{\boldsymbol{\xi}}_i}{\|\widetilde{\boldsymbol{\xi}}_i\|_2}\right), \quad j \in \{\pm 1\}, r \in [m],$$

where $\mathbf{w}_{j,r}^*$ defined in (103) is the reference point we used in the second stage. The following lemma is an adaptation of Theorem 4.4 of Cao et al. (2022) onto SGD with square loss.

**Lemma 45** (Third stage). *Under the same conditions as Theorem 36, for some $\epsilon' \in (0, 1)$, let*

$$T' = T + \left\lfloor \frac{\left\|\mathbf{W}^{(T)} - \mathbf{W}^\star\right\|_F^2}{2\eta\epsilon} \right\rfloor,$$

*where $T$ is the end of the second stage in Lemma 40. Then we would have that*

$$\max_{j \in \{\pm 1\}, r \in [m]} \left|\langle \mathbf{w}_{j,r}^{(t)}, \mathbf{v} \rangle\right| \leq \widetilde{\mathcal{O}}(\sigma_0 \|\mathbf{v}\|_2),$$

*even until $t \leq T'$. But the whole dataset has already been interpolated during this interval,*

$$\frac{1}{2\rho n} \sum_{i \in \mathcal{W}} \min_{T \leq t \leq T'} \left\{y_i - f(\mathbf{x}_i; \mathbf{W}^{(t)})\right\}^2 \leq 3\epsilon'.$$

*Proof of Lemma 45.* As the closing stage of the training dynamics, the evolution dynamics during this interval is straightforward based on all techniques developed in Appendices D and F. Inner products

$$\left\{\langle \mathbf{w}_{j,r}^{(t)}, \boldsymbol{\xi}_i \rangle, \langle \mathbf{w}_{j,r}^{(t)}, \widetilde{\boldsymbol{\xi}}_i \rangle\right\}_{i \in \mathcal{W}}$$

would firstly go through a substage in which they exponentially increase to a constant level (just as stage 1). And then the model will fit all samples indexed by $\mathcal{W}$ by memorizing these noise vectors in polynomial time. All through this interval, $\max_{j \in \{\pm 1\}, r \in [m]} |\langle \mathbf{w}_{j,r}^{(t)}, \mathbf{v} \rangle|$ would stay $\widetilde{\mathcal{O}}(\sigma_0 \|\mathbf{v}\|)$ due to the scale of $\|\mathbf{v}\|_2 / (\sigma_p \sqrt{d})$ in (108). A detailed proof is omitted here for readability. $\square$

Combine Lemmas 39, 40 and 45 to obtain the full version of Theorem 36.

