# OpenReview forum: "Benign Oscillation of Stochastic Gradient Descent with Large Learning Rate"
_ICLR.cc/2024/Conference — ICLR 2024 poster_

### Official Review · Reviewer_yxwb · 2023-10-26

**Soundness:** 2 fair
**Presentation:** 2 fair
**Contribution:** 1 poor
**Rating:** 3
**Confidence:** 4

**Summary:**

This paper studies feature learning when the learning rate is large, showing that when there is a strong oscillation in the training, only data points with weak signals are learned

**Strengths:**

I think the message and the insight is sufficiently novel and relevant -- how large learning helps "selecting" a subset of data points to be learned.

The data generation model of strong and weak patches is also quite interesting and can serve as a good starting point for future works

Also, the signal to noise ratio of the data and gradient plays an important role in the proof and discussion and, therefore, I think the result does catch some level of essence of SGD training in neural networks. Although, on this point, I think the authors should compare with the study of how signal-to-noise ratio affects the learning at a large learning rate in (1) https://arxiv.org/abs/2107.11774 and (2) https://arxiv.org/abs/2303.13093

**Weaknesses:**

In my opinion, what prevents me from recommending this paper is that the problem setting feels too artificial. The following are the specific problems in my opinion

1. The problem setting. First of all, what is a ReLU^2 activation function? I have never seen this. Is ReLU more essential to the proof or is the quadratic effect more essential to the proof? Does the result hold for ReLU? Does the result hold for the quadratic activation? This is unanswered in the paper. To me, this activation neither feels theoretically appealing nor practically relevant

2. Problem setting 2. The second layer is not trained. Essentially, we understand that training with one layer is very different from training multiple layers simultaneously. For example, see https://arxiv.org/abs/2205.11787

3. Problem setting 3. The model architecture -- the authors refer to the model as a "CNN," which just does not feel right to me. I doubt if the majority of readers would agree that this model is indeed a CNN.

In summary, the point is the same, the problem setting is not sufficiently simple to be considered theoretically essential, nor is it sufficiently realistic. This significnatly limits the relevance and significance of the results obtained

**Questions:**

See the weakness section

---

> ### Author Response · Authors · 2023-11-15
> **Response to Reviewer yxwb (Part 1)**
>
> Thanks very much for your review! We now address all your concerns and questions for our work in the following.
>
> **Q1: What is a $\rm ReLU^2$ activation function? Is $\rm ReLU$ more essential to the proof or is the quadratic effect more essential to the proof? Does the result hold for $\rm ReLU$? Does the result hold for the quadratic activation? This is unanswered in the paper. To me, this activation neither feels theoretically appealing nor practically relevant.**
>
> **A1:** First, we clarify that the $\rm ReLU^q$-style activation (or ReLU with a polynomial smoothing) is actually widely adopted in existing works on deep learning theory, see e.g., [1, 2, 3, 4, 5, 6, 7] and the references therein. We followed the setup of this line of works for our study on large learning rate NN training. Therefore, we respectfully disagree with your comment that this activation is not theoretically appealing.
>
> Moreover, we need the $\rm ReLU^2$ activation to make the loss Hessian vary locally throughout the training process, which is important to enable the benign oscillation phenonemon we aim to study. This is consistent with the well-known edge-of-stability work [8, 9, 10], where they show that SGD with different learning rates tend to find the solutions with different Hessians. In contrast, with $\rm ReLU$ activation, we get only a locally quadratic objective and the loss sharpness will remain locally unchanged for our model, then the SGD/GD may lack the ability to explore unreached region but will be stuck at a local region (i.e., it will either smoothly converge or oscillate locally).
>
> Finally, our analysis can still work if we use a pure quadratic activation and the key proof techniques will be similar. The major reason of using ReLU activations is to be aligned with the existing line of related research.
>
> **References:**
>
> [1] Zhong, Kai, et al. "Recovery Guarantees for One-hidden-layer Neural Networks." International Conference on Machine Learning. PMLR, 2017.
>
> [2] Lyu, Kaifeng, and Jian Li. "Gradient Descent Maximizes the Margin of Homogeneous Neural Networks." International Conference on Learning Representations, 2019.
>
> [3] Fan, Fenglei, Jinjun Xiong, and Ge Wang. "Universal Approximation with Quadratic Deep Networks." Neural Networks 124 (2020): 383-392.
>
> [4] Cao, Yuan, et al. "Benign Overfitting in Two-layer Convolutional Neural Networks." Advances in neural information processing systems 35, 2022.
>
> [5] Shen, Ruoqi, Sébastien Bubeck, and Suriya Gunasekar. "Data Augmentation as Feature Manipulation." In International conference on machine learning, pp. 19773-19808. PMLR, 2022.
>
> [6] Allen-Zhu, Zeyuan, and Yuanzhi Li. "Towards Understanding Ensemble, Knowledge Distillation and Self-Distillation in Deep Learning." The Eleventh International Conference on Learning Representations, 2023.
>
> [7] Zou, Difan, et al. "Understanding the Generalization of Adam in Learning Neural Networks with Proper Regularization." The Eleventh International Conference on Learning Representations, 2023.
>
> [8] Cohen, Jeremy, et al. "Gradient Descent on Neural Networks Typically Occurs at the Edge of Stability." International Conference on Learning Representations, 2020.
>
> [9] Arora, Sanjeev, Zhiyuan Li, and Abhishek Panigrahi. "Understanding Gradient Descent on the Edge of Stability in Deep Learning." International Conference on Machine Learning. PMLR, 2022.
>
> [10] Wang, Zixuan, Zhouzi Li, and Jian Li. "Analyzing Sharpness Along GD Trajectory: Progressive Sharpening and Edge of Stability." Advances in Neural Information Processing Systems 35, 2022.

---

> ### Author Response · Authors · 2023-11-15
> **Response to Reviewer yxwb (Part 2)**
>
> **Q2: Why only train one layer? Essentially, we understand that training with one layer is very different from training multiple layers simultaneously. For example, see [1].**
>
> **A2**: We only train the hidden layer because for our model since this procedure is *sufficient* to characterize the key difference between the dynamics and the generalization properties between small learning rate training and large learning rate training, which is the main focus of our paper.
>
> Moreover, we point out that the result for one-layer setup in [1] you mentioned does not apply to our problem, since the so called training with one layer setup in [1] considers the neural tangent kernel (NTK) regime with linear models. In contrast, our model focuses on the feature learning perspective with quadratic activation (nonlinear), so that we are able to show the following three types of training dynamics:
>
>  - when the learning rate $\eta$ is small, the dynamic converges smoothly to a nearby local minimum;
>  - when the learning rate $\eta$ is reasonably larger, the oscillation happens during which the model gradually learns all useful patterns from data by learning the weak signal;
>  - when the learning rate $\eta$ is too large, despite that we have not directly shown in our paper, it is obvious to see the dynamic will diverge.
>
> Then this three phase dynamic is quite similar to the non-linear dynamic shown in [1] for the two layer neural quadratic model case. Intuitively, the reason is that the $\rm ReLU^2$ activation we adopted has already provided the source of nonlinearity we want, which gives a locally varying loss Hessian and enables the benign oscillation under reasonably large learning rate training. With this in hand, we have been able to show the benefits of large learning rate training and the mechanism behind it
>
> It is indeed interesting to consider training a multiple layer neural network. But that complicates the analysis and may not change the key message we would like to convey. So we consider only training the hidden layer in this work. We will add the related work [1] and more discussions on this in the revision.
>
> **References:**
>
> [1] Zhu, L., Liu, C., Radhakrishnan, A., & Belkin, M. (2022). Quadratic Models for Understanding Neural Network Dynamics. arXiv preprint arXiv:2205.11787.
>
> **Q3: The authors refer to the model as a "CNN", which just does not feel right to me. I doubt if the majority of readers would agree that this model is indeed a CNN.**
>
> **A3:** This specific style of model has been widely adopted and named as "CNN" in a sequence of pivotal works, to name a few, see [1, 2, 3, 4, 5, 6, 7]. This specific model is a surrogate of the real-world CNN architecture and is convenient for theoretical study. More specifically, each $\mathbf w\_{j,r}$ in the hidden layer $\sigma(\langle\mathbf{w}\_{j,r},\mathbf{x}\rangle)$ can be regarded as a filter of a CNN, and different patches of the data share the same filter because of the "weight sharing" principle of the CNN architecture.
>
> **References:**
>
> [1] Du, Simon S., Jason D. Lee, and Yuandong Tian. "When is a Convolutional Filter Easy to Learn?" International Conference on Learning Representations, 2018.
>
> [2] Du, Simon, et al. "Gradient Descent Learns One-hidden-layer CNN: Don’t Be Afraid of Spurious Local Minima." International Conference on Machine Learning. PMLR, 2018.
>
> [3] Cao, Yuan, and Quanquan Gu. "Tight Sample Complexity of Learning One-hidden-layer Convolutional Neural Networks." Advances in Neural Information Processing Systems 32, 2019.
>
> [4] Shen, Ruoqi, Sébastien Bubeck, and Suriya Gunasekar. "Data Augmentation as Feature Manipulation." In International conference on machine learning, pp. 19773-19808. PMLR, 2022.
>
> [5] Allen-Zhu, Zeyuan, and Yuanzhi Li. "Towards Understanding Ensemble, Knowledge Distillation and Self-Distillation in Deep Learning." The Eleventh International Conference on Learning Representations, 2023.
>
> [6] Zou, Difan, et al. "Understanding the Generalization of Adam in Learning Neural Networks with Proper Regularization." The Eleventh International Conference on Learning Representations, 2023.
>
> [7] Cao, Yuan, et al. "Benign Overfitting in Two-layer Convolutional Neural Networks." Advances in neural information processing systems 35, 2022.

---

> > ### Author Response · Authors · 2023-11-15
> > **Response to Reviewer yxwb (Part 3)**
> >
> > **Q4: Related works on large learning rate training under different signal-to-noise ratio [1, 2].**
> >
> > **A4:** Thanks for pointing out! These are definitely nice related works and we would like to add discussions with them in revision. In short, these works stand at the perspective of loss landscape of neural network optimization under large learning rates. In contrast, we aim to explain the better generalization of large learning rate training through the lens of feature learning, which provides a different viewpoint.
> >
> > **References:**
> >
> > [1] Ziyin, L., Li, B., Simon, J. B., & Ueda, M. (2021). SGD with a Constant Large Learning Rate Can Converge to Local Maxima. arXiv preprint arXiv:2107.11774.
> >
> > [2] Ziyin, L., Li, B., Galanti, T., & Ueda, M. (2023). The Probabilistic Stability of Stochastic Gradient Descent. arXiv preprint arXiv:2303.13093.

---

> ### Author Response · Authors · 2023-11-20
> **Response to Reviewer yxwb**
>
> Dear reviewer yxwb, we thank you again for your time reviewing our paper! We have dealt with all your concerns regarding our problem setup including the model architecture and the training protocol, and we have added new references to our draft according to **Q2** and **Q4** (see Appendix A). As the end of author-review discussion period is approaching, we would appreciate it if you can kindly let us know whether your previous concerns have been fully addressed. We are happy to answer any further questions. As you also mentioned that our work conveys sufficiently novel message and insight for large learning rate NN training, if you think our work is worthy of a higher score, we would be immensely grateful!

---

> > ### Comment · Reviewer_yxwb · 2023-11-23
> > **Review part 2**
> >
> > Thanks for the detailed reply.
> >
> > While I feel more positive about the work, here is my main concern. It seems to me that the most essential component in the proof is that the activation is quadratic (and not that it is ReLU). The authors (perhaps) mention that the proof can be extended to the case of ReLU^q type activations, but it appears to me that the extension only works if $q$ is strictly larger than 1. Namely, the theory only applies to superlinear activation functions -- which almost never works in practice. More importantly, it **cannot** be applied to linear or sublinear types of activations.
> >
> > In summary, the paper shows that benign oscillation exists in activations that practically fails and there is no benign oscillation in activations that practically works, which contradicts real-life observations. Thus, let me submit one additional question here: can the authors prove that for $q\leq 1$, ReLU^q also exhibits benign oscillation?

---

> > > ### Author Response · Authors · 2023-11-23
> > > **Response 2 to Reviewer yxwb (Part 1)**
> > >
> > > Thank you very much for your response to our reply! We address your concerns in the following.
> > >
> > > First, we again emphasize that $\mathrm{ReLU}^q$ activation with $q\ge 2$ is widely used in previous works on the analysis of neural networks training, see [1-7]. Our primary consideration of using $\mathrm{ReLU}^2$ is to admit a dynamic with locally varying Hessian, wheras training **one-layer** $\mathrm{ReLU}$ network does not. And our setting of training-only-one-layer with $\mathrm{ReLU}^2$ activation is sufficient to demonstrate the rationale of benign oscillation.
> > >
> > > Further, we clarify that we have **never** claimed "benign oscillation fails in activation that practically works". Instead, as is widely observed in deep learning practice, large learning rate oscillating training can lead to better generalization, for which we have also demonstrated via training ResNet18 on Cifar10 dataset in our paper (page 2). Our simple training-only-one-layer model serves as the starting point for understanding more complicated case of training all the multiple layers with standard $\mathrm{ReLU}$ activation.
> > >
> > > To be more concrete, for example, it is very likely that we can extend our theory to training two layer $\mathrm{ReLU}$ activation neural network. Let's consider a single data and single neuron setup to illustrate the idea. Consider $f(\mathbf{x};\mathbf{w},\mathbf{h}) = h_1\sigma(\langle\mathbf{w},\mathbf{u}\rangle) + h_2\sigma(\langle \mathbf{w},\mathbf{v}\rangle)$ trying to predict $y=1$, where $\mathbf{h}$ and $\mathbf {w}$ are both trainable parameters,  $\mathbf{u}$ is the strong signal and $\mathbf{v}$ is the weak signal. Consider when ReLU is activated by parameters $\mathbf{w}^{(t)}$, we can then calculate that
> > > $$
> > >     (1): h_1^{(t+1)}\cdot\sigma(\langle\mathbf{w}^{(t+1)},\mathbf{u}\rangle) = \left(1+\eta^2{{\ell^\prime}^{(t)}}^2\|\|\mathbf{u}\|\|_2^2\right)\cdot h_1^{(t)}\cdot\sigma(\langle\mathbf{w}^{(t)},\mathbf{u}\rangle) + \eta{\ell^\prime}^{(t)}\cdot\left({h_1^{(t)}}^2\|\|\mathbf{u}\|\|_2^2+\sigma(\langle\mathbf{w}^{(t)},\mathbf{u}\rangle)^2\right)
> > > $$
> > >
> > > $$
> > >     (2): h_2^{(t+1)}\cdot\sigma(\langle\mathbf{w}^{(t+1)},\mathbf{v}\rangle) = \left(1+\eta^2{{\ell^\prime}^{(t)}}^2\|\|\mathbf{v}\|\|_2^2\right)\cdot h_2^{(t)}\cdot\sigma(\langle\mathbf{w}^{(t)},\mathbf{v}\rangle) + \eta{\ell^\prime}^{(t)}\cdot\left({h_2^{(t)}}^2\|\|\mathbf{v}\|\|_2^2+\sigma(\langle\mathbf{w}^{(t)},\mathbf{v}\rangle)^2\right)
> > > $$
> > > where ${\ell^{\prime}}^{(t)} = 1 - f(\mathbf{x};\mathbf{w}^{(t)},\mathbf{h}^{(t)})$. We are particularly interested in these two terms $h_1^{(t)}\cdot\sigma(\langle\mathbf{w}^{(t)},\mathbf{u}\rangle)$ and $h_2^{(t)}\cdot\sigma(\langle\mathbf{w}^{(t)},\mathbf{v}\rangle)$ because they represent the learning of the strong and weak signals respectively, similar to the $\langle\mathbf{w}^{(t+1)},\mathbf{u}\rangle$ and $\langle\mathbf{w}^{(t+1)},\mathbf{v}\rangle$ considered in our paper. We note that such an update formula bears great similarity with the update formula of $\langle\mathbf{w}^{(t+1)},\mathbf{u}\rangle$ and $\langle\mathbf{w}^{(t+1)},\mathbf{v}\rangle$ we considered in our work (see Eq (8) in Appendix C.1). Specifically, (see the next part of response)

---

> ### Author Response · Authors · 2023-11-23
> **Response 2 to Reviewer yxwb (Part 2)**
>
> - (Large learning rate case) From the update (1) regarding the strong signal $\mathbf{u}$, by using the same argument as in the illustrative deduction in Section 3, we can show that the oscillation of $f$ around $1$ will accumulate the loss derivatives under large learning rate, that is, $\sum_{t}{\ell^{\prime}}^{(t)}$ is increasing (see the first conclusion of Lemma 17 and its proof). Applying this fact to the update formula (2) regarding the weak signal $\mathbf{v}$, we are able to show that $h_2^{(t)}\cdot\sigma(\langle\mathbf{w}^{(t)},\mathbf{v}\rangle)$ can keep increasing, which follows from the same proof of the second conclusion of Lemma 17. This demonstrates the benefits of large learning rate oscillating training in helping the model to learn the weak features.
> - (Small learning rate case) In contrast, training two layer $\mathrm{ReLU}$ activation NN with small learning rate cannot learn the weak signal. One can consider the same two-phase analysis as in our paper. In Phase 1, with a small initialization (the initialization scalings for $\mathbf{h}$ and $\mathbf{w}$ need to be similar to enable learning both layers), the NN is trying to learn both signals simultaneously and the growth of both signals are at least exponentially fast in early stages (note that
> ${{h_1^{(t)}}^2\|\|\mathbf{u}\|\|_2^2+\sigma(\langle\mathbf{w}^{(t)},\mathbf{u}\rangle)^2\ge h_1^{(t)}\sigma(\langle\mathbf{w}^{(t)},\mathbf{u}\rangle)\|\|\mathbf{u}\|\|_2}$ and similar result can hold for $h_2^{(t)}\sigma(\langle\mathbf{w}^{(t)},\mathbf{v}\rangle)$). However, due to the difference in the strength of the two signals, when Phase 1 quickly ends, the NN only learns the strong signal but makes little progress in learning the weak signal. Next, in Phase 2, we can prove that the dynamic smoothly converges to a local minima by exploiting the learned strong signal before it can make sufficient progress in learning the weak signal. In combination of Phase 1 & 2, we can show that training the two-layer NN with $\mathrm{ReLU}$ activation fails to learn the weak signal under small learning rate.
>
> Thus, there is still a division of generalization between large learning rate training and small learning rate training if we consider training both layers with standard $\rm ReLU$ activation. This shows the potential extendibility of our proof of the benign oscillation phenomenon, at least for training two-layer $\rm ReLU$ neural networks. We will add the discussion on the potential extensions to more general settings in the revision.
>
> **References:**
>
> [1] Zhong, Kai, et al. "Recovery Guarantees for One-hidden-layer Neural Networks." International Conference on Machine Learning. PMLR, 2017.
>
> [2] Lyu, Kaifeng, and Jian Li. "Gradient Descent Maximizes the Margin of Homogeneous Neural Networks." International Conference on Learning Representations, 2019.
>
> [3] Fan, Fenglei, Jinjun Xiong, and Ge Wang. "Universal Approximation with Quadratic Deep Networks." Neural Networks 124 (2020): 383-392.
>
> [4] Cao, Yuan, et al. "Benign Overfitting in Two-layer Convolutional Neural Networks." Advances in neural information processing systems 35, 2022.
>
> [5] Shen, Ruoqi, Sébastien Bubeck, and Suriya Gunasekar. "Data Augmentation as Feature Manipulation." In International conference on machine learning, pp. 19773-19808. PMLR, 2022.
>
> [6] Allen-Zhu, Zeyuan, and Yuanzhi Li. "Towards Understanding Ensemble, Knowledge Distillation and Self-Distillation in Deep Learning." The Eleventh International Conference on Learning Representations, 2023.
>
> [7] Zou, Difan, et al. "Understanding the Generalization of Adam in Learning Neural Networks with Proper Regularization." The Eleventh International Conference on Learning Representations, 2023.

---

### Official Review · Reviewer_ahqJ · 2023-10-30

**Soundness:** 3 good
**Presentation:** 4 excellent
**Contribution:** 3 good
**Rating:** 6
**Confidence:** 3

**Summary:**

The authors study the effect of large learning rate SGD on a prediction task with a one hidden layer CNN network. More specifically, they study a binary classification problem with weak and strong signal patches and theoretically argue that there is a generalization separation between small and large learning rate SGD training:
- for small learning rate, SGD is unable to learn the weak signal
- for large learning rate, SGD is able to learn the weak signal

They argue that the oscillatory behavior of large learning rate SGD is a crucial feature of this fact.
They also illustrate these results with experiments.

**Strengths:**

The study of SGD with large learning rates is believed to be crucial to understand the good learning properties of neural networks. Yet, its analysis is in many cases very difficult to handle: the oscillations, the movement due to noise are technically challenging to analyse.

In this perpective, the authors do a very good job providing a setup for which they 'prove' that such a dynamical behavior leads SGD to good generalization.

On top of this, I find the paper very well written, the setup very clear and the explanation with the toy model on Section 3 very good.

**Weaknesses:**

In my opinion, here is a weakness of the paper:

- the authors claim to perfectly prove that the oscillatory behavior of the SGD iterates lead it to learn the weak signal, but they need to assume that the iterate oscillate in the first place. While they show that this oscillation is *consistent* with their hypothesis, it could be great that the authors comment a bit more of this necessity to assume this. How difficult would it be to remove this hypothesis ? Is the problem to control that the iterates do not diverge ? Or on the contrary that they do no converge locally ?

- If this is the second option, could the authors add some (bounded) label noise at each iteration to show this? This is what is explained to be done in a series of paper like *Label noise (stochastic) gradient descent implicitly solves the Lasso for quadratic parametrisation* L. Pillaud-Vivien et al., COLT, 2022. Could the authors comment on this?

**Questions:**

- On top of the questions raised above, I really would like to understand whether the setup described above has can also guide on what appears to be a simple problem like diagonal linear networks where the oscillations lead to sparse features.

Does the analysis of the authors shed light onto this problem ?

- Also, it appears that the initialization is very small ($d^{-1/2}$ with a large $d$). Can the authors comment on the necessity of such an unusual initialization ?

---

> ### Author Response · Authors · 2023-11-15
> **Response to Reviewer ahqj (Part 1)**
>
> Thank you very much for your detailed review and your support of our work! We address all your concerns and questions in the following.
>
> **Q1: While the authors show that the oscillation is consistent with their hypothesis, it could be great that the authors comment a bit more of this necessity to assume this. How difficult would it be to remove this hypothesis? Is the problem to control that the iterates do not diverge? Or on the contrary that they do not converge locally? If this is the second option, could the authors add some (bounded) label noise at each iteration to show this?**
>
> **A1:** Thanks for your questions! We will explain the necessity of making this assumption in the following. In short, it is intuitively possible to derive a sufficient condition that enables the stable oscillation. However, proving a rigorous and clean sufficient condition for the multiple-data setting would be highly complicated. We thus resort to summarizing this specific mode of oscillation as an assumption in order to better present the key idea and finding of our work.
>
> Now we answer your questions in detail.
>
> We first explain why we need this assumption. As we illustrate in the single training data setup (Section 3), the oscillation (in terms of the model parameter) tend **not** to cancel with each other, but to have a strict positive value, which turns out to be the driving force of the weak signal learning. *Intuitively*, the amount of the accumulation of oscillation will determine the amount of weak signal learning:
> $$
> \langle \mathbf{w}\_{y, r}^{(t\_1)},y \mathbf{v} \rangle  \geq  \langle \mathbf{w}\_{y,r}^{(t\_0)}, y\mathbf{v}\rangle\cdot\exp\bigg(\underbrace{\Omega\bigg(\sum_{t = t_0}^{t_1}\big(1 - y\cdot f(\mathbf{x};\mathbf{W}^{(t)})\big)\bigg)}_{(\star)\geq 0\text{ by Section 3}}\bigg)
> $$
> Therefore, it boils down to mathematically characterize how fast the oscillation accumulation $(\star)$ would grow, which is **not** simply to only control the iterates from divergence or to prove no local convergence. What we want is a control over the magnitude of the oscillation (i.e., $\delta$) so that the summation $(\star)$ can be lower bounded.
>
> Intuitively, for the single training data setup, one can achieve this by properly scaling the initialization and the learning rate. i.e., controlling the learning rate by using the loss smoothness parameters at the initialization and the nearby local minima. Then, as the magnitude of the oscillation can be expressed as a joint function of the loss smoothness, the learning rate, and the initialization, we can then control the magnitude via properly scaling the learning rate and the initialization so as to give a sufficient condition for certain oscillation magnitude.
>
> However, in the general multiple data setup, the training dynamic when oscillation happens could be very difficult to precisely characterize, and the specific modes of oscillation could be quite sensitive to the choice of the learning rate and initialization (different LR and initialization can result in different modes of oscillation and may require different analysis). To this end, we took the cleanest and least troublesome way by assuming that the magnitude of the oscillation is uniformly bounded away by a some $\delta>0$, which then leads to the linear accumulation we show in the paper (e.g., Eq. (7) in Section 3). This saves us from diving into the explicit mathematical characterization of the oscillation dynamic and better explain the key message regrading the benefits of oscillation we would like to convey.

---

> > ### Author Response · Authors · 2023-11-15
> > **Response to Reviewer ahqj (Part 2)**
> >
> > **Q2: Does the analysis of the authors can shed light onto this problem like diagonal linear networks where the oscillations lead to sparse features?**
> >
> > **A2:** This is an interesting direction to explore! As the diagonal linear network also has nonlinearity, it gives a locally varying sharpness which is necessary for our theory to apply.
> >
> > More specifically, it turns out that our theoretical analysis not only indicates that the weak signal is properly learned (i.e., for $\langle \mathbf{w}\_{y,r},y \mathbf{v} \rangle>0$, it increases), but can also be leveraged to prove that all the negative parts (i.e., $\langle \mathbf{w}\_{-y,r},y \mathbf{v} \rangle>0$ and $\langle \mathbf{w}\_{-y,r},y \mathbf{u} \rangle>0$) are "unlearned", that is, they are pushed towards zero (we did not explicitly prove this because an upper bound of these negative parts are sufficient for our analysis, see Lemma 16 or Lemma 30). Therefore, once we show this, we can actually conclude that the model not only learns the weak signal, but also implicitly learns a "sparse" solution to some extent. Also, we believe that learning more useful features, which is the outcome of large learning rate SGD, can sortly suppress the noise memorization, since the more noisy model could lead to higher sharpness than the model that largely learns features to fit the training data. This may also connect to the sparsity of the learned model.
> >
> > Now that the linear diagonal network shares similar nonlinearity as our model, we think our analysis can have an implication to the problem you mentioned. We will add more discussions and highlight the recent works on diagonal linear networks in the revision.
> >
> > **Q3: Why using the small initialization?**
> >
> > **A3:** The $d^{-1/2}$ coefficient is to ensure that the initialization is smaller than the oscillation magnitude $\delta$ (at least in the high dimensional region). A direct benefit of this is that by Theorem 2 (one data) or Theorem 5 (multiple data), one can immediately conclude that the weak signal is guaranteed to be learned to the scale much higher than its initialization, improving over the small learning rate training regime. Also, small initialization is also assumed in the original edge-of stability paper[1], as they need to guarantee "progressive sharpening", where the sharpness at the initialization should be smaller than the sharpness at the oscillation phase.
> >
> > So the whole thing we want is an initialization that is not too large so that it is smaller than the scale of oscillation. Any other conditions on initialization would also work if this is guaranteed, e.g., it can be relaxed to $d^{-\alpha}$ for some $\alpha>0$. Also, in our numerical demonstrations, we reproduce the benign oscillation phenomenon we predict without this somehow unusually small learning rate.
> >
> > We will add a discussion on our choice of initialization scaling in the revision.
> >
> > **References:**
> >
> > [1] Cohen, Jeremy, et al. "Gradient Descent on Neural Networks Typically Occurs at the Edge of Stability." International Conference on Learning Representations, 2020.

---

> ### Author Response · Authors · 2023-11-20
> **Response to Reviewer ahqj**
>
> Dear reviewer ahqj, we thank you again for your time reviewing our paper and your support of our work! We have revised our draft according to **Q2** (see Appendix A) and **Q3** (see Appendix C). As the end of author-review discussion period is approaching, we appreciate it if you can kindly let us know whether your previous concerns have been fully addressed. We are happy to answer any further questions. Thanks!

---

### Official Review · Reviewer_1fN5 · 2023-11-05

**Soundness:** 4 excellent
**Presentation:** 3 good
**Contribution:** 3 good
**Rating:** 8
**Confidence:** 3

**Summary:**

This paper studies the generalization benefit of using large learning rates in stochastic gradient descent. Specifically, in the setting of a two-layer convolutional neural network, it is shown that although using a large learning rate causes the loss value to oscillate, it indeed further enables SGD to do feature learning. The main result is proven under the assumption of a feature-noise data generation model, where the strong and weak features are separated by the $\ell_2$ norm. Empirical evaluations are also provided to support the theoretical results.

**Strengths:**

- The paper is well written and easy to follow, and the authors have explained the theoretical intuition clearly through the example of a single training data in Section 3.

- The generalization benefit of large learning rates is an important topic in the community. This paper presents an interesting result in this direction, though the feature-noise data generation model seems somewhat artificial.

- The theoretical analysis in this paper is very solid.

**Weaknesses:**

- It is not clear if the feature-noise data generation model is a natural model that contains separate strong feature, weak feature, and noise.

- The network structure seems to be tailored to the feature-noise data generation model, as it applies a separate weight vector to each of the patch. Although the authors claimed that the main results can be further generalized, but it is not immediately clear how.

- The oscillating condition that $|y_{i_t} f(\mathbf{x}_{i_t};\mathbf{W}^{(t)} - 1|\geq \delta$ is proposed as an assumption, rather than being proved.

**Questions:**

1. How to justify the feature-noise data general model? E.g., is this a natural assumption for real-world dataset like CIFAR-10?

2. The authors discussed some necessary conditions of Assumption 4 on the loss oscillation. Are there any comprehensible sufficient conditions?

3. The authors mentioned a bit about the edge-of-stability regime. Can the authors expand a bit more on this? For example, how is the oscillation assumption related to EoS?

---

> ### Author Response · Authors · 2023-11-15
> **Response to Reviewer 1fN5 (Part 1)**
>
> Thank you very much for your appreciation of our work! We address all your concerns and questions in the following.
>
> **Q1: How to justify the feature-noise data general model? E.g., is this a natural assumption for real-world dataset like CIFAR-10?**
>
> **A1:** The original version of this multi-patch data model can be found in the pivotal work in feature learning [1] and our version is simplified to only three patches with strong/weak noises and signals. We put the answer to this problem in two folds:
> 1. How original data model in [1] can approximate real world image-classification data? In the multi-view feature model therein, the authors assume that the images from given class should exhibits some specific signals with high probability. For example, for an image that represents a car, it is highly possible that "wheel", "window", or "headlight" appears some where in the image. Moreover, "wheel" and "wheel" usually share high similarity between images, wheras "wheel" and "window" (or other irrelvant noise/feature) are typically uncorrelated due to the high-dimensionality. This allows us to encode the features with fixed and mutually orthogonal signal vectors and set the noise orthogonal to all the signals. The reviewers can refer to Section 2.3 in [1] for a brief introduction and Appendix A for technical specifications to the data model. We also note that the authors in [1] mentioned that this data model can be considered as the intermediate ouput of previous well-trained convolution layer.
> 2. How does our model reasonably simplify the multi-view feature data model? In short, we proposed several simplifications towards the data model to make our presentation easier to follow without losing the generality.
>     - Simplification 1: We only consider the binary classification with three-patches image data. One can imagine that all the information that is irrelavant to the binary classificaiton of our interests is covered in the noise patch.
>     - Simplification 2: Fixed position signal and orthogonal Gaussian noise. In the original multi-view data model [1], the signals appear randomly among all possible patches, which aligns with the real images. But CNN is unable to identify this positional information because of the "weight sharing" principle, thus our simplication is reasonable.
>     - Simplification 3: No feature noise. The original model assume that the important features are corrupted (additively) by other irrelavant features or noises. However, in our setting, this is not essential in presenting our key mesaage. Suchg simplification can reduce some unnecessary technical issues.
>
> **References:**
>
> [1] Allen-Zhu, Z., & Li, Y. (2022, September). Towards Understanding Ensemble, Knowledge Distillation and Self-Distillation in Deep Learning. In The Eleventh International Conference on Learning Representations.

---

> > ### Author Response · Authors · 2023-11-15
> > **Response to Reviewer 1fN5 (Part 2)**
> >
> > **Q2: The authors discussed some necessary conditions of Assumption 4 on the loss oscillation. Are there any comprehensible sufficient conditions?**
> >
> > **A2:** This is a good question. Intuitively, there do exist sufficient conditions for our assumption. For example, on the simple training data setup, the oscillation phenomenon we assume can be achieved via properly scaling the initialization and the learning rate. i.e.，controlling the learning rate by using the loss smoothness parameters at the initialization and the nearby local minima, so that SGD will neither explode or smoothly converge to the nearby local minima. Then, the magnitude of the oscillation (i.e., $\delta$) can be expressed as a joint function of the loss smoothness, the learning rate, and the initialization. This implies that properly scaling the learning rate and the initialization can potentially give a sufficient condition for the stable oscillation.
> >
> > However, this is the idealized case, where we do not consider the effect of multiple data and multiple features/noises, as well as their individual learning dynamics. Deriving a rigorous and clean sufficient condition could be challenging and very complicated. Therefore, we resort to summarizing the oscillation as an assumption in order to better explain the key message regrading the benefits of oscillation we would like to convey.
> >
> > **Q3: The authors mentioned a bit about the edge-of-stability regime. Can the authors expand a bit more on this? For example, how is the oscillation assumption related to EoS?**
> >
> > **A3:** Thanks for the question and we clarify this as following. The EOS regime of neural network training and the benign oscillation phenomenon we study are related through the *oscillation* phase of the optimization process. Our work can be recognized as the study of the neural network training behaviors after the EOS regime is reached, where we show how the NN can gradually learn useful patterns from data during such an oscillation phase and leads to better generalization. This provides a different angle to study EOS rather than only looking at the largest eigenvalue of the Hessian. Therefore, we believe our research can shed lights on further understanding of EOS, compared to most existing theoretical works that typically focus on how the EOS is reached.

---

> ### Author Response · Authors · 2023-11-20
> **Response to Reviewer 1fN5**
>
> Dear reviewer 1fN5, we thank you again for your time reviewing our paper and your support of our work! As the end of author-review discussion period is approaching, we appreciate it if you can kindly let us know whether your previous concerns have been addressed. We are happy to answer any further questions. Thanks!

---

> > ### Comment · Reviewer_1fN5 · 2023-11-23
> > **Reply to rebuttal**
> >
> > I thank the authors for their explanation. I think it's important and helpful to incorporate the justification of the settings an assumptions in the main text. I don't have further questions, and I'll keep my rating.

---

### Meta-Review · Area_Chair_iSCr · 2023-12-05

**Metareview:**

This paper studies the effects of large learning rate and oscillation in stochastic gradient descent on a specific data generation model. For a neural network on this model, the paper showed that SGD with large learning rates can make correct predictions on test data with only weak features, while SGD with small learning rate could not. The reviewers agree that the paper addresses an important problem with interesting theoretical analysis. The main concerns are whether the feature-learning model is too contrived, and also the necessity to assume stable oscillation.

**Justification For Why Not Higher Score:**

The paper is already at borderline of accept/reject.

**Justification For Why Not Lower Score:**

I might be biased but I liked the paper. My bias: I have written (separated) papers on both large learning rate/edge-of-stability phenomenon and also the feature-learning model. To me, one has to make some structural assumptions on the data and the standard assumptions (e.g., Gaussian, teacher-student) often behave very differently from what's happening in practice. The feature-learning model of this paper is similar to a paper by Yuanzhi and Zeyuan (which they cited) and even in the original paper it was demonstrated that there are many phenomena that happen in this model and real data, but not standard models. I find it to be a good middle-ground for proving something useful while still having potential connections to practice.

---

### Decision · Program_Chairs · 2024-01-16

Accept (poster)